# The Survival Bandit Problem

**Charles Riou**                                                                                                   *charles@ms.k.u-tokyo.ac.jp*
*The University of Tokyo & RIKEN Center for AIP*
*Tokyo, Japan*

**Junya Honda**                                                                                                        *honda@i.kyoto-u.ac.jp*
*Kyoto University & RIKEN Center for AIP*
*Kyoto, Japan*

**Masashi Sugiyama**                                                                                               *sugi@k.u-tokyo.ac.jp*
*RIKEN Center for AIP & The University of Tokyo*
*Tokyo, Japan*

**Reviewed on OpenReview:** *https://openreview.net/forum?id=1qZyJQxOof*

## Abstract

In this paper, we introduce and study a new variant of the multi-armed bandit problem (MAB), called the survival bandit problem (S-MAB). While in both problems, the objective is to maximize the so-called cumulative reward, in this new variant, the procedure is interrupted if the cumulative reward falls below a preset threshold. This simple yet unexplored extension of the MAB follows from many practical applications. For example, when testing two medicines against each other on voluntary patients, people's lives and health are at stake, and it is necessary to be able to interrupt experiments if serious side effects occur or if the disease syndromes are not dissipated by the treatment. From a theoretical perspective, the S-MAB is the first variant of the MAB where the procedure may or may not be interrupted. We start by formalizing the S-MAB and we define its objective as the minimization of the so-called survival regret, which naturally generalizes the regret of the MAB. Then, we show that the objective of the S-MAB is considerably more difficult than the MAB, in the sense that contrary to the MAB, no policy can achieve a reasonably small (i.e., sublinear) survival regret. Instead, we minimize the survival regret in the sense of Pareto, i.e., we seek a policy whose cumulative reward cannot be improved for some problem instance without being sacrificed for another one. For that purpose, we identify two key components in the survival regret: the regret given no ruin (which corresponds to the regret in the MAB), and the probability that the procedure is interrupted, called the probability of ruin. We derive a lower bound on the probability of ruin, as well as policies whose probability of ruin matches the lower bound. Finally, based on a doubling trick on those policies, we derive a policy which minimizes the survival regret in the sense of Pareto, providing an answer to the open problem by Perotto et al. (2019).

## 1 Introduction

Many real life scenarios involve decision-making with partial information feedback, and it thus comes as no surprise that it has been a major field of study over the recent years. The historical motivating example for decision-making under partial information feedback pertains to medicine testing (see, e.g., Villar et al., 2015), and is described as follows. Consider two medicines A and B designed to cure a specific disease. As pointed out by the US Food and Drug Administration[1], before being made available for use by the public, those medicines undergo a very strict procedure composed of four pre-market safety monitoring stages, including

---

[1]https://www.fda.gov/patients/learn-about-drug-and-device-approvals/drug-development-process

a clinical research stage, where the medicines are being tested on people. In this stage, a large number $T$ of patients suffering from the disease are administered either of those medicines sequentially, and the objective is to cure as many patients as possible. During this process, it is crucial to balance two factors in apparent opposition: administering both medicines a sufficient number of times so as to gather information on the efficacy of both medicines, while at the same time, administering in priority whichever of the two medicines seems more effective in order to cure as many patients as possible. The former factor is called exploration, the latter is called exploitation, and the right balance between the two is known as the exploration-exploitation dilemma in the literature. To describe the above, the multi-armed bandit problem (MAB) has arisen in the literature as the most popular model, because of its simplicity and its rich theoretical interest.

In the basic setting of the MAB (see, e.g., Bubeck & Cesa-Bianchi, 2012 or Lattimore & Szepesvari, 2020 for an introduction), there are $K$ unknown distributions $F_1, \ldots, F_K$ bounded in $[-1, 1]$, called arms, and a horizon $T$. In our medicine testing example, each arm $k \in [K] \triangleq \{1, \ldots, K\}$ corresponds to a medicine and the distribution $F_k$ corresponds to its (randomized) effect on a patient. While the horizon $T$ corresponds to the total number of patients and each round $t \leq T$ corresponds to a patient in our example, those rounds are usually interpreted as discrete time steps, or rounds, in the MAB. At each round $t \in [T]$, an agent selects an arm $\pi_t \in [K]$ and observes a reward, denoted by $X_t^{\pi_t}$ and drawn from the distribution $F_{\pi_t}$. In our previous example, the reward $X_t^{\pi_t}$ corresponds to the effect of medicine $\pi_t$ on patient $t$. It can obviously be positive (when the medicine $\pi_t$ cures the patient), but it can also be negative when the patient is not cured by the medicine $\pi_t$ and/or some side effects negatively impact the risk-benefit balance. The objective of the problem is to maximize the expected cumulative reward, defined as $\mathbb{E}\left[\sum_{t=1}^T X_t^{\pi_t}\right]$, where the expectation is taken w.r.t. the arm distributions and the (potential) randomness in the agent's policy $(\pi_t)_{t \geq 1}$. This is equivalent to minimizing the expected cumulative regret compared to an agent who selects the arm $k \in [K]$ with the highest expectation at every round $t \leq T$, defined as $\max_{k \in [K]} \mathbb{E}\left[\sum_{t=1}^T X_t^k\right] - \mathbb{E}\left[\sum_{t=1}^T X_t^{\pi_t}\right]$.

The MAB has been extensively studied over the past decades, and both a lower bound on the regret (see Lai & Robbins, 1985 and Burnetas & Katehakis, 1996) as well as policies matching this lower bound (see, e.g., Cappé et al., 2013, Korda et al., 2013 or more recently Riou & Honda, 2020) have been derived. The MAB is not only a theoretically rich topic, it also has a broad range of applications, from the aforementioned medicine testing (see, e.g., Villar et al., 2015 or Aziz et al., 2021) to advertising and recommender systems (see, e.g., Chapelle & Li, 2011), to name a few.

The most critical aspect of those applications is that you may want to interrupt the process when the cumulative reward $\sum_{t=1}^T X_t^{\pi_t}$ becomes too low. In the medicine testing example, it is necessary to be able to stop the trials early in order to "reduce the number of patients exposed to the unnecessary risk of an ineffective investigational treatment and allow subjects the opportunity to explore more promising therapeutic alternatives", as pointed out in the US Food and Drugs Administration Act (2019, p. 4). In less specialized terms, a bad treatment not only exposes patients to health-threatening side effects, it also prevents them from receiving an efficient treatment to cure them. This poses an additional health threat, because early treatment improves outcomes in many cases, like rheumatoid arthritis, appendicitis, and bacterial pneumonia (see, e.g., Shmerling, 2021). Formally, we set a threshold $B \in \mathbb{R}_+^*$ and decide to interrupt the procedure whenever the cumulative reward becomes lower than or equal to $-B$, i.e., we interrupt the procedure at the first round $\tau$ such that $\sum_{t=1}^\tau X_t^{\pi_t} \leq -B$. In practice, $B$ is chosen prior to the experiment by, e.g., an ethics committee. We use the terminology "survival bandits problem" (S-MAB for short) to designate the MAB with this additional constraint.

While medicine testing was the original practical motivation for this work, portfolio selection in finance is another real-life application of S-MAB, where recent works have demonstrated the strong performance of classic MAB strategies (see, e.g., Hoffman et al., 2011, Shen et al., 2015 or Shen & Wang, 2016), and in particular of risk-aware MAB strategies (see, e.g., Huo & Fu, 2017). In the setting we consider, an investor has an initial budget $B > 0$ to invest sequentially on one of $K$ securities (the arms). At every round $t \leq T$, the investor selects a security $\pi_t$, receives a payoff $X_t^{\pi_t}$ (which can be reinvested), and updates its budget by $X_t^{\pi_t}$. In this setting, a straightforward constraint is imposed on the unsuccessful investor who has to stop investing when it is ruined and has no more money to invest, i.e., $B + \sum_{s=1}^t X_s^{\pi_s} \leq 0$. Following this setting

terminology, we will call the value $B$ "budget" and the first round $\tau$ such that $\sum_{t=1}^{\tau} X_t^{\pi_t} \leq -B$ if it exists "time of ruin".

Please note that both examples above embrace the regime $B \ll T$, since in medicine testing, we want the number of patients who suffer to be much smaller than the total number of voluntary patients, and in portfolio selection, the agent wants to make much more money than its initial budget $B$. In both cases, while the S-MAB procedure may stop after $O(B)$ rounds, it is desirable to achieve a cumulative reward of the order $\Theta(T)$ to fully benefit from the procedure. Consequently, in the S-MAB, the choice of arms $\pi_t$ during the earlier rounds $t$ is absolutely essential, because a bad choice of arms may break the constraint and stop the whole procedure. This is in stark contrast with the standard MAB, where the earlier rounds are used as exploration rounds in order to gather information on the arm distributions, and then to perform efficient exploitation in the later rounds. Precisely, this is the main technicality induced by the new constraint: any successful policy must exploit seemingly good arms from early rounds in order to avoid breaking the constraint and continue the procedure as long as possible. In that sense, rather than the exploration-exploitation dilemma of the MAB, the S-MAB illustrates an exploitation-exploration-exploitation dilemma which, to our knowledge, has never been explored in the literature. It is actually an open problem from the Conference on Learning Theory 2019 to define the problem, establish a (tight) bound on the best achievable performance, and provide policies which achieve that bound (see Perotto et al., 2019). To our knowledge, this paper is the first one to provide answers to this open problem.

## 2 Review of the Literature

At the time of submission of this paper, there are only two works which, to our knowledge, pertain to the S-MAB, and they both focus on the case of rewards in $\{-1, 1\}$. The first one is Perotto et al. (2022). That work introduced a modification of UCB (Auer et al. (2002)) which, empirically, seems to have a low risk of ruin. The second one is Manome et al. (2023). That work derived a generalization of UCB (Auer et al., 2002) and studied its experimental performance in the S-MAB setting, in the case of Bernoulli rewards in $\{-1, 1\}$. Both Perotto et al. (2022) and Manome et al. (2023) are purely experimental studies of the S-MAB. To the best of our knowledge, this work is the first one to bring a theoretical framework to the S-MAB.

Nevertheless, the S-MAB involves a budget $B$, as well as a conservative exploitation at the early rounds, which can also be interpreted as some risk aversion from the agent's viewpoint. It is therefore reasonable to explore the MAB literature related to those topics to see if some existing results can be applied to our setting or not. This may also give some ideas to the interested reader who may want to explore paths beyond the scope of this paper. The list of works cited in this section is by no means an exhaustive list of the existing literature of the aforementioned topics.

The first line of work we introduce here is the budgeted bandits. In this variant of the MAB, the agent is initially given a budget $B$ and at each pull of an arm $\pi_t$, the agent receives a reward $X_t^{\pi_t} \in [0, 1]$ and incurs a positive cost $c_t^{\pi_t} > 0$, such that the cumulative reward at round $t$ is $\sum_{s=1}^{t} X_t^{\pi_t}$ and the remaining budget is $B - \sum_{s=1}^{t} c_t^{\pi_t}$. The procedure stops when the agent's budget becomes negative or zero. Please note that in this variant of the MAB, there is no more horizon $T$ and instead, the objective of the agent is to maximize its expected cumulative reward as a function of the initial budget $B$. Tran-Thanh et al. (2010) first introduced the problem where the costs are deterministic, called the budget-limited bandits, and they introduced a policy of the type ETC (Explore Then Commit, see Garivier et al., 2016). Later, Tran-Thanh et al. (2012) provided a lower bound on the regret as $\Omega(\log B)$, as well as knapsack-based algorithms matching this bound. Ding et al. (2013) generalized that setting to the case of variable costs, called the MAB-BV, and introduced two algorithms which achieve the regret lower bound $\Theta(\log B)$ in the case of multinomial rewards in $\left\{0, \frac{1}{m}, \frac{2}{m}, \ldots, 1\right\}$. Those results were generalized to the case of continuous costs in $[0, 1]$ by Xia et al. (2015a). To solve that problem, an algorithm based on Thompson Sampling (see, e.g., Agrawal & Goyal, 2012) was proposed by Xia et al. (2015b), and algorithms based on UCB (see, e.g., Auer et al., 2002) and $\epsilon$-greedy were proposed by Xia et al. (2017). Cayci et al. (2020) further generalized Xia et al. (2017) to non-negatively correlated costs and rewards, whose $(2 + \gamma)$-moment is finite for some $\gamma > 0$, but such that the expectation of the cost is positive (this ensures that the procedure stops). To wrap up the review of the budgeted bandits literature, we note that several works have considered the generalization of the MAB-BV

to the multiple-play setting, where at each round, the agent pulls $L \geq 2$ of the $K$ arms, including Xia et al. (2016), Zhou & Tomlin (2018) and Rangi et al. (2019).

While both the budgeted bandits and the S-MAB have a budget constraint, in the former, the costs are positive (or have a positive expectation) and the procedure stops when the budget is totally consumed. This is in stark contrast to the S-MAB, whose procedure stops either when the budget is totally consumed, or at the horizon $T$. We can still tackle that issue by increasing the dimension of the budget constraint to 2. Precisely, if pulling arm $\pi_t$ at round $t \leq T$ induces reward $X_t^{\pi_t}$ and decreases the budget by cost $c_t^{\pi_t}$, we can define the initial bidimensional budget as $\tilde{B} \triangleq (B, B)^\top \in \mathbb{R}^2$ such that at each round $t \geq 1$, pulling arm $\pi_t$ induces reward $X_t^{\pi_t}$ and decreases the bidimensional budget by cost $\tilde{c}_t^{\pi_t} \triangleq \left(c_t^{\pi_t}, \frac{B}{T}\right)^\top \in \mathbb{R}^2$. Following that formulation, the procedure stops when one of the two budget components becomes negative or zero. This formulation is actually a particular instance of the bandits with knapsacks (BwK). Formally, the agent is given a multidimensional budget $\tilde{B} = (B, \ldots, B)^\top \in \mathbb{R}^d$ whose components are called resources, and at each round $t$, it incurs a reward $X_t^{\pi_t} \in [0, 1]$ and a cost $\tilde{c}_t^{\pi_t} \in [0, 1]^d$ which decreases all the resources. The objective of the agent is to maximize its cumulative reward before one of the resources run out.

The BwK is the second line of work we present here. It was first introduced in the setting of stochastic costs and rewards by Badanidiyuru et al. (2013). Based on applications in the field of advertising, Combes et al. (2015) studied a particular instance of the BwK where $B = \Omega(T)$, $d = K$, and pulling arm $k$ only affects resource $k$. Later, Sankararaman & Slivkins (2021) derived a problem-dependent regret bound to the knapsack problem, under the assumptions that there are only two resources including time, and that the best distribution over arms is a single arm (best-arm-optimality assumption). Several works in the literature of the BwK have extended the basic stochastic setting to the combinatorial bandits (see Cesa-Bianchi & Lugosi, 2012), including Sankararaman & Slivkins (2018) and Liu et al. (2022a). A large part of the literature on the BwK is related to the contextual bandit setting (see Wang et al., 2005). We mention here the works of Wu et al. (2015) which studied the case of fixed and deterministic costs, Li & Stoltz (2022) which studied a conversion model and its applications to sales discounting, or more recently Han et al. (2023), which provided a black-box algorithm of two online regression algorithms (see Cesa-Bianchi & Orabona, 2021) in the case of a large budget $B = \Omega(T)$. In the particular case where the rewards and costs depend linearly on the contexts, called linear BwK, Agrawal & Devanur (2016) derived an algorithm based on OFUL (see Abbasi-Yadkori et al., 2011) and OMD (Online Mirror Descent, see Cesa-Bianchi & Orabona, 2021). Sivakumar et al. (2022) further studied the linear BwK both in the stochastic and the adversarial settings, in the context of possible sparsity. The BwK in the setting of adversarial costs and rewards (see Auer et al., 1995) was first introduced in the seminal work of Immorlica et al. (2019). Some of their results were later improved by Kesselheim & Singla (2020), which studied an $\ell_p$-relaxation of the adversarial BwK. Finally, Liu et al. (2022b) studied the BwK in a non-stationary environment, where the rewards and costs are generated i.i.d. from a distribution $\mathcal{P}_t$ evolving over time, and which can be considered as a middle ground between the stochastic and adversarial settings.

The BwK is a very rich topic as introduced above, and thus an extremely large pan of the literature is dedicated to that topic. Yet, the costs are always assumed to be positive or to have a positive expectation. In that sense, any of the $d+1$ constraint is enough to ensure the termination of the procedure. This hypothesis is of course not necessary, because the procedure stops automatically at the horizon $T$. Another fundamental difference between the settings considered there and the S-MAB is that the budget $B$ is assumed to scale with $T$, and this ensures that the cumulative reward also scales with $T$. This is in stark contrast to this work, where we consider any value of the budget $B$.

We should still note that, among the many references on the BwK, Kumar & Kleinberg (2022) addressed the case where the costs may be non-positive. In their paper, they derive algorithm ExploreThenControlBudget which achieves a $O(\log T)$ regret bound, yet there is a caveat. They assumed the existence of a "zero arm" (equivalent of the "positive arm" defined in Definition 6 in this paper) which has zero reward and simply increases the budget of each of the resources. This arm removes the risk of exhausting a resource, and in practice, cannot be applied to most of the scenarios motivating this paper and mentioned in the introduction.

Another line of work which is related to the S-MAB is the conservative bandits (Wu et al., 2016). In that setting, the objective is to find a policy which minimizes the regret under the constraint that at any round $t$,

the expected cumulative reward should be larger than $(1_\alpha)R_t^{(b)}$, where $R_t^{(b)}$ denotes the expected cumulative reward of a given baseline and $\alpha \in [0, 1]$ is a parameter. That setting has also been extended to the linear and contextual bandits (Kazerouni et al., 2017, Garcelon et al., 2020). However, this line of work is inapplicable to our problem setting for two reasons. The first one is that the constraint is on the expectation of the rewards and not on the rewards themselves, and therefore, it allows a policy to pull an arm with a seemingly high expected reward even if it has high variance. The second reason is that the constraint in the conservative bandits is relative to a given baseline, and therefore, the existence of a policy satisfying the constraint is trivial: the baseline satisfies it. In contrast, in the S-MAB, no policy can guarantee that there will be no ruin.

The S-MAB is also partly related to the thresholding bandits, introduced in Locatelli et al. (2016), which formalize the problem to identify the arm distributions whose expectations are above a certain threshold $\theta$. While the original objective was to maximize the probability that all such arms are identified, Tao et al. (2019) considered the objective to minimize the sum on all the arms $k$, of the probabilities that arm $k$ is correctly identified. The thresholding bandits is a pure exploration problem, where the objective is not to maximize the expected sum of the rewards but to identify a certain subset of the arms. In contrast, the S-MAB aims at maximizing the expected sum of the rewards under a constraint.

Finally, we conclude this literature review section by mentioning the large body of work related to risk-averse bandits, which do not seek to pull the arm with the largest expectation, but instead to pull the arm maximizing some risk-averse measure. Those include the mean-variance (Sani et al., 2012, Vakili & Zhao, 2016, Zhu & Tan, 2020), functions of the expectation and the variance (Zimin et al., 2014), the moment-generating function at some parameter $\lambda$ (Maillard, 2013), the value at risk (Galichet et al., 2013), or other general measures encompassing the value at risk (Cassel et al., 2018). In the same vein, Sinha et al. (2021) introduced a model with costs and rewards, and they sought to pull the arm with the lowest cost among those with a reasonably large expectation. We also mention Chen et al. (2022), where each arm pull yields a reward and a risk level, and they aimed to pull the arm which has the largest expectation among those whose risk level is lower than a preset level. Those works are not directly related to the S-MAB, however, risk-averse strategies can be good candidates of strategies (or ideas of strategies) to solve the S-MAB in future work.

## 3 Problem Setup and Definitions

Let $T$ be the maximum round, called the horizon, and let $K \geq 1$ be the number of arms, whose indices belong to the set $[K] := \{1, \ldots, K\}$. We denote by $F_1, \ldots, F_K$ the arm distributions, and by $\mu_1, \ldots, \mu_K$ their respective expectations. We assume that $F_1, \ldots, F_K$ are bounded in $[-1, 1]$. We denote by $P_F$ the probability under $F = (F_1, \ldots, F_K)$, and by $P$ in the absence of ambiguity. Let $B > 0$ be the initial budget. Please note that in the whole paper, we will study the problem in the asymptotics of $T$, and we will give a discussion that is non-asymptotic in $B$ and $K$. The S-MAB procedure is defined as follows.

---

An agent starts with a budget $B_0 = B$. Then, at every round $t \in \{1, \ldots, T\}$ while the agent's budget satisfies $B_{t-1} > 0$,

1. the agent selects an arm $\pi_t \in [K]$;

2. the agent observes a reward $X_t^{\pi_t}$ drawn from $F_{\pi_t}$;

3. the agent updates its budget as $B_t := B_{t-1} + X_t^{\pi_t}$.

---

In the above, $(\pi_t)$ is a policy that determines the arm to pull based on the past observations. Please note that, for any $t \in \{0, \ldots, T\}$ such that $B_1, \ldots, B_{t-1} > 0$,

$$B_t = B + \sum_{s=1}^{t} X_s^{\pi_s}.$$

As a result, if $B > T$, then the boundedness of the distributions $F_1, \ldots, F_K$ in $[-1, 1]$ implies that for any $t \geq 0$, $X_t^{\pi_t} \in [-1, 1]$ and therefore, for any $t \in \{0, \ldots, T\}$, $B_t > 0$. In that case, we can remove the constraint $B_t > 0$ on the budget and this boils down to a standard MAB. In that sense, the S-MAB is an extension of the standard MAB.

Conversely, if $B$ is small, the problem becomes much harder. For example, consider the case of Bernoulli arm distributions $F_1, \ldots, F_K$ (of support $\{-1, 1\}$) of respective parameters $p_1, \ldots, p_K \in (0, 1)$. Then no matter the arms $(\pi_t)_{t \geq 1}$ chosen by the agent, it will incur the rewards $-1, \ldots, -1$ ($B$ times) for the first $B$ rounds with probability at least $\min_{1 \leq k \leq K}(1 - p_k)^B > 0$. Consequently, the procedure will stop after $B$ rounds with positive probability.

In a nutshell, the initial budget $B > 0$ is a parameter of the difficulty of the problem. Given some arm distributions fixed, the problem difficulty increases as $B$ decreases. In this paper, we focus on the most difficult case, i.e., when $B > 0$ is small and of constant order in the asymptotic regime $T \to +\infty$.

### 3.1 Definition of the Ruin

The key difference between the S-MAB procedure defined above and the standard MAB is the budget constraint, which states that if the agent's budget $B_t$ becomes negative or zero at some round $t \leq T$, the agent has to stop playing immediately. W.l.o.g., in this paper, we define policies $(\pi_t)$ for any $t \geq 1$ (also beyond $T$) to ensure that the time of ruin of $(\pi_t)$ below is well-defined.

**Definition 1.** *For any policy $\pi$ and any initial budget $B > 0$, the time of ruin is defined as*

$$\tau(B, \pi) := \inf \left\{ t \geq 1 : B + \sum_{s=1}^{t} X_s^{\pi_s} \leq 0 \right\},$$

*where the infimum above is equal to $+\infty$ if the above set is empty. Furthermore, for any $k \in [K]$, we denote by $\tau(B, k)$ the time of ruin of the constant policy $\pi_t = k$ for any $t \geq 1$.*

Using the vocabulary of the time of ruin, the budget constraint simply states that the agent plays until round $\min(T, \tau(B, \pi))$. We can then translate the S-MAB procedure as a standard MAB with horizon $\min(T, \tau(B, \pi))$.

It might thus be tempting to simply use an existing bandit strategy and try to derive a regret bound with the new horizon $\min(T, \tau(B, \pi))$. However, the probability of ruin of a stochastic process until a finite horizon (as opposed to $+\infty$) is in general very complicated. This difficulty is exacerbated when this horizon is random, and even depends on the procedure $\pi$ itself, as is the case with $\min(T, \tau(B, \pi))$. In our setting where the asymptotics $T \to +\infty$ is considered, $P(\tau(B, \pi) < \infty)$ is a reasonable approximation of the probability $P(\tau(B, \pi) \leq T)$ that the ruin occurs before the horizon $T$.

**Definition 2.** *Given a policy $\pi = (\pi_t)_{t \geq 1}$,*

- *the probability of ruin is defined as $P(\tau(B, \pi) < \infty)$;*

- *the probability of survival is defined as $P(\tau(B, \pi) = \infty) = 1 - P(\tau(B, \pi) < \infty)$.*

Please note that, if all the arms $k \in [K]$ have the probability of survival $P(\tau(B, k) = \infty) = 0$, then any policy $\pi$ also has the probability of survival $P(\tau(B, \pi) = \infty) = 0$, and its cumulative reward will be smaller than $-B$. Therefore, in the rest of the paper, we make the following assumption.

**Assumption 1.** *There exists an arm $k \in [K]$ with a positive probability of survival: $P(\tau(B, k) = \infty) > 0$.*

### 3.2 Definition of the Objective

In this subsection, we define the objective of the agent performing the S-MAB procedure. Recall from the previous subsection that the S-MAB can be seen as an extension of the standard MAB with random

horizon $\min(T, \tau(B, \pi))$. The objective of the standard MAB is to maximize the expected cumulative reward $\mathbb{E}\left[\sum_{t=1}^{T} X_t^{\pi_t}\right]$, and we naturally extend this definition to the S-MAB as follows:

$$\mathrm{Rew}_T(\pi) := \mathbb{E}\left[S_T\right] \quad \text{where} \quad S_T \triangleq \sum_{t=1}^{\min(T, \tau(B, \pi))} X_t^{\pi_t} = \sum_{t=1}^{T} X_t^{\pi_t} \mathbb{1}_{\tau(B, \pi) \geq t-1}.$$

In the S-MAB setting, the following remark is fundamental.

**Remark 1.** *Let an agent perform a policy $\pi$. Then, assume that the agent plays until some round $t_0 \geq 1$ and incurs the rewards $X_1^{\pi_1}, \ldots, X_{t_0}^{\pi_{t_0}}$ without being ruined. Further assume that at this precise round $t_0$, the agent realizes that $B_{t_0} > T - t_0$. Then, since the rewards are bounded in $[-1, 1]$, it is clear that, for any $t \in \{t_0 + 1, \ldots, T\}$,*

$$B_t = B_{t_0} + \sum_{s=t_0+1}^{t} X_s^{\pi_s} \geq B_{t_0} - (t - t_0) \geq B_{t_0} - (T - t_0) > 0,$$

*in other words, the agent cannot be ruined anymore. Then, the remaining procedure from round $t_0 + 1$ is a standard MAB (without the risk of ruin), and our agent can perform any standard MAB policy for the remaining rounds $\{t_0 + 1, \ldots, T\}$ and enjoy the cumulative reward of such a policy. On the other hand, if our agent was not aware of the horizon $T$, then it would not be able to verify whether or not the condition $B_{t_0} > T - t_0$ is satisfied. As a result, such an agent would have to care about the risk of ruin until the horizon $T$, and play more conservatively until horizon $T$. For that reason, a policy aware of the horizon $T$ has a significant advantage over one which is not, and hence can achieve a higher cumulative reward.*

In this paper, our focus is on maximizing the expected cumulative reward among all policies which may be aware of the horizon $T$. As a result, instead of focusing on policies $\pi = (\pi_t)_{t \geq 1}$ which are not aware of the horizon $T$, we need to study and formalize the general framework of policies $\pi^T = (\pi_t^T)_{t \geq 1}$ which may depend on the horizon $T$.

**Definition 3.** $\pi = (\pi^T)_{T \geq 1}$ *is a sequence of policies if, for any $T \geq 1, \pi^T = (\pi_t^T)_{t \in \{1, \ldots, T\}}$ satisfies*

$$\forall t \in \{1, \ldots, T\}, \ \pi_t^T \in \sigma(U_1^T, \pi_1^T, X_1^{\pi_1^T}, \ldots, U_{t-1}^T, \pi_{t-1}^T, X_{t-1}^{\pi_{t-1}^T}, U_t^T),$$

*where $\sigma(\cdot)$ is used to denote the sigma algebra and $(U_t^T)$ is some potential randomization for the policy $(\pi_t^T)$ independent from the preceding history.*

In this paper (as in a large part of the MAB literature), we are interested in the asymptotics in $T$. For this reason, we will study the asymptotics of $\mathrm{Rew}_T(\pi^T)$ in $T \to +\infty$, where the policy $\pi^T$ can depend on $T$.

A sequence of policies $\pi = (\pi^T)_{T \geq 1}$ will be called *anytime* if there exists a policy $(\tilde{\pi}_t)_{t \geq 1}$ such that, for any $t \geq 1$ and any $T \geq t$, $\pi_t^T = \tilde{\pi}_t$. We will often identify such a sequence of policies with policy $\tilde{\pi}$, and when we say that a policy $\tilde{\pi}$ is anytime, it formally means that the sequence of policies is written in the above way.

**Remark 2.** *Please note that, in the world of sequences of policies, anytime sequences of policies are the analog of anytime policies. Indeed, given a policy $\pi = (\pi_t)_{t \geq 1}$, let $\pi_t^T \triangleq \pi_t$ for any $T \geq t$. The sequence of policies $(\pi^T)_{T \geq 1}$ where $\pi^T = (\pi_t^T)_{1 \leq t \leq T}$ is anytime and is the equivalent of the anytime policy $\pi$ in the world of sequential policies.*

Now, similarly to the standard MAB, given a sequence of policies $\pi = (\pi^T)_{T \geq 1}$, we want to compare its expected cumulative reward to other sequences of policies $\tilde{\pi} = (\tilde{\pi}^T)_{T \geq 1}$ in order to understand how well it performs. For that purpose, we introduce the regret in the following definition.

**Definition 4.** *Given any sequences of policies $\pi = (\pi^T)_{T \geq 1}$ and $\tilde{\pi} = (\tilde{\pi}^T)_{T \geq 1}$, the relative regret rate of $\pi$ with respect to $\tilde{\pi}$ is defined as*

$$\mathrm{Reg}_F(\pi \| \tilde{\pi}) := \limsup_{T \to +\infty} \frac{\mathrm{Rew}_T(\tilde{\pi}^T) - \mathrm{Rew}_T(\pi^T)}{T}.$$

First, please note that this regret is asymptotic in $T$, because we want to capture the main term in $T$ in the reward. Secondly, in Definition 4, we compare the reward of the sequence of policies $\pi$ with the reward of some sequence of policies $\tilde{\pi}$. Ideally, our objective would be to find a sequence of policies $\pi$ which has a higher reward than any other sequence of policies $\tilde{\pi}$, in other words, which would satisfy $\mathrm{Reg}_F(\pi\|\tilde{\pi}) \leq 0$ for any arm distributions $F = (F_1, \ldots, F_K)$ and any sequence of policy $\tilde{\pi}$.

In the standard MAB (without the risk of ruin), we know that such policies $\pi$ exist and even second-order terms (in $T$) in the reward decomposition have been derived such that $\mathrm{Rew}_T(\pi) = \max_{k \in [K]} \mu_k T + O(\log T)$ (see, e.g., Auer et al., 2002). In contrast, in the S-MAB setting (with the risk of ruin), the policy which achieves the best bound is not known. Actually, we do not even know (yet) in what sense there exists a "best" bound.

As a matter of fact, contrary to the MAB, in the S-MAB, no sequence of policies $\pi$ dominates all the other sequences of policies $\tilde{\pi}$, as stated in the next proposition, whose proof is given in Appendix B.

**Proposition 1.** *For any sequence of policies $\pi$, $\sup_F \sup_{\pi'} \mathrm{Reg}_F(\pi\|\pi') > 0$.*

The key message is that whatever the sequence of policies $\pi$ that you choose, there exist some arm distributions $F = (F_1, \ldots, F_K)$ such that another sequence of policies $\tilde{\pi}$ has a significantly higher reward (by a term of order $\Omega(T)$) than $\pi$. It is now hopeless to look for a policy which is "absolutely better" than any other one and instead, we look for a Pareto-optimal policy, as defined below.

**Definition 5.** *A sequence of policies $\pi$ is said to be (regret-wise) Pareto-optimal if, for any sequence of policies $\pi'$,*

$$\sup_F \mathrm{Reg}_F(\pi\|\pi') > 0 \implies \inf_F \mathrm{Reg}_F(\pi\|\pi') < 0.$$

The notion of Pareto-optimal sequences of policies is related to the concept of strictly dominated strategies in game theory. Assume that an agent performs a sequence of policies $\pi$ and that there exists another sequence of policies $\pi'$ such that:

- for any arm distributions $F'$, $\mathrm{Reg}_{F'}(\pi\|\pi') \geq 0$, and

- for some arm distributions $F$, it even holds that $\mathrm{Reg}_F(\pi\|\pi') > 0$.

Then, the agent should simply not use the sequence of policies $\pi$ and instead use $\pi'$, because it is:

- always at least as good as $\pi$, and

- in some cases, even strictly better than $\pi$.

In the language of game theory, $\pi$ is called a strictly dominated strategy, and our goal is to find a strategy which is not strictly dominated, which we call here a (regret-wise) Pareto-optimal sequence of policies. This challenging problem is an open question from COLT 2019 (Perotto et al., 2019), and our solution is based on proof techniques from various fields including information theory, stochastic processes theory or even predictions. As such, the intermediate results of this paper can be of interest for the reader interested in any of those fields and their applications.

## 4 Strategy and Structure of the Paper

In this section and until Section 7, we study the case of multinomial arm distributions of support $\{-1, 0, 1\}$, referred to as multinomial distributions in this paper. The generalization to arm distributions bounded in $[-1, 1]$ is done in Section 9.

We first explain the choice of the reward model (Section 4.1), and why we made Assumption 1 and what the problem objective becomes when this objective does not hold (Section 7.2.4). We next give an intuition on why the study of the probability of ruin is fundamental to maximize the expected cumulative reward (Section 4.2), and we conclude with the structure of the rest of the paper (Section 4.3).

### 4.1 Reward Models

In this paper, we will mostly focus on multinomial arm distributions of support $\{-1, 0, 1\}$ (simply referred to as multinomial arm distributions in the rest of this paper). This subsection discusses why this setting is of particular interest.

**Why it is reasonable to study Integer Rewards.** W.l.o.g., let us assume that the budget $B \in \mathbb{N}$, and consider the specific instance of two arms, $F_1$ and $F_2$, of respective support $\{-1, 0, 1\}$ and $\{-1/2, 0, 1/2\}$. Assume that at some time $t_0$, the cumulative reward of a policy $\pi$ reaches

$$\sum_{t=1}^{t_0} X_t^{\pi_t} = -B + 1.$$

Then, at round $t_0 + 1$, selecting arm 1 increases the risk of ruin, while selecting arm 2 does not. At that stage, it is optimal to select arm 2, even if $\mu_1 > 0$ and $\mu_2 < 0$. Such situations may occur in the more general setting of arm distributions bounded in $[-1, 1]$, when the cumulative reward falls in the interval $(-B, -B + K)$. This creates a gap between our algorithms' upper bounds and our lower bounds. However, this gap is small, as we justify in Section 9.

**Why Bernoulli rewards are not sufficient.** As explained in Section 4.2, minimizing the probability of ruin is crucial to maximize the expected cumulative reward of the S-MAB. In the case of multinomial arm distributions, the best constant strategy in hindsight which minimizes the probability of ruin selects arm

$$k^P \triangleq \underset{k \in [K]}{\arg\max} \, \frac{P_{F_k}(X = +1)}{P_{F_k}(X = -1)},$$

(see Lemma 4), while the one which maximizes the expected cumulative reward selects arm

$$k^E \triangleq \underset{k \in [K]}{\arg\max} \, \{P_{F_k}(X = +1) - P_{F_k}(X = -1)\}.$$

In general, $k^P \neq k^E$, and this phenomenon is also captured by the multinomial setting we consider. On the other hand, Bernoulli arm distributions only have one real parameter, and do not capture this phenomenon, essential for the understanding of the survival bandit problem.

### 4.2 Strategy of the Paper

By definition of the time of ruin $\tau(B, \pi)$, it holds that

$$\sum_{t=1}^{T \wedge \tau(B,\pi)} X_t^{\pi_t} \geq -B. \tag{1}$$

Furthermore, since the rewards are bounded in $[-1, 1]$, if $\tau(B, \pi) > T$, then it holds that

$$\sum_{t=1}^{T \wedge \tau(B,\pi)} X_t^{\pi_t} = \sum_{t=1}^{\tau(B,\pi)} X_t^{\pi_t} - \sum_{t=T+1}^{\tau(B,\pi)} X_t^{\pi_t} \leq -B + (\tau(B,\pi) - T). \tag{2}$$

Finally, since the rewards are bounded in $[-1, 1]$, it is clear that

$$\sum_{t=1}^{T \wedge \tau(B,\pi)} X_t^{\pi_t} \leq T. \tag{3}$$

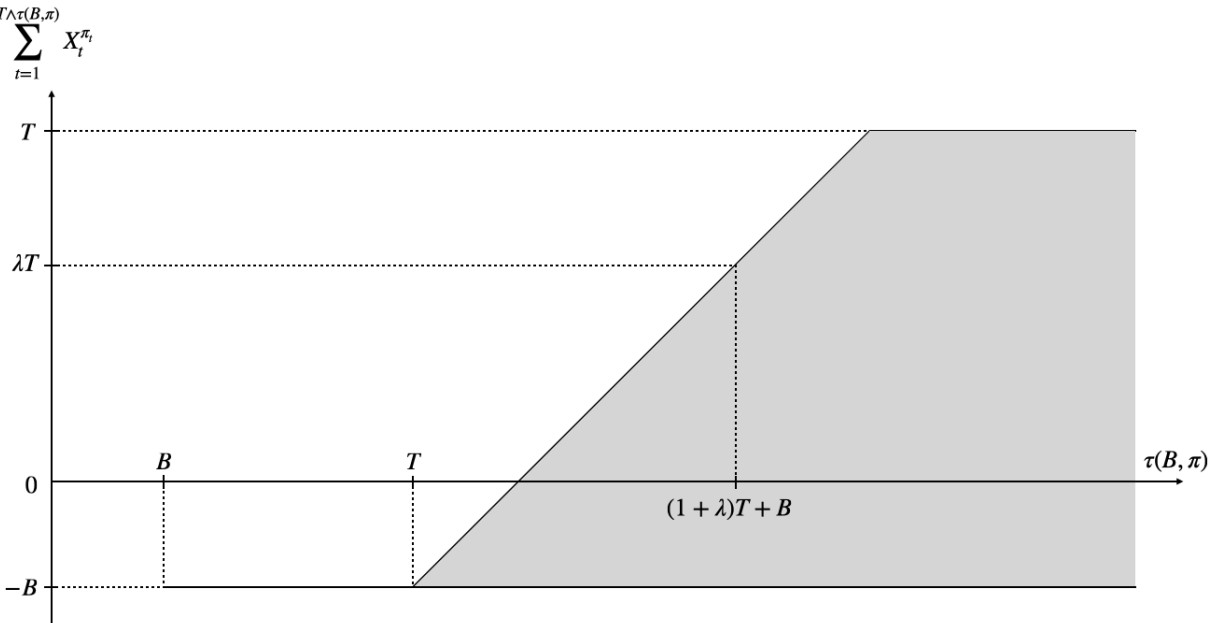

Figure 1: The probability of ruin constraints the cumulative reward.

Eq. (1), (2) and (3) draw a set where the cumulative reward is constrained, shaded in gray in Figure 1. That figure shows that a low time of ruin prevents the cumulative reward to become large. In particular, to have an expected cumulative reward which is larger than $\lambda T$ (for some positive $\lambda$), Figure 1 shows that $\tau(B, \pi)$ needs to be bigger that $\lambda T + B$ with large probability, i.e., the following needs to be large:

$$P\left(\tau(B, \pi) \geq (1 + \lambda)T + B\right).$$

While we do not know $\lambda$ yet, $(1+\lambda)T+B$ is large when $T$ is large (standard assumption in bandit problems), and therefore, the above quantity is largely related to the probability of ruin $P(\tau(B, \pi) < \infty)$.

### 4.3 Structure of the Paper

We start by studying the case of multinomial arm distributions of support $\{-1, 0, 1\}$ (referred to as multinomial distributions in the sequel) in Sections 5 to 7, and then we will extend some of our results to the general case of arm distributions bounded in $[-1, 1]$ in Section 9. The outline of the paper is as follows:

- in Section 5, we study the probability of ruin and derive the first main result of this paper: a tight non-asymptotic lower bound (in the sense of Pareto) on the probability of ruin in Theorem 1.

- in Section 6, we introduce EXPLOIT, a framework (or set) of policies whose probability of ruin matches the lower bound of Section 5. We further show that those policies cannot be regret-wise Pareto-optimal, and hence the need to adapt them to achieve our desired objective.

- in Section 7, we introduce the policy EXPLOIT-UCB-DOUBLE, which performs a doubling trick over an EXPLOIT policy. We show the second main result of this paper in Theorem 3: EXPLOIT-UCB-DOUBLE is regret-wise Pareto-optimal. We corroborate this theoretical result with experiments showing the experimental performance of EXPLOIT-UCB-DOUBLE. This section provides an answer to an open problem from COLT 2019 (Perotto et al., 2019).

- in Section 9, we generalize the results of Sections 5–7 to the general case of arm distributions bounded in $[-1, 1]$. In particular, we generalize the result of Theorem 1 on the probability of ruin, which is the third main result of this paper, and further relate it to the probability of ruin of i.i.d. stochastic processes, which is a result of independent interest. Finally, we discuss the challenges of extending Theorem 3 to that setting.

# 5   Study of the Probability of Ruin

In this section, we study the probability of ruin of anytime policies in the case of multinomial arm distributions (of support $\{-1, 0, 1\}$). We first derive one of the main results of this paper: a tight lower bound on the probability of ruin (Theorem 1), which will be generalized to distributions bounded in $[-1, 1]$ in Proposition 7 of Section 9. We further relate this bound to the probability of ruin of i.i.d. random walks on $\{-1, 0, 1\}$ in Lemma 1, which is a result of independent interest.

## 5.1   Trivial Case and Assumption

Consider the following example.

**Example 1.** *Assume that there exists an arm $k \in [K]$ whose distribution $F_k$ is such that $P_{X \sim F_k}(X \geq 0) = 1$. Then, if the initial budget $B$ satisfies $B > K-1$, the policy $\pi$ which chooses the arm with the highest empirical average of rewards, i.e., defined by*

$$\forall t \geq 1, \ \pi_t \triangleq \arg\max_{k \in [K]} \frac{\sum_{s=1}^{t-1} X_s^{\pi_t} \mathbb{1}_{\pi_s = k}}{\sum_{s=1}^{t-1} \mathbb{1}_{\pi_s = k}},$$

*has no risk of ruin: $P(\tau(B, \pi) < \infty) = 0$.*

The above case is simple, because the distribution $F_k$ only yields non-negative rewards. In this case, the trivial bound on the probability of ruin

$$P(\tau(B, \pi) < \infty) \geq 0$$

is tight, and we eliminate this case by assuming that there is no positive or zero arm distribution, as defined below.

**Definition 6.** *A distribution $F$ is called a zero distribution (resp. a positive distribution) if $P_{X \sim F}(X = 0) = 1$ (resp. if $P_{X \sim F}(X \geq 0) = 1$ and $P_{X \sim F}(X > 0) > 0$). We say that an arm $k$ is a zero arm (resp. a positive arm) if its distribution $F_k$ is zero (resp. positive).*

In this section, we make the following assumption.

**Assumption 2.** *There is no positive or zero arm.*

We will also consider the following weaker assumption.

**Assumption 3.** *There is no zero arm.*

Please note that the policies introduced in Sections 6 to 7 will achieve $P(\tau(B, \pi) < \infty) = 0$ in the case where there is a positive or zero arm.

## 5.2   Main Results on the Probability of Ruin

Let $\mathcal{F}_{\{-1,0,1\}}$ be the set of multinomial distributions of support $\{-1, 0, 1\}$ which are not positive or zero (see Definition 6). Similarly to the cumulative reward, for any $F = (F_1, \ldots, F_K) \in \mathcal{F}_{\{-1,0,1\}}^K$ and any policies $\pi, \pi'$, we define the *relative risk of ruin* of $\pi$ with respect to $\pi'$ as

$$P_{\text{ruin}}(\pi \| \pi') := P_F(\tau(B, \pi) < \infty) - P_F(\tau(B, \pi') < \infty),$$

where the dependency on $F$ is omitted in the notation. Please note that, since we study this problem for fixed $B$ small with regards to $T$, there is no limit in the definition of $P_{\text{ruin}}(\pi \| \pi')$. Yet, similarly to Proposition 1, we can prove that no policy $\pi$ achieves $\sup_{\pi'} \sup_F P_{\text{ruin}}(\pi \| \pi') \leq 0$, and for that reason, we focus on a policy which is Pareto-optimal in the sense of the probability of ruin, which is formalized as follows.

**Definition 7.** *A policy $\pi$ is said to be ruin-wise Pareto-optimal if, for any policy $\pi'$,*

$$\sup_F P_{\text{ruin}}(\pi \| \pi') > 0 \implies \inf_F P_{\text{ruin}}(\pi \| \pi') < 0.$$

Before stating the main result of this section, we need to define, for any arm distributions $F = (F_1, \ldots, F_K) \in \mathcal{F}_{\{-1,0,1\}}^K$,

$$\gamma(F_k) := \inf_{Q:\mathbb{E}_{X \sim Q}[X] < 0} \frac{\mathrm{KL}(Q \| F_k)}{\mathbb{E}_{X \sim Q}[-X]} \geq 0. \tag{4}$$

The main result of this section is a Pareto-type lower bound on the probability of ruin.

**Theorem 1.** *Let $(\alpha_k)_{k \in [K]}$ be such that for any $k$, $\alpha_k > 0$ and $\sum_{k=1}^K \alpha_k = 1$. For any policy $\pi$,*

$$\inf_{F \in \mathcal{F}_{\{-1,0,1\}}^K} \left\{ P_F(\tau(B, \pi) < \infty) - \exp\left( -B \sum_{k=1}^K \alpha_k \gamma(F_k) \right) \right\} < 0$$

$$\implies \sup_{F \in \mathcal{F}_{\{-1,0,1\}}^K} \left\{ P_F(\tau(B, \pi) < \infty) - \exp\left( -B \sum_{k=1}^K \alpha_k \gamma(F_k) \right) \right\} > 0. \tag{5}$$

The main ingredient of the proof of Theorem 1 is given in Section 5.4, and the rest of the proof is given in Appendix D.

Please note that this theorem gives a lower bound on the probability of ruin. A weaker version of Theorem 1 is that there exist some arm distributions $F \in \mathcal{F}_{\{-1,0,1\}}^K$ such that

$$P_F(\tau(B, \pi) < \infty) \geq \exp\left( -B \sum_{k=1}^K \alpha_k \gamma(F_k) \right). \tag{6}$$

But Theorem 1 is a little stronger than that. Actually, it states that there are two cases:

- either the lower bound (6) holds for *all* arm distributions $F \in \mathcal{F}_{\{-1,0,1\}}^K$;

- or it holds with strict inequality for some $F \in \mathcal{F}_{\{-1,0,1\}}^K$.

We conclude this subsection by a lemma providing an insightful interpretation of the lower bound (6).

**Lemma 1.** *For any distributions $F = (F_1, \ldots, F_K) \in \mathcal{F}_{\{-1,0,1\}}^K$ and any arm $k \in [K]$,*

$$\frac{1}{B} \log P_{F_k}(\tau(B, k) < \infty) = -\gamma(F_k).$$

**Remark 3.** *Whereas the statement of Lemma 1 uses the KL divergence through the definition of $\gamma(F_k)$ in (4), the probability of ruin of a stochastic process is usually analyzed using the moment-generating function, which is also found in the proof of this lemma given in Appendix E. The relation between them is discussed in Lemma 5 in Appendix C, and we interchangeably use both representations.*

In the next subsection, we give an interpretation of the bound of Theorem 1. In particular, we explain the Pareto-type bound and its connection to game theory in Section 5.3.1, we give an interpretation of the $\gamma(F_k)$ from the bound in Section 5.3.2, and in Section 5.3.3, we give an interpretation of Theorem 1 based on Lemma 1 which will give rise to the policies matching this bound in Section 6.

## 5.3 Interpretation of the Lower Bound

In this subsection, we give an interpretation of the lower bound of Theorem 1 and the coefficient $\gamma(F_k)$ that appear in the bound.

### 5.3.1   A Pareto-type Bound

The lower bound is of Pareto-type. It states that for any policy $\pi$, if there exist some arm distributions $F = (F_1, \ldots, F_K)$ such that the probability of ruin of $\pi$ is lower than

$$\exp\left(-B\sum_{k=1}^{K}\alpha_k\gamma(F_k)\right),$$

then there exist some other arm distributions $G = (G_1, \ldots, G_K)$ such that the probability of ruin of $\pi$ is larger than

$$\exp\left(-B\sum_{k=1}^{K}\alpha_k\gamma(G_k)\right).$$

Such a lower bound is said to be of Pareto-type, and corresponds to the notion of strict dominance in game theory (von Neumann & Morgenstern, 1944): consider a game between one player and the environment, which is articulated as follows:

1. simultaneously and without consulting with one another, the player chooses a policy $\pi$ and the environment chooses a $K$-tuple of arm distributions $F = (F_1, \ldots, F_K)$;

2. the player receives the score

$$\exp\left(-B\sum_{k=1}^{K}\alpha_k\gamma(F_k)\right) - P_F(\tau(B,\pi) < \infty).$$

The objective of the player is to maximize its score. Then, Theorem 1 states that no strategy $\pi$ can ensure a score which is always non-negative and sometimes positive. In other words, if the score is money (in USD), then there is no strategy which guarantees that the player will almost surely make money.

In terms of policy, this translates as non strict dominance. Let $\pi$ and $\pi'$ be two policies, and assume that

- for any $F$ chosen by the environment, the score of $\pi$ is as good as $\pi'$;

- and there exists some $F$ such that the score of $\pi$ is (strictly) larger than the score of $\pi'$.

In this case, we say that $\pi'$ is *strictly dominated* by $\pi$, in the sense that $\pi$ is always at least as good as $\pi'$, and sometimes strictly better than $\pi'$. In such a situation, from a score maximization perspective, it is always better to choose policy $\pi$ over policy $\pi'$, and we can discard policy $\pi'$ as an unreasonable choice of a policy. As a consequence, before choosing a policy $\pi$, it is fundamental to determine if it is strictly dominated.

In Section 6, we prove that there exists a policy whose probability of ruin is equal to

$$\exp\left(-B\sum_{k=1}^{K}\alpha_k\gamma(F_k)\right)$$

for any arm distributions $F = (F_1, \ldots, F_K)$, and therefore, such a policy is not strictly dominated and reasonable from the perspective of the risk of ruin.

### 5.3.2   The Coefficients Gamma

In this subsection, we give an interpretation of the quantities $\gamma(F_k)$ from the bound of Theorem 1. If $F_k \in \mathcal{F}_{\{-1,0,1\}}$, a direct application of Lemmas 1 and 4 yields

$$\gamma(F_k) = -\frac{1}{B}\log P_{F_k}(\tau(B,k) < \infty) = \max\left(0, \log\frac{P_{X\sim F_k}(X = +1)}{P_{X\sim F_k}(X = -1)}\right).$$

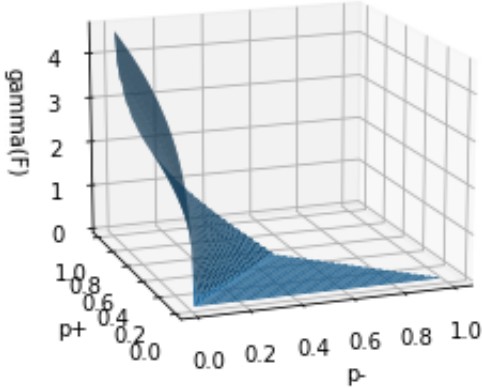

Figure 2: $\gamma(F_k)$ as a function of $(p_k^-, p_k^+)$.

For any $F_k \in \mathcal{F}_{\{-1,0,1\}}, \gamma(F_k) \in [0, +\infty]$ is a measure of the chance of survival of the policy which consistently selects arm $k$. The following equivalence holds:

$$P_{F_k}(\tau(B, k) < \infty) = 1 \iff P_{X \sim F_k}(X = +1) \leq P_{X \sim F_k}(X = -1) \iff \gamma(F_k) = 0,$$

in other words, the constant policy which only selects arm $k$ reaches the ruin systematically if and only if $\gamma(F_k) = 0$. At the other end of the spectrum,

$$P_{F_k}(\tau(B, k) < \infty) = 0 \iff P_{X \sim F_k}(X = -1) = 0 \iff \gamma(F_k) = +\infty,$$

and the constant policy which only selects arm $k$ never observes the ruin if and only if $\gamma(F_k) = +\infty$.

Between those extreme cases, $\gamma(F_k)$ is a function which is non-decreasing in $p_k^+ \triangleq P(X = +1)$ and non-increasing in $p_k^- \triangleq P(X = -1)$. The whole plot of $\gamma(F_k)$ as a function of $(p_k^-, p_k^+)$ is represented in Figure 2.

### 5.3.3 Divide the Budget and Conquer

In this subsection, we provide an interpretation of the bound of Theorem 1 which gives rise to a class of policies matching this bound. Let $(\alpha_k)_{k \in [K]}$ be such that for any $k \in [K], \alpha_k B \in \mathbb{N}$. An application of Lemma 1 to the budget $\alpha_k B$ for any $k$ yields

$$\prod_{k=1}^{K} P_{F_k}(\tau(\alpha_k B, k) < \infty) = \exp\left(-B \sum_{k=1}^{K} \alpha_k \gamma(F_k)\right). \tag{7}$$

This gives another expression of the lower bound in (6), which is easier to interpret. This lower bound is the product on the arms $k \in [K]$ of the probability of ruin of the constant policy $\pi_t = k$ with budget $\alpha_k B$. This suggests that it might be interesting to divide the budget $B$ into shares $(\alpha_k B)_{k \in [K]}$, and to allocate to each arm a budget share $\alpha_k B$. This fundamental interpretation will be the basis to define the EXPLOIT policies in Section 6.

Finally, please note that each of the factors in (7) can further be computed explicitly, using Lemma 4:

$$\exp\left(-B \sum_{k=1}^{K} \alpha_k \gamma(F_k)\right) = \prod_{k=1}^{K} \min\left(1, \left(\frac{P_{X \sim F_k}(X = -1)}{P_{X \sim F_k}(X = 1)}\right)^{\alpha_k B}\right).$$

Nevertheless, there is no explicit formula for $\gamma(F_k)$ in the general case of non-multinomial rewards. The non-multinomial case is discussed in Section 9.

### 5.4 Sketch and Main Ingredient of the Proof of Theorem 1

The proof of the non-asymptotic bound of Theorem 1 is given in Appendix D. This proof consists of (i) derivation of an asymptotic bound given in Lemma 2 below, and (ii) turning it into a non-asymptotic bound by using a sub-additivity argument on the probability of ruin. Please note that the proof of this lemma is conducted in the generality of distributions bounded in $[-1, 1]$ (not necessarily multinomial).

**Lemma 2.** *Fix an arbitrary policy $\pi$ and distributions $(Q_1, \ldots, Q_K)$ such that $\mathbb{E}_{X \sim Q_k}[X] < 0$ for all $k \in [K]$. Then, there exists a probability vector $\beta(Q) = (\beta_1(Q), \ldots, \beta_K(Q))$ such that for any distributions $(F_1, \ldots, F_K)$,*

$$\liminf_{B \to +\infty} \frac{1}{B} \log P_{(F_1, \ldots, F_K)} \left( \tau(B, \pi) < \frac{3B}{\Delta_Q} \right) \geq - \sum_{k=1}^{K} \beta_k(Q) \frac{\mathrm{KL}(Q_k \| F_k)}{\mathbb{E}_{X \sim Q_k}[-X]},$$

*where $\Delta_Q = \min_{i \in [K]} \mathbb{E}_{X \sim Q_i}[-X] > 0$.*

*Proof.* Let $Q = (Q_1, \ldots, Q_K)$ be a vector of distributions such that $\mathbb{E}_{X \sim Q_k}[X] < 0$ for all $k \in [K]$, and let $\Delta_Q := \min_{i \in [K]} \mathbb{E}_{X \sim Q_i}[-X] > 0$. We denote by $N_k(\tau)$ the number of pulls of arm $k$ until $\tau(B, \pi)$, and by $n_k$ its realization. Denoting by $Y_k^n$ the reward of the $n$-th pull of arm $k$, let

$$\mathcal{H}_\tau = \left( \left( Y_1^1, Y_1^2, \ldots, Y_1^{N_1(\tau)} \right), \left( Y_2^1, Y_2^2, \ldots, Y_2^{N_2(\tau)} \right), \ldots, \left( Y_K^1, Y_K^2, \ldots, Y_K^{N_K(\tau)} \right) \right),$$

and $h_t$ be its realization. Please note that for any realization $h_t$, $\left| B + \sum_{k \in [K]} \sum_{m \in [n_k]} y_k^m \right| \leq 1$. We further denote by $T(Q)$ the set of "typical" realizations $h_t$ satisfying

$$\begin{cases} \left| \sum_{k=1}^{K} \left( n_k \mathrm{KL}(Q_k \| F_k) - \sum_{m=1}^{n_k} \log \frac{dQ_k}{dF_k}(y_k^m) \right) \right| \leq \frac{t}{B^{\frac{1}{4}}}, \\ \left| \sum_{k=1}^{K} \left( n_k \mathbb{E}_{X \sim Q_k}[X] - \sum_{m=1}^{n_k} y_k^m \right) \right| \leq \frac{t \Delta_Q}{B^{\frac{1}{4}}}, \\ \sum_{k=1}^{K} \sum_{m=1}^{n_k} y_k^m \leq -\frac{t \Delta_Q}{2}. \end{cases} \tag{8}$$

Such realizations are "typical" under $Q$ in the sense that $\lim_{B \to +\infty} Q(\mathcal{H}_\tau \in T(Q)) = 1$ (shown by, e.g., Hoeffding's inequality, see Appendix D.1 for details). We can see from (8) that any typical $h_t$ satisfies

$$t \leq \frac{3B}{\Delta_Q} \quad \text{and} \quad \left| \sum_{k=1}^{K} \frac{n_k}{B} \mathbb{E}_{X \sim Q_k}[-X] - 1 \right| \leq \frac{4}{B^{\frac{1}{4}}}. \tag{9}$$

In particular, denoting $r(h_t) := \frac{(n_1, \ldots, n_K)}{B}$, (9) implies that $r(h_t)$ can take at most $O(\mathrm{poly}(B))$ values, and hence there exists $\tilde{r}$ such that

$$\lim_{B \to +\infty} \frac{1}{B} \log Q \left( r(\mathcal{H}_\tau) = \tilde{r} | \mathcal{H}_\tau \in T(Q) \right) = 0. \tag{10}$$

By performing a change of distribution and using (8), we can bound

$$P_{(F_1, \ldots, F_K)} \left( \tau(B, \pi) < \frac{3B}{\Delta_Q} \right)$$

$$\geq P_{(F_1, \ldots, F_K)} \left( \mathcal{H}_\tau \in T(Q), r(\mathcal{H}_\tau) = \tilde{r} \right)$$

$$= \sum_{\substack{h_t \in T(Q): \\ r(h_t) = \tilde{r}}} Q(h_t) \exp \left( - \sum_{k=1}^{K} \sum_{m=1}^{n_k} \log \frac{dQ_k}{dF_k}(y_k^m) \right)$$

$$\geq \sum_{\substack{h_t \in T(Q): \\ r(h_t) = \tilde{r}}} Q(h_t) \exp \left( - \sum_{k=1}^{K} n_k \mathrm{KL}(Q_k \| F_k) - \frac{t}{B^{\frac{1}{4}}} \right)$$

$$\geq \sum_{\substack{h_t \in T(Q): \\ r(h_t)=\tilde{r}}} Q(h_t) \exp\left\{-B\left(\sum_{k=1}^{K} \tilde{r}_k \mathrm{KL}(Q_k\|F_k) + \frac{3}{\Delta_Q B^{\frac{1}{4}}}\right)\right\}$$

$$= \exp\left\{-B\left(\sum_{k=1}^{K} \tilde{r}_k \mathbb{E}_{X\sim Q_k}[-X]\frac{\mathrm{KL}(Q_k\|F_k)}{\mathbb{E}_{X\sim Q_k}[-X]} + \frac{3}{\Delta_Q B^{\frac{1}{4}}}\right)\right\} Q\left(\mathcal{H}_\tau \in T(Q), r(\mathcal{H}_\tau) = \tilde{r}\right).$$

For any $k \in [K]$, we introduce the normalized version of $\tilde{r}_k \mathbb{E}_{X\sim Q_k}[-X]$, which we denote by

$$\beta_k(Q) := \frac{\tilde{r}_k \mathbb{E}_{X\sim Q_k}[-X]}{\sum_{j=1}^{K} \tilde{r}_j \mathbb{E}_{X\sim Q_j}[-X]}.$$

Eq. (9) implies that $\left|\sum_{j=1}^{K} \tilde{r}_j \mathbb{E}_{X\sim Q_j}[-X] - 1\right| \leq \frac{4}{B^{\frac{1}{4}}}$, and hence

$$\frac{1}{B} \log P_{(F_1,\ldots,F_K)}\left(\tau(B,\pi) < \frac{3B}{\Delta_Q}\right)$$

$$\geq -\left(1 + \frac{4}{B^{\frac{1}{4}}}\right)\sum_{k=1}^{K} \beta_k(Q)\frac{\mathrm{KL}(Q_k\|F_k)}{\mathbb{E}_{X\sim Q_k}[-X]} - \frac{3}{\Delta_Q B^{\frac{1}{4}}} + \frac{1}{B}\log Q\left(\mathcal{H}_\tau \in T(Q), r(\mathcal{H}_\tau) = \tilde{r}\right)$$

$$\xrightarrow{B\to+\infty} -\sum_{k=1}^{K} \beta_k(Q)\frac{\mathrm{KL}(Q_k\|F_k)}{\mathbb{E}_{X\sim Q_k}[-X]},$$

by (10), concluding the proof of Lemma 2. $\qquad\square$

## 6 The EXPLOIT Framework

In this section, we introduce a framework of anytime policies, called EXPLOIT, which achieve the lower bound on the probability of ruin given in Theorem 1. We further study the regret of all such policies.

In Sections 6 and 7, our study is conducted in the case of multinomial arm distributions, and therefore, it holds that

$$P(\tau(B,\pi) \leq T) = P(\tau(\lceil B\rceil, \pi) \leq T)$$

for any policy $\pi$. As a result, in these sections, we assume w.l.o.g. that $B \in \mathbb{N}$.

### 6.1 Definition of the EXPLOIT Framework

Let $B_1, \ldots, B_K$ be arbitrary positive integers such that $B_1 + \cdots + B_K = B$. Applying (7) to $\alpha_k \triangleq \frac{B_k}{B}$ for any $k \in [K]$, the lower bound of Theorem 1 implies that

$$P_F(\tau(B,\pi) < \infty) \geq \prod_{k=1}^{K} P_{F_k}(\tau(B_k, k) < \infty)$$

for some arm distributions $F \in \mathcal{F}_{\{-1,0,1\}}^K$. This right hand-side of this inequality corresponds to the probability of ruin of any policy $\pi$ which allocates budget $B_1$ to arm 1, $B_2$ to arm 2 and so on, and plays $\pi_t = k$ only if arm $k$ has not exceeded its budget $B_k$. We say that such policies belong to the EXPLOIT framework, which is formalized below.

**Definition 8.** *Given some positive integers $B_1, \ldots, B_K$ such that $B_1 + \cdots + B_K = B$, we say that a policy $\pi = (\pi_t)_{t\geq 1}$ belongs to the framework EXPLOIT$(B_1, \ldots, B_K)$ if, at any round $t \geq 1$,*

$$\pi_t \in \left\{k \in [K] : B_k + \sum_{s=1}^{t-1} X_s^{\pi_s} \mathbb{1}_{\pi_s=k} > 0\right\}.$$

**Remark 4.** *The initial choice of the budget shares $B_1, \ldots, B_K$ may depend on some possible prior information we may have on the arms. Assume that there is a prior distribution $\mathcal{D}_k$ for $\gamma(F_k)$ through, e.g., already observed samples. We further assume that the priors are independent. Then the expected probability of ruin is given by*

$$
\mathbb{E}_{\substack{\gamma(F_1)\sim\mathcal{D}_1 \\ \gamma(F_K)\sim\mathcal{D}_K}} \left[ \prod_{k=1}^{K} \exp\left(-B_k\gamma(F_k)\right) \right] = \prod_{k=1}^{K} \mathbb{E}_{\gamma(F_k)\sim\mathcal{D}_k} \left[ \exp\left(-B_k\gamma(F_k)\right) \right].
$$

*Then, it seems reasonable to take $B_k$ minimizing this quantity, which is trivially $B_k = B/K$ when $\mathcal{D}_k$ is the same for all the arms $k$. In general, one can easily see that the optimal $B_k$ is the one such that*

$$
\frac{\mathbb{E}\left[\gamma(F_k)\exp\left(-B_k\gamma(F_k)\right)\right]}{\mathbb{E}\left[\exp\left(-B_k\gamma(F_k)\right)\right]}
$$

*is the same for all the arms. Still, this corresponds to the Bayesian formulation and we do not go further in this paper.*

*Without any prior information on the arms, we have no reason to choose $B_1 \gg B_2$ e.g., and we will choose the budget shares as close as possible. If $B = n_K K + b$, with $0 \leq b < K$ is the Euclidian division of $B$ by $K$, a possibility is to choose $B_1 = \cdots = B_b = n_K + 1$ and $B_{b+1} = \ldots B_K = n_K$.*

All the policies in EXPLOIT$(B_1, \ldots, B_K)$ have the same probability of ruin, given in the next proposition.

**Proposition 2.** *Given some positive integers $B_1, \ldots, B_K$ such that $B_1 + \cdots + B_K = B$, all the policies in EXPLOIT$(B_1, \ldots, B_K)$ achieve the same probability of ruin, given by*

$$
p^{\mathrm{EX}}(B_1, \ldots, B_K) \triangleq \prod_{k=1}^{K} P_{F_k}(\tau(B_k, k) < \infty) = \exp\left(-\sum_{k=1}^{K} B_k\gamma(F_k)\right).
$$

Importantly, the probability of ruin of the policies in EXPLOIT match the lower bound of Theorem 1, and therefore, all the policies in EXPLOIT are ruin-wise Pareto-optimal. This is a major strength of the EXPLOIT framework: given $B_1, \ldots, B_K$, you can choose any policy in EXPLOIT$(B_1, \ldots, B_K)$ to try to maximize the expected cumulative reward without sacrificing the probability of ruin.

**Remark 5.** *When $B$ is a multiple of $K$, the policy $\pi$ which selects arm $\pi_t \in \arg\max_{k\in[K]}\left\{\sum_{s=1}^{t-1} X_s^{\pi_s}\mathbb{1}_{\pi_s=k}\right\}$ with the highest cumulative reward belongs to EXPLOIT$\left(\frac{B}{K}, \ldots, \frac{B}{K}\right)$. While it is intuitive that such a policy is very conservative, it was a priori not obvious that many policies (all the policies in EXPLOIT$\left(\frac{B}{K}, \ldots, \frac{B}{K}\right)$) achieve the same probability of ruin. Actually, this result becomes false in the general case of rewards in $[-1, 1]$ (not necessarily integers) studied in Section 9.*

**Convention 1.** *From now on, given $B = n_K K + b$ with $0 \leq b < K$, we consider the "more symmetric" case $B_1 = \cdots = B_b = n_K + 1$ and $B_{b+1} = \cdots = B_K = n_K$. We use the shortcut notation EXPLOIT instead of EXPLOIT$(B_1, \ldots, B_K)$ in that specific instance and denote $p^{\mathrm{EX}} \triangleq p^{\mathrm{EX}}(B_1, \ldots, B_K)$.*

## 6.2 Expected Cumulative Reward of Policies in EXPLOIT

By nature, EXPLOIT policies are very conservative. Precisely, they allow a budget of $B_k$ for the exploration of each arm $k \in [K]$. In the previous subsection, we showed that, thanks to this limited exploration, they are ruin-wise Pareto-optimal, i.e., they achieve a small probability of ruin (in the sense of Pareto-optimality). In this subsection, we show that the downside of this limited exploration is that the expected cumulative reward of EXPLOIT policies is fairly low, upper-bounded as shown in the following proposition, whose proof is in Appendix F.

**Proposition 3.** *Assume that the budget $B$ is a multiple of $K$. W.l.o.g., assume that $\mu_1 \geq \cdots \geq \mu_K$. Then, for any policy $\pi$ in EXPLOIT,*

$$
\mathbb{E}\left[\sum_{t=1}^{T} X_t^{\pi_t}\mathbb{1}_{\tau(B,\pi)\geq t-1}\right] \leq \left(1 - p^{\mathrm{EX}}\right)\sum_{k=1}^{K} w_k\mu_k \times T + o(T) \overset{(*)}{\leq} \left(1 - p^{\mathrm{EX}}\right)\max_{k\in[K]}\mu_k \times T + o(T), \tag{11}
$$

---

**Algorithm 1** EXPLOIT-UCB($B$)

---

**for** $t = 1, \ldots, T$ **do**

    Set $\mathcal{A}_t := \left\{ k \in [K] : \sum_{s=1}^{t-1} X_s^{\pi_s} \mathbb{1}_{\pi_s = k} \geq -\frac{B}{K} + 1 \right\}$.

    **if** $\mathcal{A}_t \neq \emptyset$ **then**

        Pull arm $\arg\max_{k \in \mathcal{A}_t} \hat{X}_{t-1}^k + \sqrt{\frac{6 \log(t-1)}{N_k(t-1)}}$.

    **else**

        Pull arm $\arg\max_{k \in [K]} \hat{X}_{t-1}^k$.

---

*where for any* $k \in [K], w_k = \frac{P\left(\tau\left(\frac{B}{K}, k\right) = \infty\right) \prod_{j=1}^{k-1} P\left(\tau\left(\frac{B}{K}, j\right) < \infty\right)}{1 - p^{\text{EX}}}$. *Besides, when two arms have positive and different expectations,* $(*)$ *is a strict inequality.*

As we will see in Section 7, there exists a non-anytime policy whose expected cumulative reward is equal to the right hand-side of (11). This implies that no EXPLOIT policy is regret-wise Pareto-optimal (although they are all ruin-wise Pareto-optimal).

As explained before, a policy in EXPLOIT will allow a budget share $B_k$ for the exploration of each arm $k \in [K]$. Therefore, it may stop pulling arm $k$ after only $B_k$ pulls even without encountering ruin before the horizon $T$. This lack of exploration of arm $k$ penalizes the cumulative reward of arm $k$ and is reflected in the coefficient $(1 - p^{\text{EX}})w_k$ in the middle term of the bound of Proposition 3.

### 6.3 Bandit Algorithms in the EXPLOIT Framework

We know that all the policies in EXPLOIT are ruin-wise Pareto-optimal and achieve the same probability of ruin (this is the result of Proposition 2). In this subsection, we exhibit an EXPLOIT policy which achieves the highest possible cumulative reward within EXPLOIT, given as the middle bound in (11). Though such a policy would not be regret-wise Pareto-optimal as suggested from Proposition 3, it will serve as a basis for the construction of such an optimal policy. For any arm $k \in [K]$ and any round $t$, we introduce $N_k(t) := \sum_{s=1}^{t} \mathbb{1}_{\pi_s = k}$ as the number of pulls of arm $k$ until round $t$, and $\hat{X}_t^k := \frac{1}{N_k(t)} \sum_{s=1}^{t} \mathbb{1}_{\pi_s = k} X_s^{\pi_s}$ as the empirical mean of arm $k$ at round $t$.

We start from the classic bandit algorithm UCB (Upper Confidence Bound, see Auer et al., 2002) and make it "fit" in the EXPLOIT framework. This defines EXPLOIT-UCB($B$) in Algorithm 1 as the policy which, at each round $t \geq 1$, performs UCB among the arms whose cumulative reward is larger than $-\frac{B}{K}$. As EXPLOIT-UCB($B$) is in EXPLOIT, it naturally achieves the optimal bound on the probability of ruin. In addition, it also achieves the middle bound in (11) on the reward, which is asymptotically optimal among EXPLOIT policies, as shown in the next proposition, whose proof is in Appendix G.

**Proposition 4.** *Under the hypotheses of Proposition 3, the expected cumulative reward of EXPLOIT-UCB($B$) satisfies*

$$\mathbb{E}\left[\sum_{t=1}^{T} X_t^{\pi_t} \mathbb{1}_{\tau(B,\pi) \geq t-1}\right] = \left(1 - p^{\text{EX}}\right) \sum_{k=1}^{K} w_k \mu_k \times T + o(T),$$

*where for any* $k \in [K], w_k = \frac{P\left(\tau\left(\frac{B}{K}, k\right) = \infty\right) \prod_{j=1}^{k-1} P\left(\tau\left(\frac{B}{K}, j\right) < \infty\right)}{1 - p^{\text{EX}}}$.

More than the bound itself, the above result states that a standard MAB algorithm "made to fit in EXPLOIT" achieves the best possible reward within EXPLOIT. Here is the intuition behind it. All EXPLOIT policies have the same risk of ruin, and when there is ruin, all policies receive the total reward $-B$. Therefore, a good EXPLOIT policy can only make a difference in the case when there is no ruin. In that case, it should achieve a high cumulative reward, i.e., behave closely to a good standard MAB policy like UCB.

**Remark 6.** *The choice of the constant* 6 *in the square root is different from the original UCB and is only made for simplicity of the proof.*

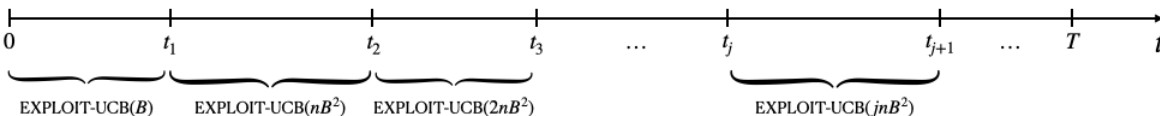

Figure 3: Description of EXPLOIT-UCB-DOUBLE

EXPLOIT-UCB-DOUBLE first behaves like EXPLOIT-UCB($B$), until the cumulative reward reaches $nB^2$. Then, it performs like EXPLOIT-UCB($nB^2$), until the cumulative reward reaches $2nB^2$. Then, it performs like EXPLOIT-UCB($2nB^2$), until the cumulative reward reaches $3nB^2$, and so on.

# 7 A Regret-wise Pareto-optimal Policy

In this section, we make a slight modification on the policy EXPLOIT-UCB($B$) (see Algorithm 1) so that it achieves a large cumulative reward, while keeping its probability of ruin small. The resulting policy is EXPLOIT-UCB-DOUBLE (given in Algorithm 2) and is proven to be regret-wise Pareto-optimal, hence giving an answer to the open problem in Perotto et al. (2019).

## 7.1 A (Regret-wise) Pareto-optimal Policy: EXPLOIT-UCB-DOUBLE

We start from EXPLOIT-UCB($B$) (Algorithm 1). As this policy belongs to EXPLOIT, it is ruin-wise Pareto-optimal. However, as shown in Proposition 3, its cumulative reward is rather low, because its exploration is limited by the budget it allocates to each arm. We tackle this issue by performing a kind of doubling trick (see, e.g., Cesa-Bianchi & Lugosi, 2006) on the budget, which relaunches the exploration when the cumulative reward is large enough.

Let $n \in \mathbb{N}$ be a hyperparameter chosen in advance and for any integer $j \geq 0$, let

$$t_j := \inf \left\{ t \in \{0, \ldots, \min(\tau(B, \pi), T)\} : B + \sum_{s=1}^{t} X_s^{\pi_s} > jnB^2 \right\},$$

with the convention that $t_j = \min(\tau(B, \pi), T) + 1$ if the above set is empty. At each round $t$, EXPLOIT-UCB-DOUBLE (see Figure 3 and Algorithm 2) performs

- EXPLOIT-UCB($B$) pretending that the initial budget is $B$ if $t < t_1$;

- EXPLOIT-UCB($jnB^2$) pretending that the initial budget is $jnB^2$ if $t_j \leq t < t_{j+1}$.

A more visual description of EXPLOIT-UCB-DOUBLE is given in Figure 3 and its pseudo-code is given in Algorithm 2. We denote by $\pi^n$ the policy associated with EXPLOIT-UCB-DOUBLE with input parameter $n$.

The underlying principle of EXPLOIT-UCB-DOUBLE is that, as long as the cumulative reward is low, it performs safely like an EXPLOIT policy in order to minimize the probability of ruin. Then, progressively as the cumulative reward becomes larger, EXPLOIT-UCB-DOUBLE allocates more budget for exploration and behaves more similarly to UCB, so that it starts the cumulative reward maximization.

The next proposition, whose proof is given in Appendix H.1, shows that the probability of ruin of EXPLOIT-UCB-DOUBLE is close to the one of an EXPLOIT policy.

**Proposition 5.** *Given $n \geq 1$, the probability of ruin of EXPLOIT-UCB-DOUBLE is bounded as follows:*

$$P\left(\tau(B, \pi^n) < \infty\right) \leq p^{\text{EX}} + \frac{(p^{\text{EX}})^{nB}}{1 - (p^{\text{EX}})^{nB}} \stackrel{(*)}{=} p^{\text{EX}} + o_{T \to +\infty}(1),$$

*where (\*) holds if $n \xrightarrow{T \to +\infty} +\infty$.*

---

**Algorithm 2** EXPLOIT-UCB-DOUBLE

---

**input:** parameter $n \in \mathbb{N}$; budget $B > 0$.

$j := 0;\ t_0 := 0$.

  **for** $t = 1, \ldots, T$ **do**

  $\quad$ **if** $B + \sum_{s=1}^{t-1} X_s^{\pi_s^n} > (j+1)nB^2$ **then**

  $\quad\quad$ Set $j := j + 1$ and then, set $t_j := t - 1$.

  $\quad$ Set $\mathcal{A}_t := \left\{ k \in [K] : \sum_{s=t_j+1}^{t-1} X_s^{\pi_s^n} \mathbb{1}_{\pi_s^n = k} \geq -\frac{B + \sum_{s=1}^{t_j} X_s^{\pi_s^n}}{K} + 1 \right\}$;

  $\quad$ **for** $k = 1, \ldots, K$ **do**

  $\quad\quad$ Set $N_k(t-1) := \sum_{s=1}^{t-1} \mathbb{1}_{\pi_s^n = k}$ and $\hat{X}_{t-1}^k := \frac{1}{N_k(t-1)} \sum_{s=1}^{t-1} X_s^{\pi_s^n} \mathbb{1}_{\pi_s^n = k}$.

  $\quad$ **if** $\mathcal{A}_t \neq \emptyset$ **then**

  $\quad\quad$ Pull arm $\arg\max_{k \in \mathcal{A}_t} \hat{X}_{t-1}^k + \sqrt{\frac{6 \log(t-1)}{N_k(t-1)}}$.

  $\quad$ **else**

  $\quad\quad$ Pull arm $\arg\max_{k \in [K]} \hat{X}_{t-1}^k$.

---

Proposition 5 shows that the progressively increasing exploration of EXPLOIT-UCB-DOUBLE is reasonable from the perspective of the probability of ruin, because it maintains the probability of ruin almost as small as the one of an EXPLOIT policy, which is ruin-wise Pareto-optimal.

The next proposition shows that this progressively increasing exploration is also reasonable from the perspective of the reward maximization. The next proposition, whose proof can be found in Appendix H.2, shows that, in the case when there is no ruin, the expected cumulative reward of EXPLOIT-UCB-DOUBLE is asymptotically equal to the best possible expected reward.

**Proposition 6.** *Let $n$ be such that $n \xrightarrow{T \to +\infty} +\infty$ and that $n = o(T^{1/4})$. Then, under Assumption 3, the reward given no ruin of EXPLOIT-UCB-DOUBLE with input parameter $n$ is bounded from below by*

$$\mathbb{E}\left[ \sum_{t=1}^{T} X_t^{\pi_t^n} \mathbb{1}_{\tau(B,\pi^n) \geq t-1} \,\middle|\, \tau(B, \pi^n) \geq T \right] \geq \max_{k \in [K]} \mu_k T + o(T). \tag{12}$$

The proof of Proposition 6 also holds in the case $B = T$, that is, when there is no risk of ruin. Therefore, if you apply EXPLOIT-UCB-DOUBLE to a standard MAB (without a risk of ruin), then its expected cumulative reward will be equal to $\max_{k \in [K]} \mu_k T + o(T)$. In the standard MAB terminology, this means that its expected cumulative regret is sublinear: $\max_{k \in [K]} \mu_k T - \text{Rew}_T(\pi^n) = o(T)$.

Now, if we gather the results of Proposition 5 and Proposition 6, the former states that the probability of ruin of EXPLOIT-UCB-DOUBLE is very small (it is almost ruin-wise Pareto-optimal), and when it does not ruin, the latter states that its cumulative reward is asymptotically maximal. Therefore, its cumulative reward is almost asymptotically optimal, and for any value of the input parameter $n$, EXPLOIT-UCB-DOUBLE is "almost" regret-wise Pareto-optimal. This is formalized in the next theorem.

**Theorem 2.** *For any sequence of policies $\pi'$, under the assumptions of Proposition 6, the anytime policy EXPLOIT-UCB-DOUBLE (given in Algorithm 2) with input parameter $n$ satisfies*

$$\sup_{F \in \mathcal{F}_{\{-1,0,1\}}^K} \text{Reg}_F(\pi^n \| \pi') > 0 \implies \inf_F \text{Reg}_F(\pi^n \| \pi') < \frac{(p^{\text{EX}})^{nB}}{1 - (p^{\text{EX}})^{nB}} \max_{k \in [K]} \mu_k.$$

The reason why EXPLOIT-UCB-DOUBLE with arbitrary $n$ is only "almost" regret-wise Pareto-optimal is that, for a very large horizon $T$, the additional exploration of EXPLOIT-UCB-DOUBLE induces an additional risk of ruin of order $\frac{(p^{\text{EX}})^{nB}}{1 - (p^{\text{EX}})^{nB}}$, which is of constant order if $n$ is not allowed to depend on $T$. Yet, if $n$ is allowed to depend on $T$, then we can drop the "almost" in the above theorem.

Following Propositions 5 and 6, any $n = o(T^{1/4})$ such that $n \xrightarrow{T \to +\infty} +\infty$ is a valid choice. Yet, within that range, a larger $n$ will increase $t_1$ and hence, EXPLOIT-UCB-DOUBLE will behave longer like an EXPLOIT policy, undermining its cumulative regret in the long term. As a result, we recommend the subpolynomial (in $T$) $n = \log T$, which also showed a better practical performance than other values of $n$ (see the experiments in Section 8), and we will formulate our final theorems for the specific choice $n = \log T$.

**Theorem 3.** *Under the assumptions of Theorem 2, the (non-anytime) sequence of policies EXPLOIT-UCB-DOUBLE with input parameter $n = \log T$ is regret-wise Pareto-optimal.*

The proof of Theorems 2 and 3 is given in Appendix I.

**Remark 7.** *In the above theorem, $n$ depends on $T$ and therefore, the sequence of policies is no longer anytime, as explained in Section 3. This is the price to pay to achieve the regret-wise Pareto-optimality.*

**Remark 8.** *When there exists a zero arm (see Definition 6), there exists a possibility that EXPLOIT-UCB-DOUBLE continues to pull that arm without increasing the budget or being ruined. We employed Assumption 3 to exclude this case but this assumption can be removed if we use a modified version of EXPLOIT-UCB-DOUBLE such that the exploration is relaunched at round $t = \log T$. However, this modification requires the knowledge of $T$, losing the anytime property and this is beyond the scope of this work.*

Finally, please note that Theorems 2 and 3 provide answers to the open problem by Perotto et al., 2019. Some discussion on the extent of the results described in this subsection is provided in the next subsection.

## 7.2 Discussion

In this subsection, we summarize the theoretical strengths and limitations of EXPLOIT-UCB-DOUBLE.

### 7.2.1 Cumulative Reward given No Ruin

EXPLOIT-UCB is an EXPLOIT policy, and as such, it has the advantage to be ruin-wise Pareto-optimal, but this comes at the cost of a budget-limited exploration. Because of this limited exploration, its cumulative reward is in general smaller than $(1 - p^{\text{EX}}) \max_{k \in [K]} \mu_k T + o(T)$ (Proposition 3), which implies

$$\max_{k \in [K]} \mu_k T - \mathbb{E}\left[ \sum_{t=1}^{T} X_t^{\pi_t} \right] = \Omega(T).$$

Now, assume that you have to deploy a policy $\pi$ for a critical application of the standard MAB (with no risk of ruin). Since the application is critical, you decide to apply a conservative strategy, and for that, you set some arbitrary $B > 0$ and you apply EXPLOIT-UCB (for any $t \geq \tau(B, \pi)$, the choice of arms is arbitrary). Then, the above result shows that the policy $\pi$ will likely have a linear regret, which is not satisfactory.

However, EXPLOIT-UCB-DOUBLE solves that issue by progressively increasing the exploration as its cumulative reward grows (Proposition 6). This is very important because it means that EXPLOIT-UCB-DOUBLE can be reasonably applied to a standard MAB setting and achieve a sublinear regret (in the sense of the standard MAB). This result is valid for any choice of input parameter $n$ for EXPLOIT-UCB-DOUBLE.

### 7.2.2 Probability of Ruin and Regret-wise Pareto Optimality

As a matter of fact, the input parameter $n$ controls the risk of ruin of EXPLOIT-UCB-DOUBLE. On the one hand, if $n$ is chosen constant and independent of $T$, then EXPLOIT-UCB-DOUBLE is an anytime policy, but it is not regret-wise Pareto-optimal. This comes as no surprise, because it cannot compete against all the non-anytime sequences of policies. Yet, it is "almost" regret-wise Pareto-optimal, up to the term $\frac{(p^{\text{EX}})^{nB}}{1-(p^{\text{EX}})^{nB}} \max_{k \in [K]} \mu_k$, which is exponentially small in $n$ and $B$ (Theorem 2). On the other hand, if $n = \log T$ depends on $T$, then EXPLOIT-UCB-DOUBLE is not an anytime sequence of policies anymore, but it becomes regret-wise Pareto-optimal (Theorem 3).

From a practical perspective, how to choose the parameter $n$? Naturally, if the horizon $T$ is known before the experiment, then Theorem 3 recommends the choice $n = \log T$ and provides good theoretical guarantees

for it. Yet, if $T \leq 10^{10}$, then $1 \leq n \leq 23$ and there are not so many candidates for a good choice of $n$. In practice, $n = 1$ is a decent choice and brings a decent reward (which is almost as good as $n = \log T$). Experiments are given in the next subsection.

### 7.2.3 Anytime vs. Regret-wise Pareto Optimality

*Is there an anytime policy which is regret-wise Pareto-optimal?* As explained in Section 4.1, in the case of multinomial arms, unlike in the case of Bernoulli arms, the arm $k^P$ with the largest probability of survival, and the arm $k^E$ with the largest expectation, may be different. In that case, even the best policy which knows the arm distributions is not trivial. Intuitively, such a policy should pull arm $k^P$ until a certain round $\tilde{t}$ to minimize the risk of ruin, and then pull arm $k^E$ until $T$, but $\tilde{t}$ should depend on $T$. It is therefore intuitive that the best such policy is not anytime, and we raise the following open question:

**Conjecture 1.** *No anytime policy is regret-wise Pareto-optimal for all arm distributions in $\mathcal{F}_{\{-1,0,1\}}$.*

Please note that, as explained in Section 4.1, such a matter does not arise in the case of Bernoulli arm distributions, because in that case, $k^E = k^P$. The case of Bernoulli distributions does not capture this subtlety, and this is why we focused on the more difficult case of multinomial distributions in this paper.

### 7.2.4 Assumption 1 and Beyond

In this subsection, we explain why Assumption 1 is made, and we discuss on a potential variant of the S-MAB to the case where it does not hold.

Originally, the standard objective of the MAB is to maximize the expected cumulative reward $\text{Rew}_T$. This also fits into many applications, including those in medicine testing and in finance which have been mentioned in the introduction. In the standard MAB without the risk of ruin, it is known (see, e.g., Auer et al., 2002) that successful strategies $\pi$ achieve

$$\text{Rew}_T(\pi) = \max_{k \in [K]} \mu_k T + o(T).$$

In order to improve upon this bound, the expected cumulative regret $\text{Reg}_T$ was defined as the difference

$$\text{Reg}_T(\pi) = \max_{k \in [K]} \mu_k T - \text{Rew}_T(\pi).$$

The study of $\text{Reg}_T(\pi)$ yields the remaining part of the asymptotic development (in $T$) of $\text{Rew}_T$, and it is known that some successful strategies achieve $\text{Reg}_T(\pi) = O(\log T)$ (see, e.g., Auer et al., 2002).

In contrast, in the S-MAB, Assumption 1 ensures that the ruin may not occur. If we do not assume so, then the ruin occurs almost surely. At the time of ruin, the cumulative reward is equal to $-B$ by definition:

$$\sum_{t=1}^{\tau(B,\pi)} X_t^{\pi_t} = -B$$

and therefore,

$$\text{Rew}_T \triangleq \mathbb{E}\left[\sum_{t=1}^{T \wedge \tau(B,\pi)} X_t^{\pi_t}\right] = P\left(\tau(B,\pi) \leq T\right)(-B) + \left(1 - P\left(\tau(B,\pi) > T\right)\right)\mathbb{E}\left[\sum_{t=1}^{T} X_t^{\pi_t}\Big|\tau(B,\pi) > T\right].$$

When Assumption 1 does not hold, then any strategy $\pi$ satisfies

$$P\left(\tau(B,\pi) \leq T\right) = 1 + o(1),$$

and hence, we can define a notion of regret as

$$-B - \text{Rew}_T = -\left(1 - P\left(\tau(B,\pi) \leq T\right)\right)\left[\mathbb{E}\left[\sum_{t=1}^{T} X_t^{\pi_t}\Big|\tau(B,\pi) > T\right] + B\right].$$

| Experiment No | Budget | Arms | Experiment No | Budget | Arms |
|---|---|---|---|---|---|
| (1) | $B = 9$ | $\{F^{(11)}, F^{(7)}, F^{(12)}\}$ | (4) | $B = 9$ | $\{F^{(6)}, F^{(7)}, F^{(8)}\}$ |
| (2) | $B = 9$ | $\{F^{(1)}, F^{(2)}, F^{(3)}\}$ | (5) | $B = 30$ | $\{F^{(1)}, F^{(2)}, F^{(3)}\}$ |
| (3) | $B = 9$ | $\{F^{(4)}, F^{(5)}, F^{(3)}\}$ | (6) | $B = 30$ | $\{F^{(8)}, F^{(9)}, F^{(10)}\}$ |

Table 1: Settings Considered

Naturally, minimizing the regret as defined above yields a bound which is of order smaller than constant. From an application perspective, this is hard to justify for many practitioners: for example, in the financial portfolio example mentioned in the introduction, this corresponds to trying to increase the gain by an amount of money negligible compared to the initial budget $B$ and the horizon $T$. Yet, from a theoretical perspective, the above problem is very interesting, and will be the object of future work.

## 8 Experiments

In this section, we evaluate the empirical performance of the algorithms EXPLOIT-UCB and EXPLOIT-UCB-DOUBLE (for various parameters $n$) that we introduced in this paper.

### 8.1 Setting

**Algorithms implemented:** we have compared the performance of our algorithms EXPLOIT-UCB and EXPLOIT-UCB-DOUBLE (with parameters $n \in \{1, \lceil \log T \rceil, 100\}$, where $\lceil \log T \rceil = 10$) to the classic bandit algorithms UCB (Auer et al., 2002) and Multinomial Thompson Sampling (MTS), which is a generalization of Thompson sampling to multinomial rewards (Riou & Honda, 2020), and to the other existing S-MAB algorithms Gambler-UCB1 (Perotto et al., 2022) and GWA-UCB1 (Manome et al., 2023).

**Important note:** let us mention here that both Gambler-UCB1 and GWA-UCB1 are purely empirical, and have no theoretical guarantee. Moreover, they have only been tested for rewards in $\{-1, 0, 1\}$. Besides, we mention that there is a mistake in the definition of GWA-UCB1, making it not applicable to settings with (possibly) negative rewards, so we slightly modified the formula to shift the rewards from $\{-1, 1\}$ to $\{0, 2\}$.

**Setting:** for all the experiments performed, we consider a bandit setting with horizon $T = 20000$ with $K = 3$ multinomial arms of common support $\{-1, 0, 1\}$ and distributions $F^{(i_1)}, F^{(i_2)}$ and $F^{(i_3)}$, where $i_1, i_2, i_3 \in \{1, \ldots, 12\}$. The distributions $F^{(i)}$ for $i \in \{1, \ldots, 12\}$ are described below:

$$F^{(1)} = \text{Mult}(0.4, 0.12, 0.48); \quad F^{(2)} = \text{Mult}(0.04, 0.88, 0.08); \quad F^{(3)} = \text{Mult}(0.5, 0.1, 0.4);$$
$$F^{(4)} = \text{Mult}(0.48, 0, 0.52); \quad F^{(5)} = \text{Mult}(0.04, 0.91, 0.05); \quad F^{(6)} = \text{Mult}(0.45, 0, 0.55);$$
$$F^{(7)} = \text{Mult}(0.05, 0.85, 0.1); \quad F^{(8)} = \text{Mult}(0.5, 0, 0.5); \quad F^{(9)} = \text{Mult}(0.495, 0, 0.505);$$
$$F^{(10)} = \text{Mult}(0.049, 0.9, 0.051); \quad F^{(11)} = \text{Mult}(0.4, 0.1, 0.5); \quad F^{(12)} = \text{Mult}(0.6, 0, 0.4).$$

We further describe all the settings in Table 1. Please note that they have been chosen to emphasize on the technicalities that are specific to the S-MAB, and which do not appear in the standard MAB. For that purpose, we have chosen a low budget $B = 9$ in settings (1)–(4), because the case of large budget resembles to the standard MAB. In many experiments, e.g. in experiment (1), the parameters of the arms have been chosen such that the arm $k$ with the largest expectation $\mu_k$ and the arm $k'$ with the lowest probability of ruin, i.e. the largest $\gamma(F_{k'})$, are different. Furthermore, in many settings, the arm expectations are very close such that a good policy has to balance well the exploration in order to identify the arm with the largest expectation without causing the ruin.

**Evaluation metric:** in order to evaluate the practical performance of the algorithms introduced in this paper, we define the survival regret as

$$\text{S-Reg}_T(\pi) = \left( 1 - \exp\left( -\frac{B}{K} \sum_{k=1}^{K} \gamma(F_k) \right) \right) \max_{k \in [K]} \mu_k T - \text{Rew}_T(\pi),$$

which is a hypothetical regret with respect to a policy that achieves the Pareto-optimal ruin probability and always pulls the arm with the highest expected reward given no ruin.

## 8.2 Results

The survival regret of the algorithms considered is given in Figure 4. All the curves are averages over 200 simulations. The corresponding average survival time $\min(T, \tau(B, \pi))$ of the algorithms is given in Table 2. The average proportion of ruins of the algorithms is given in Table 3.

**EXPLOIT-UCB-DOUBLE performs the best.** Over the experiments we conducted, it is quite clear that EXPLOIT-UCB-DOUBLE largely outperforms all the other tested policies in most settings. In the few cases where they do not (precisely when the budget is large, and hence, the S-MAB resembles a standard bandit problem), its regret is low and comparable to UCB.

**MTS has a large regret.** In contrast to the standard bandit problems, where MTS is optimal both theoretically and in practice (Riou & Honda, 2020), for the S-MAB, the regret of MTS is rather large, even larger than the one of UCB, as it was also noted in Perotto et al. (2022) and Manome et al. (2023). This lack of performance of MTS is due to frequent ruins: in experiments (1)–(4) where the budget is low ($B = 9$), the proportion of ruins of MTS is 2 to 3 times as large as the one from any other algorithm considered. The explanation behind this phenomenon is that MTS is a Bayesian algorithm which has a randomized exploration component. While this randomization contributes to its success in the standard bandit problem, in the S-MAB, it is the source of frequent ruins and hence, drags its performance down considerably.

**Gambler-UCB1 is consistently the worst.** In almost all the experiments conducted, Gambler-UCB1 has the largest regret by far. Interestingly, its proportion of ruins is generally not so high. We recall that Gambler-UCB1 replaces $\log t$ by $\log B_t$ in the UCB selection policy. This clearly decreases the exploration at every time step, instead of increasing it to achieve the standard exploration-exploitation dilemma, undermining its performance. This is in line with the results of Perotto et al. (2022), which found out empirically Gambler-UCB1 performs less well than UCB.

**UCB performs quite well.** In many of the experiments above, UCB performs remarkably well, with a regret which comes close to the one of EXPLOIT-UCB-DOUBLE. This is not so surprising, because EXPLOIT-UCB-DOUBLE has a hyperparameter $n$, which controls its level of exploration: when $n$ is large, it exploits more and behaves almost like an EXPLOIT policy; when $n$ is low, it explores more, and behaves almost like UCB. In particular, when the budget is large like in experiments (5) and (6), the exploration does not need to be constrained so much, and UCB performs similarly to EXPLOIT-UCB-DOUBLE.

**GWA-UCB1 is inconsistent.** The regret of GWA-UCB1 is often very large (experiments (1), (2), (4) and (5)), and occasionally very low (experiments (3) and (6)). We recall that GWA-UCB1 has been defined as a policy in a parametric class of policies containing UCB, and that the parameters have been tuned empirically in Manome et al. (2023). The inconsistency of GWA-UCB1 is also not a surprising fact, because the regret bound in the S-MAB is in the sense of Pareto, suggesting that an algorithm performing too well in some setting would perform poorly in some other. Such a phenomenon is especially expected when hyperparameters are tuned empirically by hand, potentially creating the observed inconsistency.

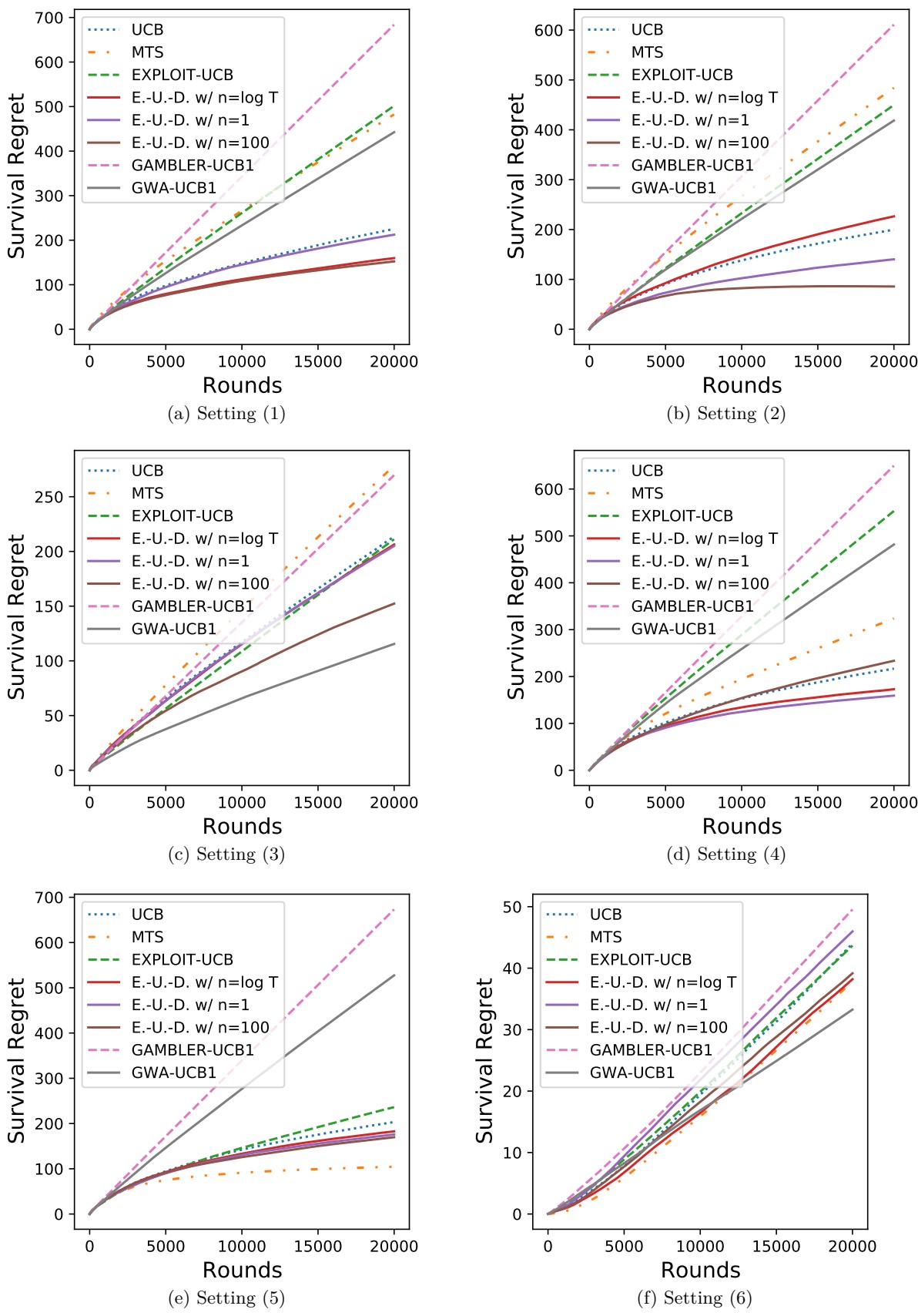

Figure 4: Survival Regret of EXPLOIT-UCB-DOUBLE, EXPLOIT-UCB, UCB, MTS, Gambler-UCB1 and GWA-UCB1, in the settings (1)–(6).

| Setting | UCB | MTS | EX-UCB | EX-D($\log T$) | EX-D(1) | EX-D(100) | Gambling | GWA-UCB1 |
|---------|-----|-----|--------|---------------|---------|-----------|----------|----------|
| (1) | 17917 | 14541 | 18806 | 18309 | 18508 | 18506 | 19303 | 19503 |
| (2) | 18215 | 13352 | 18904 | 17911 | 18909 | 19603 | 18211 | 19010 |
| (3) | 8244 | 5790 | 12862 | 8858 | 9156 | 10663 | 15059 | 12989 |
| (4) | 18512 | 16326 | 18709 | 19107 | 18805 | 18015 | 18755 | 18711 |
| (5) | 19903 | 20000 | 20000 | 20000 | 20000 | 20000 | 19215 | 20000 |
| (6) | 10290 | 9644 | 12735 | 11751 | 10621 | 12721 | 15053 | 14980 |

Table 2: Average Survival Time.

| Setting | UCB | MTS | EX-UCB | EX-D($\log T$) | EX-D(1) | EX-D(100) | Gambling | GWA-UCB1 |
|---------|-----|-----|--------|---------------|---------|-----------|----------|----------|
| (1) | 0.10 | 0.27 | 0.06 | 0.08 | 0.07 | 0.07 | 0.04 | 0.03 |
| (2) | 0.09 | 0.33 | 0.05 | 0.10 | 0.05 | 0.02 | 0.09 | 0.05 |
| (3) | 0.60 | 0.72 | 0.36 | 0.57 | 0.56 | 0.48 | 0.26 | 0.36 |
| (4) | 0.07 | 0.18 | 0.06 | 0.04 | 0.06 | 0.10 | 0.07 | 0.07 |
| (5) | 0.01 | 0 | 0 | 0 | 0 | 0 | 0.04 | 0 |
| (6) | 0.64 | 0.64 | 0.46 | 0.55 | 0.60 | 0.49 | 0.35 | 0.36 |

Table 3: Proportion of Ruin of the Algorithms.

**The hyperparameter in EXPLOIT-UCB-DOUBLE does not change much in practice.** While fixed values of $n$ do not offer the theoretical guarantees of Theorem 3, in practice, the value of $n$ does not have much impact on the practical performance of EXPLOIT-UCB-DOUBLE. Actually, a good value for $n$ being of the order of $\log T$, choosing a small $1 \leq n \leq 10$ is likely to give a good performance for relatively small horizon $T \leq 10^{10}$. In contrast, for very large horizons (which may happen in some of the financial applications of the S-MAB mentioned in the introduction), e.g., $T \gg 10^{100}$, we expect the parameter $n$ to have a more determining impact on the regret.

**The case of large budget.** In the case of a large initial budget (experiments (5) and (6)), all the policies with strong theoretical guarantees (UCB, MTS, EXPLOIT-UCB-DOUBLE and EXPLOIT-UCB-DOUBLE) perform rather well. In experiment (5), GWA-UCB1 and Gambler-UCB1 seem to have a linear regret, while all the other policies have a much lower regret. Among them, MTS clearly has the lowest regret. This was predictable: when the budget is large, the problem resembles to a standard bandit problem, for which MTS is known to be theoretically and practically optimal. Besides, the regret, as well as the proportion of ruins and the average time of ruin of UCB, EXPLOIT-UCB ad EXPLOIT-UCB-DOUBLE are very similar. This comes as no surprise, because when the budget is large, then the constraints of the EXPLOIT framework is loose, and all of those algorithms behave like UCB most of the time. In experiment (6), the setting is designed such that the arms are extremely hard to distinguish: the arm expectations are very similar and close to 0. In that case, even the large initial budget does not preclude ruin, and all the algorithms suffer from a large regret.

## 9 Extension to the Case of Non-integer Rewards

This section is structured in two subsections:

1. first, we generalize our results on the probability of ruin to the case of rewards in $[-1, 1]$;

2. secondly, we generalize our policies to the case of rewards in $[-1, 1]$ and explain the challenges in deriving regret-wise and ruin-wise Pareto-optimality-type theoretical guarantees.

### 9.1 Generalized Results on the Probability of Ruin

Let $\mathcal{F}_{[-1,1]}$ be the set of distributions bounded in $[-1, 1]$ which are not positive or zero (see Definition 6). Theorem 1 can be generalized as follows.

**Proposition 7.** *Let* $(\alpha_k)_{k \in [K]}$ *be as in Theorem 1. For any policy* $\pi$,

$$\inf_{F \in \mathcal{F}_{[-1,1]}^K} \left\{ P_{(F_1,\ldots,F_K)}(\tau(B, \pi) < \infty) - \exp\left(-(B+1)\sum_{k=1}^K \alpha_k \gamma(F_k)\right) \right\} < 0$$

$$\implies \sup_{F \in \mathcal{F}_{[-1,1]}^K} \left\{ P_{(F_1,\ldots,F_K)}(\tau(B, \pi) < \infty) - \exp\left(-(B+1)\sum_{k=1}^K \alpha_k \gamma(F_k)\right) \right\} > 0. \quad (13)$$

The proof of Proposition 7 follows the proof of Theorem 1, except for the subadditivity argument, and it is given in Appendix D, next to the proof of Theorem 1.

The above bound is most likely not tight, because the subadditivity argument that we used is not tight. Actually, this is the main technicality which separates the general case from the multinomial case. In the multinomial case, it is known that at the time of ruin, the cumulative reward is exactly $-\lceil B \rceil$. In the general case, however, the cumulative reward at the time of ruin is stochastic and depends on the arm distributions as well as the policy $\pi$.

Similarly to the case of rewards in $\{-1, 0, 1\}$, we state a lemma providing an insightful interpretation of the bound (13). While in the case of rewards in $\{-1, 0, 1\}$, Lemma 1 gave an easy expression of $\gamma(F_k)$, in the general case, it is expressed as a limit.

**Lemma 3.** *For any* $k \in [K]$,

$$\forall F_k \in \mathcal{F}_{[-1,1]}, \quad \frac{1}{B} \log P_{F_k}(\tau(B, k) < \infty) \leq \liminf_{B' \to \infty} \frac{1}{B'} \log P_{F_k}(\tau(B', k) < \infty) = -\gamma(F_k).$$

This lemma means that $\gamma(F_k)$ can be related to the probability of ruin $P_{F_k}(\tau(B, k) < \infty)$ of the stochastic process with increments i.i.d. from $F_k$.

### 9.2 Generalized Results on the Regret

We first generalize the EXPLOIT framework to rewards in $[-1, 1]$, and then give the results on EXPLOIT-UCB-DOUBLE based on the new EXPLOIT framework.

#### 9.2.1 Generalization of the EXPLOIT Framework

For the sake of clarity, we generalize the framework $\text{EXPLOIT}(B_1, \ldots, B_K)$ only in the case $B_1 = \cdots = B_K = \frac{B}{K}$ (but we could similarly generalize it to any $(B_1, \ldots, B_K)$), and we will refer to this generalization as EXPLOIT. Let $(Y_k^s)_{s \geq 1}$ be the rewards from arm $k \in [K]$. The main problem in the general case is that the sum of rewards is not necessarily an integer, and hence if you pull arm $k$ until the first round $t$ such that $\sum_{s=1}^t Y_k^s \leq -\frac{B}{K}$, then we have

$$\sum_{s=1}^t Y_k^s = -\frac{B}{K} - \kappa, \quad \text{with} \ \ \kappa \in [0, 1).$$

If $\kappa$ is large, then this will reduce the budget of the other arms. To remedy that issue, we conservatively decide to stop the exploration when $\sum_{s=1}^t Y_k^s < -\frac{B}{K} + 1$, so that each arm has a budget share between $\frac{B}{K} - 1$ and $\frac{B}{K}$. This is formalized as follows.

**Definition 9.** *For any* $k \in [K]$, *denoting by* $(Y_k^s)_{s \geq 1}$ *the rewards from arm* $k$, *let* $\tau_k^< := \inf \{ t \geq 1 : \sum_{s=1}^t Y_k^s < -\frac{B}{K} + 1 \}$. *We say that a policy* $\pi$ *belongs to the EXPLOIT framework if:*

- *for any* $t < \sum_{k=1}^{K} \tau_k^<$, $\pi_t \in \left\{ k \in [K] : \sum_{s=1}^{t} \mathbb{1}_{\pi_s = k} < \tau_k^< \right\}$, *and*

- *for any* $t \geq \sum_{k=1}^{K} \tau_k^<$, $\pi_t = \arg\max \left\{ \sum_{s=1}^{t} X_s^{\pi_s} \mathbb{1}_{\pi_s = k} \right\}$ *(in case of tie, $\pi_t$ is the smallest arm index).*

We decompose $B = n_K K + b$ for an integer $n_K$ and $0 \leq b < K$, and let $\alpha_k(B) \triangleq \frac{n_K + \mathbb{1}_{k+1 \leq b}}{B}$ for any $k \in [K]$, which we denote by $\alpha_k$ in the absence of ambiguity on the initial budget $B$. Similarly to the multinomial case, all the distributions in EXPLOIT have the same probability of ruin $p^{\mathrm{EX}}\left(\frac{B}{K}, \ldots, \frac{B}{K}\right)$. However, in the general case, this value is not deterministic. Yet, it can be bounded easily following the definition of EXPLOIT.

**Proposition 8.** *It holds that*

$$\exp\left(-B \sum_{k=1}^{K} \alpha_k(B) \gamma(F_k)\right) \leq p^{\mathrm{EX}}\left(\frac{B}{K}, \ldots, \frac{B}{K}\right) \leq \exp\left(-B \sum_{k=1}^{K} \left(\alpha_k(B) - \frac{1}{B}\right) \gamma(F_k)\right).$$

The RHS of Proposition 8 does not match the lower bound of Proposition 7. However, if we apply it to the initial budget $B' = B + K + 1$, it yields

$$p^{\mathrm{EX}}\left(\frac{B'}{K}, \ldots, \frac{B'}{K}\right) \leq \exp\left(-(B+1) \sum_{k=1}^{K} \alpha_k(B+1) \gamma(F_k)\right),$$

which coincides with the lower bound in Proposition 7. Therefore, the looseness of the ruin probability in the general case (w.r.t. the bound of Proposition 7) corresponds to a budget loss of at most $K + 1$.

### 9.2.2 Generalization of EXPLOIT-UCB-DOUBLE

We define EXPLOIT-UCB and EXPLOIT-UCB-DOUBLE based on the newly defined EXPLOIT framework for rewards in $[-1, 1]$. The pseudo-code for EXPLOIT-UCB-DOUBLE remains the same as in the multinomial case and is given in Algorithm 2. Interestingly, most of the results can be extended from the multinomial case to the general case.

**Proposition 9.** *Let $\pi^n$ be the policy associated to the policy EXPLOIT-UCB-DOUBLE with input parameter $n$. Its probability of ruin is upper bounded by*

$$P(\tau(B, \pi^n) < \infty) \leq p^{\mathrm{EX}} + \frac{(p^{\mathrm{EX}})^{nB}}{1 - (p^{\mathrm{EX}})^{nB}}.$$

*The cumulative reward given no ruin of EXPLOIT-UCB-DOUBLE is bounded from below by*

$$\mathbb{E}\left[\sum_{t=1}^{T} X_t^{\pi_t^n} \mathbb{1}_{\tau(B, \pi^n) \geq t-1} \,\middle|\, \tau(B, \pi^n) \geq T\right] \geq \max_{k \in [K]} \mu_k T + o(T).$$

*As a consequence of the above results, it holds that, for any sequence of policies $\pi'$,*

$$\sup_{F \in \mathcal{F}_{\{-1,0,1\}}^K} \mathrm{Reg}_F(\pi^n \| \pi') > 0 \implies \inf_F \mathrm{Reg}_F(\pi^n \| \pi') < \frac{(p^{\mathrm{EX}})^{nB}}{1 - (p^{\mathrm{EX}})^{nB}} \max_{k \in [K]} \mu_k.$$

As explained before, the main difficulty is that in general, for any arm $k$, $\sum_{s=1}^{\tau_k^<} Y_k^s$ is not deterministic (using the notations of Definition 8). As a result, even for EXPLOIT policies, the probability of ruin cannot be decomposed as a product of independent ruin probabilities of the arms. The same reason leads to the looseness in the subadditivity bound (see Lemma 7), leading to the $(B+1)$ factor in the bound of Proposition 7 instead of $B$. For that reason, we believe that the bound of Proposition 7 is not tight, and the question of the regret-wise Pareto-optimality of EXPLOIT-UCB-DOUBLE in the general case is open.

As a final point of consideration, please note that in the general case as well, the reward given no ruin of EXPLOIT-UCB-DOUBLE (see Proposition 6) is asymptotically optimal and equal to $\max_{k \in [K]} \mu_k T$, which makes it worth applying to more standard bandit settings where an algorithm with a stronger exploitation component is desired.

## 10 Conclusion

In this paper, we introduced the S-MAB, an extension of the MAB with a risk of ruin, which naturally follows from many practical applications but is considerably more difficult to study. For example, contrary to the MAB, no policy can achieve a sublinear regret in the standard sense, because every single pull increases considerably the probability of ruin. Our contributions are threefold:

- we formally defined the problem, defined the objective to achieve with the regret-wise Pareto-optimality and introduced the key notion to our problem with the time of ruin and the probability of ruin. Furthermore, we explained how an optimal policy needs to minimize the probability of ruin while at the same time maximize the cumulative reward given no ruin, which are two concepts in apparent opposition;

- we studied the probability of ruin, on which we provided both a lower bound (Theorem 1) and policies achieving this lower bound (EXPLOIT policies);

- using a doubling trick over an EXPLOIT policy, we derived a policy which is almost regret-wise Pareto-optimal, and can be made exactly Pareto-optimal if the policy knows the horizon before starting the procedure. This provides an answer to an open problem from COLT 2019 (see Perotto et al., 2019).

Along the way, we raised several open questions which we keep for future work: in the case of integer rewards, is there a policy which is regret-wise Pareto-optimal and does not depend on the horizon? In the general case of bounded rewards, most of our results extend, except that the lower bound on the probability of ruin is seemingly not tight and we did not prove that EXPLOIT-UCB-DOUBLE is regret-wise Pareto-optimal with $n = \log T$. Can we improve upon those?

This is a fairly new and yet unexplored problem, but we believe that it is very rich and paves the way to a myriad of new questions.

### Acknowledgments

JH was supported by JSPS, KAKENHI Grant Number JP21K11747, Japan. MS was supported by Institute for AI and Beyond, UTokyo. The authors would like to thank anonymous reviewers for their insightful reviews.

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

## A  General Classic Results

### A.1  A Classic Lemma in the Theory of Stochastic Processes

The following result is classic in the theory of stochastic processes and we state it without a proof. It will be used throughout the paper.

**Lemma 4.** *Consider the setting of $K \geq 1$ arms of multinomial distributions $F_1, \ldots, F_K$ (of common support $\{-1, 0, 1\}$). Let $B > 0$ be a positive integer. Then,*

$$P(\tau(B, k) < \infty) = \min\left(1, \left(\frac{P_{X \sim F_k}(X = -1)}{P_{X \sim F_k}(X = 1)}\right)^B\right).$$

## B  Proof of the Linearity of the Classic Regret (Proposition 1)

In this appendix, we prove a slightly stronger version of Proposition 1, namely that Proposition 1 even holds in the case where the supremum on $F$ is taken on Bernoulli arm distributions of support $\{-1, 1\}$. Let $\mathcal{F}^K_{\{-1,1\}}$ be the set of $K$-tuples of Bernoulli arm distributions of support $\{-1, 1\}$.

**Proposition 10.** *Assume that the initial budget $B > 2$. For any policy $\pi$, it holds that*

$$\sup_{F \in \mathcal{F}^K_{\{-1,1\}}} \sup_{\tilde{\pi}} \text{Reg}_F(\pi \| \tilde{\pi}) > 0.$$

*Proof.* Let $\pi = (\pi^T)_{T \geq 1}$ be a sequence of policies, and let $k_0 \in [K]$ such that there are infinitely many $T \geq 1$ such that

$$P(\pi_1^T = k_0) \leq \frac{1}{K}. \tag{14}$$

We denote by $S$ the set of all such $T \geq 1$:

$$S := \left\{T \geq 1 : P(\pi_1^T = k_0) \leq \frac{1}{K}\right\},$$

and we note that $|S| = \infty$. We then let $F_1, \ldots, F_K$ be the Bernoulli arm distributions, with respective parameters $p_1, \ldots, p_K$ such that $k_0$ is the only optimal arm:

$$p_{k_0} > \max_{k \neq k_0} p_k.$$

W.l.o.g., we can assume that $k_0 = 1$. Recall that, by definition, for any $k \in [K]$,

$$p_k = P_{X \sim F_k}(X = +1) = 1 - P_{X \sim F_k}(X = -1).$$

Let $\Delta := \frac{1 - p_1}{p_1}$. Denoting by $\text{Rew}_T(1)$ the reward of the (optimal) policy $\pi_t = 1$ for any $t \geq 1$, it holds that

$$\text{Rew}_T(1) - \text{Rew}_T(\pi^T) = \mathbb{E}\left[\sum_{t=1}^{T} X_t^1 \mathbb{1}_{\tau(B,1) \geq t-1}\right] - \mathbb{E}\left[\sum_{t=1}^{T} X_t^{\pi_t} \mathbb{1}_{\tau(B,\pi) \geq t-1}\right].$$

Let us then compute the cumulative expected reward of the policy pulling only arm 1:

$$\mathbb{E}\left[\sum_{t=1}^{T} X_t^1 \mathbb{1}_{\tau(B,1) \geq t-1}\right] = \mu_1 \sum_{t=1}^{T} P\left(\tau(B, 1) \geq t - 1\right)$$
$$= \mu_1 P\left(\tau(B, 1) \geq T\right) T + o(T)$$
$$= \mu_1 P\left(\tau(B, 1) = \infty\right) T + o(T).$$

Then, using Lemma 4, we deduce

$$\mathbb{E}\left[\sum_{t=1}^{T} X_t^1 \mathbb{1}_{\tau(B,1)\geq t-1}\right] = \mu_1 \left(1 - \Delta^{\lceil B \rceil}\right) T + o(T)$$

$$= (2p_1 - 1)\left(1 - \Delta^{\lceil B \rceil}\right) T + o(T). \qquad (15)$$

The cumulative reward of $\pi^T$ is upper-bounded by

$$\mathbb{E}\left[\sum_{t=1}^{T} X_t^{\pi_t^T} \mathbb{1}_{\tau(B,\pi^T)\geq t-1}\right] = \mathbb{E}\left[X_1^{\pi_1^T} + \sum_{t=2}^{T} X_t^{\pi_t^T} \mathbb{1}_{\tau(B,\pi^T)\geq t-1}\right]$$

$$\leq \mathbb{E}\left[X_1^{\pi_1^T} + \sum_{t=2}^{T} X_t^1 \mathbb{1}_{\tau(B,\pi^T)\geq t-1}\right]$$

$$= \mathbb{E}\left[X_1^{\pi_1^T}\right] + \mathbb{E}\left[\sum_{t=2}^{T} X_t^1 \mathbb{1}_{\forall s\leq t-1, B+\sum_{r=1}^{s} X_r^{\pi_r^T} > 0}\right]$$

$$= \mathbb{E}\left[X_1^{\pi_1^T}\right] + P\left(X_1^{\pi_1^T} = 1\right) \mathbb{E}\left[\sum_{t=2}^{T} X_t^1 \mathbb{1}_{\forall s\leq t-1, B+\sum_{r=1}^{s} X_r^{\pi_r^T} > 0}\Big| X_1^{\pi_1^T} = 1\right]$$

$$+ \left(1 - P\left(X_1^{\pi_1^T} = 1\right)\right) \mathbb{E}\left[\sum_{t=2}^{T} X_t^1 \mathbb{1}_{\forall s\leq t-1, B+\sum_{r=1}^{s} X_r^{\pi_r^T} > 0}\Big| X_1^{\pi_1^T} = -1\right]$$

$$= \mathbb{E}\left[X_1^{\pi_1^T}\right] + P\left(X_1^{\pi_1^T} = 1\right) \mathbb{E}\left[\sum_{t=2}^{T} X_t^1 \mathbb{1}_{\forall s\leq t-1, B+1+\sum_{r=2}^{s} X_r^1 > 0}\right]$$

$$+ \left(1 - P\left(X_1^{\pi_1^T} = 1\right)\right) \mathbb{E}\left[\sum_{t=2}^{T} X_t^1 \mathbb{1}_{\forall s\leq t-1, B-1+\sum_{r=2}^{s} X_r^1 > 0}\right].$$

But then, using (15), we know that

$$\mathbb{E}\left[\sum_{t=2}^{T} X_t^1 \mathbb{1}_{\forall s\leq t-1, B+1+\sum_{r=2}^{s} X_r^1 > 0}\right] = (2p_1 - 1)\left(1 - \Delta^{\lceil B \rceil+1}\right) T + o(T)$$

and

$$\mathbb{E}\left[\sum_{t=2}^{T} X_t^1 \mathbb{1}_{\forall s\leq t-1, B-1+\sum_{r=2}^{s} X_r^1 > 0}\right] = (2p_1 - 1)\left(1 - \Delta^{\lceil B \rceil-1}\right) T + o(T).$$

Therefore, we can deduce that the agent's total cumulative reward is upper-bounded by

$$\mathbb{E}\left[\sum_{t=1}^{T} X_t^{\pi_t} \mathbb{1}_{\forall s\leq t-1, B+\sum_{r=1}^{s} X_r^{\pi_r^T} > 0}\right] \leq (2p_1 - 1)T \times$$

$$\left(P\left(X_1^{\pi_1^T} = 1\right)\left(1 - \Delta^{\lceil B \rceil+1}\right) + \left(1 - P\left(X_1^{\pi_1^T} = 1\right)\right)\left(1 - \Delta^{\lceil B \rceil-1}\right)\right) + o(T).$$

We then decompose

$$P\left(X_1^{\pi_1^T} = 1\right) = \sum_{k=1}^{K} P\left(\pi_1^T = k\right) P\left(X_1^{\pi_1^T} = 1\right)$$

$$= \sum_{k=1}^{K} p_k P\left(\pi_1^T = k\right),$$

and by (14), if $T \in S$, then

$$P\left(X_1^{\pi_1^T} = 1\right) \le \frac{1}{K}p_1 + \left(1 - \frac{1}{K}\right)\max_{k \ne 1} p_k < p_1.$$

Now, please note that the function

$$p \mapsto p\left(1 - \Delta^{\lceil B \rceil + 1}\right) + (1 - p)\left(1 - \Delta^{\lceil B \rceil - 1}\right)$$

is strictly increasing on $[0, p_1]$ and equal to $1 - \Delta^{\lceil B \rceil}$ at $p = p_1$. As a result, there exists $\delta > 0$ such that, for any $T \in S$,

$$\mathbb{E}\left[\sum_{t=1}^{T} X_t^{\pi_t^T} \mathbb{1}_{\forall s \le t-1, B + \sum_{r=1}^{s} X_r^{\pi_r^T} > 0}\right] \le (2p_1 - 1)T\left(1 - \Delta^{\lceil B \rceil} - \delta\right) + o(T)$$

$$= \mathbb{E}\left[\sum_{t=1}^{T} X_t^1 \mathbb{1}_{\tau(B,1) \ge t-1}\right] - (2p_1 - 1)\delta T + o(T).$$

By definition of the regret, this implies

$$\sup_{\tilde{\pi}} \operatorname{Reg}_F(\pi \| \tilde{\pi}) \ge (2p_1 - 1)\delta,$$

concluding the proof of the proposition. $\qquad\square$

**Remark 9.** *In the proof of this proposition we considered an instance such that $p_1 < 1$ so that $\Delta > 0$. In the case $p_1 = 1$ or more generally when there exists a positive arm, it is possible to achieve zero regret if the initial budget is $B > K - 1$ with, for instance, the policy $\pi$ which performs the classic bandit algorithm UCB (see, e.g., Bubeck & Cesa-Bianchi, 2012) when $B_{t-1} > K - 1$, and pulls one of the arms which has not yielded a negative reward otherwise.*

## C  Properties of KL Divergence

In this appendix we prepare lemmas on the KL divergence used for the analysis of the ruin probability.

**Lemma 5.** *Let $P \in \mathcal{F}_{[-1,1]}$ be a distribution over $[-1,1]$ with a positive expectation and let $\Lambda(\lambda) := \log \mathbb{E}\left[e^{\lambda X}\right]$ be its logarithmic moment-generating function. Then, there exists $\lambda' < 0$ such that $\Lambda(\lambda') = 0$ and it satisfies*

$$\inf_{Q: E_{X \sim Q}[X] < 0} \frac{\mathrm{KL}(Q \| P)}{\mathbb{E}_{X \sim Q}[-X]} = -\lambda'.$$

*Proof.* First we have

$$\lim_{\lambda \to -\infty} \Lambda(\lambda) = \infty$$

since $P[X < 0] > 0$ because the distribution $P$ is not positive or zero by definition of $\mathcal{F}_{[-1,1]}^K$. Therefore, by the continuity of $\Lambda(\lambda)$ there exists $\lambda' < 0$ such that $\Lambda(\lambda') = 0$ since $\Lambda(0) = 0$ and $\Lambda'(0) = \mathbb{E}_{X \sim P}[X] > 0$.

Now, let

$$\Lambda^*(x) := \sup_{\lambda} \{\lambda x - \Lambda(\lambda)\}$$

be the Fenchel-Legendre transform of $\Lambda(\lambda)$ and define $x' = \Lambda'(\lambda')$. Then we have $\Lambda^*(x') = \lambda' x' - \Lambda(\lambda') = \lambda' x'$. Therefore,

$$\inf_{x < 0} \frac{\Lambda^*(x)}{-x} \le \frac{\Lambda^*(x')}{-x'}$$

$$= \frac{\lambda' x'}{-x'}$$

$$= -\lambda'.$$

On the other hand,

$$
\begin{aligned}
\inf_{x<0} \frac{\Lambda^*(x)}{-x} &= \inf_{x<0} \sup_{\lambda} \frac{\lambda x - \Lambda(\lambda)}{-x} \\
&\geq \inf_{x<0} \frac{\lambda' x - \Lambda(\lambda')}{-x} \\
&= \inf_{x<0} \frac{\lambda' x}{-x} \\
&= -\lambda'.
\end{aligned}
$$

Therefore we see that

$$
\inf_{x<0} \frac{\Lambda^*(x)}{-x} = -\lambda'.
$$

It is well-known as the relation between Cramer's theorem and Sanov's theorem (see, e.g., Dembo & Zeitouni, 2009) that for $x < \mathbb{E}_{X \sim P}[X]$,

$$
\Lambda^*(x) = \inf_{Q:\mathbb{E}_{X \sim Q}[X] \leq x} \mathrm{KL}(Q\|P),
$$

which concludes the proof of the lemma. $\qquad\square$

**Lemma 6.** *Let $Q$ be an arbitrary distribution such that $\mathbb{E}_{X \sim Q}[X] < 0$ and fix $\epsilon > 0$. Then, there exists $P$ such that $\mathbb{E}_{X \sim P}[X] > 0$ and*

$$
\frac{\mathrm{KL}(Q\|P)}{\mathbb{E}_{X \sim Q}[-X]} \leq \inf_{Q':\mathbb{E}_{X \sim Q'}[X] < 0} \frac{\mathrm{KL}(Q'\|P)}{\mathbb{E}_{X \sim Q'}[-X]} (1 + \epsilon). \tag{16}
$$

*Proof.* Let $p \in \left(0, \min\left\{\frac{\mathbb{E}_{X \sim Q}[-X]}{1 + \mathbb{E}_{X \sim Q}[-X]}, 1 - \exp\left(-\frac{(\mathbb{E}_{X \sim Q}[X])^2}{2(\frac{1}{\epsilon}+1)}\right)\right\}\right)$, and let $Q_p = (1-p)Q + p\delta_{\{1\}}$ be the mixture of $Q$ and the point mass at $X = 1$. Let $\Lambda_p(\lambda) = \log \mathbb{E}_{X \sim Q_p}[e^{\lambda X}]$ be the logarithmic moment-generating function of $Q_p$ and $\lambda^* > 0$ be such that $\Lambda_p(\lambda^*) = 0$. Such $\lambda^*$ exists and satisfies $\Lambda_p'(\lambda^*) > 0$ since $\Lambda_p(0) = 0$ and $\Lambda_p'(0) = (1-p)\mathbb{E}_{X \sim Q}[X] + p < 0$ by $p \leq \frac{\mathbb{E}_{X \sim Q}[-X]}{1 + \mathbb{E}_{X \sim Q}[-X]}$.

Let $P_p$ be the distribution such that $\mathrm{d}P_p/\mathrm{d}Q_p(x) = e^{\lambda^* x - \Lambda_p(\lambda^*)} = e^{\lambda^* x}$. Then we have $\mathbb{E}_{X \sim P_p}[X] = \Lambda_p'(\lambda^*) > 0$. Here note that $\mathbb{E}_{X \sim P_p}[e^{-\lambda^* X}] = \mathbb{E}_{X \sim Q_p}[e^{-\lambda^* X} e^{\lambda^* X - \Lambda_p(\lambda^*)}] = 1$. Therefore, by Lemma 5 we have

$$
\inf_{Q':\mathbb{E}_{X \sim Q'}[X] < 0} \frac{\mathrm{KL}(Q'\|P_p)}{\mathbb{E}_{X \sim Q'}[-X]} = \lambda^*. \tag{17}
$$

On the other hand,

$$
\begin{aligned}
\mathrm{KL}(Q\|P_p) &= \mathbb{E}_{X \sim Q}\left[\mathbb{1}_{X<1} \log \frac{\mathrm{d}Q}{\mathrm{d}P_p}(X) + \mathbb{1}_{X=1} \log \frac{\mathrm{d}Q}{\mathrm{d}P_p}(X)\right] \\
&= \mathbb{E}_{X \sim Q}\left[\mathbb{1}_{X<1} \log \frac{1}{1-p} \frac{\mathrm{d}Q_p}{\mathrm{d}P_p}(X)\right] + Q(X=1) \log \frac{Q(X=1)}{P_p(X=1)} \\
&\leq \log \frac{1}{1-p} + \mathbb{E}_{X \sim Q}\left[\mathbb{1}_{X<1} \log \frac{\mathrm{d}Q_p}{\mathrm{d}P_p}(X)\right] + Q(X=1) \log \frac{Q(X=1)}{Q_p(X=1)e^{\lambda^* \cdot 1}} \\
&= \log \frac{1}{1-p} + \mathbb{E}_{X \sim Q}\left[\mathbb{1}_{X<1}(-\lambda^* X)\right] - Q(X=1)\lambda^* \cdot 1 + Q(X=1) \log \frac{Q(X=1)}{Q_p(X=1)} \\
&= \log \frac{1}{1-p} + \lambda^* \mathbb{E}_{X \sim Q}[-X] + Q(X=1) \log \frac{Q(X=1)}{Q_p(X=1)} \\
&\leq \log \frac{1}{1-p} + \lambda^* \mathbb{E}_{X \sim Q}[-X],
\end{aligned}
$$

which, combined with (17), implies that

$$\frac{\mathrm{KL}(Q\|P_p)}{\mathbb{E}_{X\sim Q}[-X]} \le \inf_{Q':\mathbb{E}_{X\sim Q'}[X]<0} \frac{\mathrm{KL}(Q'\|P)}{\mathbb{E}_{X\sim Q'}[-X]} + \frac{\log\frac{1}{1-p}}{\mathbb{E}_{X\sim Q}[-X]}. \tag{18}$$

Comparing (17) and (18) with (16), we see that it is sufficient to show that

$$\frac{\log\frac{1}{1-p}}{\mathbb{E}_{X\sim Q}[-X]} \le \epsilon\lambda^* \tag{19}$$

to show (16). Note that we obtain from Pinsker's inequality that

$$\lambda^* \ge \frac{\mathrm{KL}(Q\|P_p) - \log\frac{1}{1-p}}{\mathbb{E}_{X\sim Q}[-X]}$$

$$\ge \frac{2\left(\frac{\mathbb{E}_{X\sim Q}[X]}{2} - \frac{\mathbb{E}_{X\sim P_p}[X]}{2}\right)^2 - \log\frac{1}{1-p}}{\mathbb{E}_{X\sim Q}[-X]}$$

$$\ge \frac{\frac{(\mathbb{E}_{X\sim Q}[X])^2}{2} - \log\frac{1}{1-p}}{\mathbb{E}_{X\sim Q}[-X]},$$

recalling that $P_p$ and $Q$ are supported over $[-1, 1]$, and have positive and negative expectations, respectively. Therefore we obtain (19) since

$$\frac{1}{\lambda^*}\frac{\log\frac{1}{1-p}}{\mathbb{E}_{X\sim Q}[-X]} \le \frac{\log\frac{1}{1-p}}{\frac{(\mathbb{E}_{X\sim Q}[X])^2}{2} - \log\frac{1}{1-p}} \le \epsilon,$$

where the last inequality follows from since $p \le 1 - \exp\left(-\frac{(\mathbb{E}_{X\sim Q}[X])^2}{2(\frac{1}{\epsilon}+1)}\right)$. $\qquad\square$

## D Detailed Proof of the Lower Bound on the Probability of Ruin (Theorem 1 and Proposition 7)

In this section, we give a detailed proof of Theorem 1 and Proposition 7. The proof of the lower bound both in the case of multinomial arms of support $\{-1, 0, 1\}$ (Theorem 1) and in the general case of rewards bounded in $[-1, 1]$ (Proposition 7) stems from the asymptotic lower bound, which is common to both aforementioned cases and is given in Theorem 4. The passage from the asymptotic to the non-asymptotic bound relies on sub-additivity properties, which is given in Lemma 7 and for which formulas differ depending on the case considered.

If for all the arms $k \in [K]$,

$$\inf_{Q_k:\mathbb{E}_{X\sim Q_k}[X]<0} \frac{\mathrm{KL}(Q_k\|F_k)}{\mathbb{E}_{X\sim Q_k}[-X]} = 0,$$

then the result becomes trivial. This is why we are going to make the following assumption in the proof:

**Assumption 4.** *There exists an arm $k \in [K]$ such that $P(\tau(B, k) = \infty) > 0$.*

### D.1 Details of the Proof of Lemma 2

In this subsection, we provide the justification for

$$\lim_{B\to+\infty} Q(\mathcal{H}_\tau \in T(Q)) = 1,$$

which was omitted in the main text.

For any $t \geq 1$ and any $n = (n_1, \ldots, n_K)$ such that $\sum_{k=1}^{K} n_k = t$, we introduce the following random events:

$$
\begin{aligned}
U(n,t) &:= \left\{ \left| \sum_{k=1}^{K} \left( n_k \mathrm{KL}(Q_k \| F_k) - \sum_{m=1}^{n_k} \log \frac{dQ_k}{dF_k}(y_k^m) \right) \right| \leq \frac{t}{B^{\frac{1}{4}}} \right\}, \\
V(n,t) &:= \left\{ \left| \sum_{k=1}^{K} \left( n_k \mathbb{E}_{X \sim Q_k}[X] - \sum_{m=1}^{n_k} y_k^m \right) \right| \leq \frac{t\Delta_Q}{B^{\frac{1}{4}}} \right\}, \\
W(n,t) &:= \left\{ \sum_{k=1}^{K} \sum_{m=1}^{n_k} y_k^m \leq -\frac{t\Delta_Q}{2} \right\}.
\end{aligned}
$$

Let $h_t$ be a realization of $\mathcal{H}_\tau$. Then, please note that the probability of the event $\{h_t \in T(Q)\}$ is uniformly bounded independently of the policy $\pi$ by the probability of the following event:

$$
\forall n = (n_1, \ldots, n_K) \ \text{s.t.} \ \sum_{k=1}^{K} n_k = t : \ U(n,t), \ V(n,t), \ W(n,t).
$$

For any $k \in [K]$, let

$$
d_k := \max_{y_1 \in [-1,1]} \log \frac{dQ_k}{dF_k}(y_1) - \min_{y_2 \in [-1,1]} \log \frac{dQ_k}{dF_k}(y_2) \quad \text{and} \quad D := \max_{k \in [K]} d_k.
$$

Then, a direct application of Hoeffding's inequality gives the bounds

$$
Q(U(n,t)^c) \leq 2 \exp\left( -\frac{2t}{D\sqrt{B}} \right),
$$

$$
Q(V(n,t)^c) \leq 2 \exp\left( -\frac{t\Delta_Q^2}{2\sqrt{B}} \right),
$$

$$
Q(W(n,t)^c) \leq \exp\left( -\frac{t\Delta_Q^2}{2} \right).
$$

Let $C := \max\left\{ \frac{D}{2}, \frac{2}{\Delta_Q^2} \right\}$, this implies

$$
\max\left\{ Q(U(n,t)^c), Q(V(n,t)^c), 2Q(W(n,t)^c) \right\} \leq 2 \exp\left( -\frac{t}{C\sqrt{B}} \right).
$$

Using this result, as well as a union bound, we can then bound the probability

$$
\begin{aligned}
Q(h_t \notin T(Q)) &\leq Q\left( \exists n = (n_1, \ldots, n_K) : U(n,t)^c \text{ or } V(n,t)^c \text{ or } W(n,t)^c \right) \\
&\leq \sum_{\substack{n=(n_1,\ldots,n_K) \\ n_1+\cdots+n_K=t}} Q\left( U(n,t)^c \text{ or } V(n,t)^c \text{ or } W(n,t)^c \right) \\
&\leq \sum_{\substack{n=(n_1,\ldots,n_K) \\ n_1+\cdots+n_K=t}} \left\{ Q\left( U(n,t)^c \right) + G\left( V(n,t)^c \right) + Q\left( W(n,t)^c \right) \right\} \\
&\leq 5(t+1)^K \exp\left( -\frac{t}{C\sqrt{B}} \right).
\end{aligned} \tag{20}
$$

We can now bound the desired probability by first using the decomposition

$$
Q(\mathcal{H}_\tau \notin T(Q)) = Q\left( \tau > \frac{3B}{\Delta_Q}, \mathcal{H}_\tau \notin T(Q) \right) + Q\left( \tau \leq \frac{3B}{\Delta_Q}, \mathcal{H}_\tau \notin T(Q) \right).
$$

Then, the first term can be easily bounded using Hoeffding's inequality again:

$$
\begin{aligned}
Q\left(\tau > \frac{3B}{\Delta_Q}, \mathcal{H}_\tau \notin T(Q)\right) &\le Q\left(\tau > \frac{3B}{\Delta_Q}\right) \\
&= Q\left(\forall t \in \left\{1, \ldots, \frac{3B}{\Delta_Q}\right\}, \exists n = (n_1, \ldots, n_K): \ W(n, t)^c\right) \\
&\le Q\left(\exists n = (n_1, \ldots, n_K): \ W(n, B)^c\right) \\
&\le \sum_{\substack{(n_1, \ldots, n_K) \\ n_1 + \cdots + n_K = B}} Q\left(W(n, B)^c\right) \\
&\le (B+1)^K \exp\left(-\frac{t}{C\sqrt{B}}\right).
\end{aligned}
$$

The second term in the decomposition is decomposed using a union bound and (20):

$$
\begin{aligned}
Q\left(\tau \le \frac{3B}{\Delta_Q}, \mathcal{H}_\tau \notin T(Q)\right) &\le Q\left(\exists t \in \left\{B, \ldots, \frac{3B}{\Delta_Q}\right\}: \ h_t \notin T(Q)\right) \\
&\le \sum_{t=B}^{\frac{3B}{\Delta_Q}} Q(h_t \notin T(Q)) \\
&\le \sum_{t=B}^{\frac{3B}{\Delta_Q}} 5(t+1)^K \exp\left(-\frac{t}{C\sqrt{B}}\right).
\end{aligned}
$$

Then, for any $t \ge B \ge (2KC)^2$, it holds that $5(t+1)^K \exp\left(-\frac{t}{C\sqrt{B}}\right) \le 5\exp\left(-\frac{t}{2C\sqrt{B}}\right)$ and therefore

$$
\begin{aligned}
Q\left(\tau \le \frac{3B}{\Delta_Q}, \mathcal{H}_\tau \notin T(Q)\right) &\le 5\sum_{t=B}^{\frac{3B}{\Delta_Q}} \exp\left(-\frac{t}{2C\sqrt{B}}\right) \\
&= 5 \times \frac{e^{-\frac{\sqrt{B}}{2C}} - e^{-\frac{\frac{3B}{\Delta_Q}+1}{2C\sqrt{B}}}}{1 - e^{-\frac{1}{2C\sqrt{B}}}}.
\end{aligned}
$$

We then deduce that, for $B \ge (2KC)^2$,

$$
Q(\mathcal{H}_\tau \notin T(Q)) \le (B+1)^K e^{-\frac{\sqrt{B}}{C}} + 5 \times \frac{e^{-\frac{\sqrt{B}}{2C}} - e^{-\frac{\frac{3B}{\Delta_Q}+1}{2C\sqrt{B}}}}{1 - e^{-\frac{1}{2C\sqrt{B}}}},
$$

and therefore, that

$$
\lim_{B\to+\infty} Q(\mathcal{H}_\tau \in T(Q)) = 1.
$$

∎

## D.2 Asymptotic Lower Bound

The main result of this subsection is the asymptotic lower bound on the probability of ruin. This result will serve as a basis in the proof of the non-asymptotic lower bound of both Theorem 1 and Proposition 7, and for that reason, it is conducted in the general case of arm distributions in $\mathcal{F}_{[-1,1]}^K$.

**Theorem 4.** *Let $(\alpha_k)_{k\in[K]}$ such that for any $k \in [K]$, $\alpha_k > 0$ and $\sum_{k=1}^K \alpha_k = 1$. There exists no policy $\pi$ such that, for any set of arms $(F_1, \ldots, F_K)$,*

$$
\liminf_{B\to+\infty} \frac{1}{B} \log P_{(F_1, \ldots, F_K)}\left(\tau(B, \pi) < \infty\right) \le -\sum_{k=1}^K \alpha_k \inf_{Q_k: \mathbb{E}_{X\sim Q_k}[X]<0} \frac{\mathrm{KL}(Q_k\|F_k)}{\mathbb{E}_{X\sim Q_k}[-X]},
$$

*with a strict inequality for some* $(F_1, \ldots, F_K)$.

We define, for any tuple of distributions $F = (F_1, \ldots, F_K)$,

$$A_F(\alpha_1, \ldots, \alpha_K) := \sum_{k=1}^{K} \alpha_k \inf_{Q_k : \mathbb{E}_{X \sim Q_k}[X] < 0} \frac{\mathrm{KL}(Q_k \| F_k)}{\mathbb{E}_{X \sim Q_k}[-X]},$$

which we write $A_F$ in the absence of ambiguity on the $(\alpha_k)_{k \in [K]}$. The inequality in Theorem 4 means that

$$\liminf_{B \to +\infty} \sup_{F} \frac{\log P_{(F_1, \ldots, F_K)}(\tau(B, \pi) < \infty)}{B A_F} \geq -1. \tag{21}$$

*Proof.* Recall from Lemma 2 that for any $Q = (Q_1, \ldots, Q_K)$ such that $\mathbb{E}_{Q_i}[X] < 0$ for any $i \in [K]$,

$$\liminf_{B \to +\infty} \frac{1}{B} \log P_{(F_1, \ldots, F_K)}(\tau(B, \pi) < \infty) \geq -\sum_{k=1}^{K} \beta_k(Q) \frac{\mathrm{KL}(Q_k \| F_k)}{\mathbb{E}_{X \sim Q_k}[-X]}, \tag{22}$$

where $\beta(Q) = (\beta_1(Q), \ldots, \beta_K(Q))$ satisfies

$$\forall k \in [K], \beta_k(Q) \geq 0 \quad \text{and} \quad \sum_{k=1}^{K} \beta_k(Q) = 1. \tag{23}$$

Let us fix $(\alpha_k)_{k \in [K]}$ such that for any $k \in [K]$, $\alpha_k > 0$ and $\sum_{k=1}^{K} \alpha_k = 1$. We are going to show that no policy $\pi$ can achieve both

$$\forall (F_1, \ldots, F_K), \ \liminf_{B \to +\infty} \frac{1}{B} \log P_{(F_1, \ldots, F_K)}(\tau(B, \pi) < \infty) \leq -\sum_{k=1}^{K} \alpha_k \inf_{Q_k : \mathbb{E}_{X \sim Q_k}[X] < 0} \frac{\mathrm{KL}(Q_k \| F_k)}{\mathbb{E}_{X \sim Q_k}[-X]} \tag{24}$$

and

$$\exists (F_1, \ldots, F_K), \ \liminf_{B \to +\infty} \frac{1}{B} \log P_{(F_1, \ldots, F_K)}(\tau(B, \pi) < \infty) < -\sum_{k=1}^{K} \alpha_k \inf_{Q_k : \mathbb{E}_{X \sim Q_k}[X] < 0} \frac{\mathrm{KL}(Q_k \| F_k)}{\mathbb{E}_{X \sim Q_k}[-X]}. \tag{25}$$

Let us then fix a policy $\pi$ such that there exists a distribution $\bar{P} = (\bar{P}_1, \ldots, \bar{P}_K)$ and $\epsilon > 0$ such that

$$\liminf_{B \to +\infty} \frac{1}{B} \log P_{(\bar{P}_1, \ldots, \bar{P}_K)}(\tau(B, \pi) < \infty) \leq -\sum_{k=1}^{K} \alpha_k \bar{\gamma}_k - \epsilon, \tag{26}$$

where we denoted, for any $k \in [K]$,

$$\bar{\gamma}_k := \inf_{Q : \mathbb{E}_{X \sim Q}[X] < 0} \frac{\mathrm{KL}(Q \| \bar{P}_k)}{\mathbb{E}_{X \sim Q}[-X]} \quad \text{and} \quad \bar{\gamma}_{\max} := \max_{k \in [K]} \bar{\gamma}_k > 0.$$

Please note that the positivity of $\bar{\gamma}_{\max}$ relies on Assumption 4. We are going to show that there exists $\bar{P}^* = (\bar{P}_1^*, \ldots, \bar{P}_K^*)$ such that, denoting

$$\bar{\gamma}_k^* := \inf_{Q : \mathbb{E}_{X \sim Q}[X] < 0} \frac{\mathrm{KL}(Q \| \bar{P}_k^*)}{\mathbb{E}_{X \sim Q}[-X]}, \ \bar{\gamma}_{\min}^* := \min\{\bar{\gamma}_k^* : k \in [K], \bar{\gamma}_k^* > 0\} \quad \text{and} \quad \epsilon' := \frac{\epsilon \alpha_{\min} \bar{\gamma}_{\min}^*}{4(K-1) \bar{\gamma}_{\max}},$$

the following holds:

$$\liminf_{B \to +\infty} \frac{1}{B} \log P_{(\bar{P}_1^*, \ldots, \bar{P}_K^*)}(\tau(B, \pi) < \infty) \geq -\sum_{k=1}^{K} \alpha_k \bar{\gamma}_k^* + \epsilon'.$$

We define $\bar{Q} = (\bar{Q}_1, \ldots, \bar{Q}_K)$ such that, for any $k \in [K], \mathbb{E}_{X \sim \bar{Q}_k}[X] < 0$ and

$$\frac{\mathrm{KL}(\bar{Q}_k \| \bar{P}_k)}{\mathbb{E}_{X \sim \bar{Q}_k}[-X]} \leq \bar{\gamma}_k + \frac{\epsilon}{2}. \tag{27}$$

Denoting $\alpha_{\min} := \min_{k \in [K]} \alpha_k > 0$, we then introduce the set

$$\mathcal{K} := \left\{ k \in [K] : \beta_k(\bar{Q}) \leq \alpha_k - \frac{\alpha_{\min}\epsilon}{2(K-1)\bar{\gamma}_{\max}} \right\}.$$

Let us prove that $\mathcal{K}$ is not empty. Indeed, (26) can be re-written as

$$\liminf_{B \to +\infty} \frac{1}{B} \log P_{(\bar{P}_1, \ldots, \bar{P}_K)}(\tau(B, \pi) < \infty) \leq -\sum_{k=1}^{K} \alpha_k \bar{\gamma}_k - \epsilon$$

$$= -\sum_{k=1}^{K} (\alpha_k \bar{\gamma}_k + \alpha_k \epsilon)$$

$$= -\sum_{k=1}^{K} \left( \alpha_k + \frac{\alpha_k \epsilon}{\bar{\gamma}_k} \right) \bar{\gamma}_k. \tag{28}$$

Then, applying (22) to $Q = \bar{Q}$ and $F = \bar{P}$ and using (27), we deduce that

$$\liminf_{B \to +\infty} \frac{1}{B} \log P_{(\bar{P}_1, \ldots, \bar{P}_K)}(\tau(B, \pi) < \infty) \geq -\sum_{k=1}^{K} \beta_k(\bar{Q}) \frac{\mathrm{KL}(\bar{Q}_k \| \bar{P}_k)}{\mathbb{E}_{X \sim \bar{Q}_k}[-X]}$$

$$\geq -\sum_{k=1}^{K} \beta_k(\bar{Q}) \left( \bar{\gamma}_k + \frac{\epsilon}{2} \right)$$

$$= -\sum_{k=1}^{K} \left( \beta_k(\bar{Q}) + \frac{\alpha_k \epsilon}{2\bar{\gamma}_k} \right) \bar{\gamma}_k. \tag{29}$$

Then, we deduce from (28) and (29) that

$$-\sum_{k=1}^{K} \left( \alpha_k + \frac{\alpha_k \epsilon}{\bar{\gamma}_k} \right) \bar{\gamma}_k \geq -\sum_{k=1}^{K} \left( \beta_k(\bar{Q}) + \frac{\alpha_k \epsilon}{2\bar{\gamma}_k} \right) \bar{\gamma}_k,$$

or in other words, that

$$\sum_{k=1}^{K} \left( \beta_k(\bar{Q}) - \alpha_k - \frac{\alpha_k \epsilon}{2\bar{\gamma}_k} \right) \underbrace{\bar{\gamma}_k}_{\geq 0} \geq 0.$$

This is equivalent to

$$\sum_{k:\bar{\gamma}_k > 0} \left( \beta_k(\bar{Q}) - \alpha_k - \frac{\alpha_k \epsilon}{2\bar{\gamma}_k} \right) \underbrace{\bar{\gamma}_k}_{>0} \geq 0.$$

We then deduce that there exists $k_0 \in [K]$ such that $\beta_{k_0}(\bar{Q}) \geq \alpha_k + \frac{\alpha_k \epsilon}{2\bar{\gamma}_{k_0}}$. With (23), it implies that

$$\sum_{j \neq k_0} \beta_j(\bar{Q}) = 1 - \beta_{k_0}(\bar{Q}) \leq 1 - \alpha_{k_0} - \frac{\alpha_{k_0} \epsilon}{2\bar{\gamma}_{k_0}} \leq \sum_{j \neq k_0} \alpha_j - \frac{\alpha_{\min} \epsilon}{2\bar{\gamma}_{\max}}.$$

We deduce that there exists $j \in [K]$ such that $\beta_j(\bar{Q}) \leq \alpha_j - \frac{\alpha_{\min}\epsilon}{2(K-1)\bar{\gamma}_{\max}}$, proving that $\mathcal{K}$ is not empty. Then, we define the distribution $\bar{P}^* = (\bar{P}_1^*, \ldots, \bar{P}_K^*)$ as follows:

- for any $k \notin \mathcal{K}$, let $\bar{P}_k^* := \bar{Q}_k$, and please note that $\mathbb{E}_{X \sim \bar{P}_k^*}[X] < 0$;

- for any $k \in \mathcal{K}$, let $\bar{P}_k^*$ a distribution such that $\mathbb{E}_{X \sim \bar{P}_k^*}[X] > 0$ and

$$\frac{\mathrm{KL}(\bar{Q}_k \| \bar{P}_k^*)}{\mathbb{E}_{X \sim \bar{Q}_k}[-X]} \leq \bar{\gamma}_k^* \left(1 + \frac{\alpha_{\min}\epsilon}{4(K-1)\bar{\gamma}_{\max}}\right) = \bar{\gamma}_k^* + \frac{\alpha_{\min}\epsilon \bar{\gamma}_k^*}{4(K-1)\bar{\gamma}_{\max}}, \tag{30}$$

where $\bar{\gamma}_k^* = \inf_{Q:\mathbb{E}_{X \sim Q}[X]<0} \frac{\mathrm{KL}(Q\|\bar{P}_k^*)}{\mathbb{E}_{X \sim Q}[-X]}$ and $\bar{\gamma}_{\min}^* = \min\{\bar{\gamma}_k^* : k \in \mathcal{K}, \bar{\gamma}_k^* > 0\}$. Note that this distribution $\bar{P}_k^*$ indeed exists by Lemma 6.

Since $\mathcal{K} \neq \emptyset$ and by definition of $\bar{P}^*$, we have

$$\sum_{k=1}^K \beta_k(\bar{Q}) \frac{\mathrm{KL}(\bar{Q}_k \| \bar{P}_k^*)}{\mathbb{E}_{X \sim \bar{Q}_k}[-X]} = \sum_{k \notin \mathcal{K}} \beta_k(\bar{Q}) \underbrace{\frac{\mathrm{KL}(\bar{Q}_k \| \bar{Q}_k)}{\mathbb{E}_{X \sim \bar{Q}_k}[-X]}}_{0} + \sum_{k \in \mathcal{K}} \beta_k(\bar{Q}) \frac{\mathrm{KL}(\bar{Q}_k \| \bar{P}_k^*)}{\mathbb{E}_{X \sim \bar{Q}_k}[-X]}$$

$$= \sum_{k \in \mathcal{K}} \beta_k(\bar{Q}) \frac{\mathrm{KL}(\bar{Q}_k \| \bar{P}_k^*)}{\mathbb{E}_{X \sim \bar{Q}_k}[-X]}.$$

Then, (30) implies that

$$\sum_{k \in \mathcal{K}} \beta_k(\bar{Q}) \frac{\mathrm{KL}(\bar{Q}_k \| \bar{P}_k^*)}{\mathbb{E}_{X \sim \bar{Q}_k}[-X]} \leq \sum_{k \in \mathcal{K}} \beta_k(\bar{Q}) \bar{\gamma}_k^* + \frac{\alpha_{\min}\epsilon}{4(K-1)\bar{\gamma}_{\max}} \sum_{k \in \mathcal{K}} \beta_k(\bar{Q}) \bar{\gamma}_k^*$$

$$\leq \sum_{k \in \mathcal{K}} \beta_k(\bar{Q}) \bar{\gamma}_k^* + \frac{\alpha_{\min}\epsilon}{4(K-1)} \sum_{k \in \mathcal{K}} \frac{\bar{\gamma}_k^*}{\bar{\gamma}_{\max}}.$$

By definition of $\mathcal{K}$, for any $k \in \mathcal{K}, \beta_k(\bar{Q}) \leq \alpha_k - \frac{\alpha_{\min}\epsilon}{2(K-1)\bar{\gamma}_{\max}}$ and we deduce that

$$\sum_{k \in \mathcal{K}} \beta_k(\bar{Q}) \frac{\mathrm{KL}(\bar{Q}_k \| \bar{P}_k^*)}{\mathbb{E}_{X \sim \bar{Q}_k}[-X]} \leq \sum_{k \in \mathcal{K}} \alpha_k \bar{\gamma}_k^* - \frac{\alpha_{\min}\epsilon}{2(K-1)} \sum_{k \in \mathcal{K}} \frac{\bar{\gamma}_k^*}{\bar{\gamma}_{\max}} + \frac{\alpha_{\min}\epsilon}{4(K-1)} \sum_{k \in \mathcal{K}} \frac{\bar{\gamma}_k^*}{\bar{\gamma}_{\max}}$$

$$\leq \sum_{k \in \mathcal{K}} \alpha_k \bar{\gamma}_k^* - \frac{\alpha_{\min}\epsilon \bar{\gamma}_{\min}^*}{4(K-1)\bar{\gamma}_{\max}}, \tag{31}$$

where the inequality (31) comes from the fact that $\mathcal{K}$ is not empty. Injecting (31) in (22) (with $P = P^*$ and $Q = \bar{Q}$), we have:

$$\liminf_{B \to +\infty} \frac{1}{B} \log P_{(\bar{P}_1^*, \ldots, \bar{P}_K^*)}(\tau(B, \pi) < \infty) \geq -\sum_{k=1}^K \alpha_k \bar{\gamma}_k^* + \frac{\alpha_{\min}\epsilon \bar{\gamma}_{\min}^*}{4(K-1)\bar{\gamma}_{\max}}.$$

Recall that, by definition,

$$\epsilon' = \frac{\epsilon \alpha_{\min} \bar{\gamma}_{\min}^*}{4(K-1)\bar{\gamma}_{\max}}.$$

We deduce the following

$$\liminf_{B \to +\infty} \frac{1}{B} \log P_{(\bar{P}_1^*, \ldots, \bar{P}_K^*)}(\tau(B, \pi) < \infty) \geq -\sum_{k=1}^K \alpha_k \bar{\gamma}_k^* + \epsilon',$$

which concludes the proof of the theorem. $\qquad\square$

### D.3 Sub-additivity of the Optimal Log Probability of Ruin

The passage from the asymptotic to the non-asymptotic lower bound on the probability of ruin relies on sub-additivity properties described in the next lemma, which is the main result of this subsection.

**Lemma 7.** *Let $\tilde{t} \in \mathbb{R}_+^* \cup \{+\infty\}$. For any $B_1, B_2 > 0$, we have*

$$
\inf_\pi \sup_F \frac{\log P\left(\tau(B_1 + B_2 + 1, \pi) < \tilde{t}\right)}{A_F} \leq \inf_\pi \sup_F \frac{\log P\left(\tau(B_1, \pi) < \tilde{t}\right)}{A_F} + \inf_\pi \sup_F \frac{\log P\left(\tau(B_2, \pi) < \tilde{t}\right)}{A_F}.
$$

*In the case of multinomial arm distributions of support $\{-1, 0, 1\}$, if $B_1$ and $B_2$ are positive integers, the previous bound can be refined as*

$$
\inf_\pi \sup_F \frac{\log P\left(\tau(B_1 + B_2, \pi) < \tilde{t}\right)}{A_F} \leq \inf_\pi \sup_F \frac{\log P\left(\tau(B_1, \pi) < \tilde{t}\right)}{A_F} + \inf_\pi \sup_F \frac{\log P\left(\tau(B_2, \pi) < \tilde{t}\right)}{A_F}.
$$

*Proof.* Let

$$
\pi_1^* \in \arg\min_\pi \sup_F \frac{\log P\left(\tau(B_1, \pi) < \tilde{t}\right)}{A_F},
$$
$$
\pi_2^* \in \arg\min_\pi \sup_F \frac{\log P\left(\tau(B_2, \pi) < \tilde{t}\right)}{A_F}.
$$

Besides, we denote by $\tilde{\pi}$ the policy such that

$$
\tilde{\pi}_t := \begin{cases} (\pi_1^*)_t & \text{if } t < \min\left(\tau(B_1, \tilde{\pi}), \tilde{t}\right), \\ (\pi_2^*)_{t - \tau(B_1, \tilde{\pi})} & \text{(ignoring the previously observed rewards) otherwise.} \end{cases}
$$

Let $B_2' \in \{B_2, B_2 + 1\}$. Then, it is clear that

$$
P\left(\tau(B_1 + B_2', \tilde{\pi}) < \tilde{t}\right)
$$
$$
= P\left(\exists 1 \leq t_{1+2} \leq \tilde{t} : B_1 + B_2' + \sum_{s=1}^{t_{1+2}} X_s^{\tilde{\pi}_s} < 0\right)
$$
$$
= P\left(\exists 1 \leq t_1, t_{1+2} \leq \tilde{t} : B_1 + \sum_{s=1}^{t_1} X_s^{\tilde{\pi}_s} < 0, \ B_1 + B_2' + \sum_{s=1}^{t_{1+2}} X_s^{\tilde{\pi}_s} < 0\right)
$$
$$
= P\left(\exists 1 \leq t_1 \leq \tilde{t} : B_1 + \sum_{s=1}^{t_1} X_s^{\tilde{\pi}_s} < 0\right)
$$
$$
\times P\left(\exists \tau(B_1, \tilde{\pi}) \leq t_{1+2} \leq \tilde{t} : B_1 + B_2' + \sum_{s=1}^{t_{1+2}} X_s^{\tilde{\pi}_s} < 0 \,\middle|\, \tau(B_1, \tilde{\pi}) < \tilde{t}\right)
$$
$$
= P(\tau(B_1, \pi_1^*) < \tilde{t}) \times P\left(\exists \tau(B_1, \tilde{\pi}) \leq t_{1+2} \leq \tilde{t} : B_1 + B_2' + \sum_{s=1}^{t_{1+2}} X_s^{\tilde{\pi}_s} < 0 \,\middle|\, \tau(B_1, \pi_1^*) < \tilde{t}\right)
$$
$$
= P(\tau(B_1, \pi_1^*) < \tilde{t})
$$
$$
\times P\left(\exists \tau(B_1, \tilde{\pi}) \leq t_{1+2} \leq \tilde{t} : B_1 + \sum_{s=1}^{\tau(B_1, \pi_1^*)} X_s^{(\pi_1^*)_s} + B_2' + \sum_{s=\tau(B_1, \pi_1^*)+1}^{t_{1+2}} X_s^{(\pi_2^*)_s} < 0 \,\middle|\, \tau(B_1, \pi_1^*) < \tilde{t}\right).
$$

From there, we are going to study separately the general case and the case of multinomial distributions of support $\{-1, 0, 1\}$.

**First case:**    in the general case, we choose $B_2' = B_2 + 1$, and since the rewards are bounded in $[-1, 1]$, then

$$\tau(B_1, \pi_1^*) < \tilde{t} \implies B_1 + \sum_{s=1}^{\tau(B_1, \pi_1^*)} X_s^{(\pi_1^*)_s} + 1 \geq 0.$$

Hence,

$$
\begin{aligned}
&P\left(\tau(B_1 + B_2 + 1, \tilde{\pi}) < \tilde{t}\right) \\
&\leq P(\tau(B_1, \pi_1^*) < \tilde{t}) \\
&\quad \times P\left(\exists \tau(B_1, \pi_1^*) \leq t_{1+2} \leq \tilde{t} : B_2 + \sum_{s=\tau(B_1, \pi_1^*)+1}^{t_{1+2}} X_s^{(\pi_2^*)_s} < 0 \middle| \tau(B_1, \pi_1^*) < \tilde{t}\right) \\
&\leq P(\tau(B_1, \pi_1^*) < \tilde{t}) \\
&\quad \times P\left(\exists \tau(B_1, \pi_1^*) \leq t_{1+2} \leq \tilde{t} + \tau(B_1, \pi_1^*) : B_2 + \sum_{s=\tau(B_1, \pi_1^*)+1}^{t_{1+2}} X_s^{(\pi_2^*)_s} < 0 \middle| \tau(B_1, \pi_1^*) < \tilde{t}\right) \\
&= P(\tau(B_1, \pi_1^*) < \tilde{t}) \times P(\tau(B_2, \pi_2^*) < \tilde{t}).
\end{aligned}
$$

This yields

$$
\begin{aligned}
\inf_{\pi} \sup_{F} \frac{\log P\left(\tau(B_1 + B_2 + 1, \pi) < \tilde{t}\right)}{A_F} &\leq \sup_{F} \frac{\log P\left(\tau(B_1 + B_2 + 1, \tilde{\pi}) < \tilde{t}\right)}{A_F} \\
&\leq \sup_{F} \frac{\log P\left(\tau(B_1, \pi_1^*) < \tilde{t}\right)}{A_F} + \sup_{F} \frac{\log P\left(\tau(B_2, \pi_2^*) < \tilde{t}\right)}{A_F} \\
&= \inf_{\pi} \sup_{F} \frac{\log P\left(\tau(B_1, \pi) < \tilde{t}\right)}{A_F} + \inf_{\pi} \sup_{F} \frac{\log P\left(\tau(B_2, \pi_2^*) < \tilde{t}\right)}{A_F},
\end{aligned}
$$

which concludes the general case.

**Second case:**    in the case of multinomial arm distributions of support $\{-1, 0, 1\}$,

$$\tau(B_1, \pi_1^*) < \tilde{t} \implies B_1 + \sum_{s=1}^{\tau(B_1, \pi_1^*)} X_s^{(\pi_1^*)_s} = 0.$$

Therefore, choosing $B_2' = B_2$, we have

$$
\begin{aligned}
&P\left(\tau(B_1 + B_2, \tilde{\pi}) < \tilde{t}\right) \\
&\leq P(\tau(B_1, \pi_1^*) < \tilde{t}) \\
&\quad \times P\left(\exists \tau(B_1, \pi_1^*) \leq t_{1+2} \leq \tilde{t} : B_2 + \sum_{s=\tau(B_1, \pi_1^*)+1}^{t_{1+2}} X_s^{(\pi_2^*)_s} < 0 \middle| \tau(B_1, \pi_1^*) < \tilde{t}\right) \\
&\leq P(\tau(B_1, \pi_1^*) < \tilde{t}) \\
&\quad \times P\left(\exists \tau(B_1, \pi_1^*) \leq t_{1+2} \leq \tilde{t} + \tau(B_1, \pi_1^*) : B_2 + \sum_{s=\tau(B_1, \pi_1^*)+1}^{t_{1+2}} X_s^{(\pi_2^*)_s} < 0 \middle| \tau(B_1, \pi_1^*) < \tilde{t}\right) \\
&= P(\tau(B_1, \pi_1^*) < \tilde{t}) \times P(\tau(B_2, \pi_2^*) < \tilde{t}).
\end{aligned}
$$

This yields

$$
\begin{aligned}
\inf_{\pi} \sup_{F} \frac{\log P\left(\tau(B_1 + B_2, \pi) < \tilde{t}\right)}{A_F} &\leq \sup_{F} \frac{\log P\left(\tau(B_1 + B_2, \tilde{\pi}) < \tilde{t}\right)}{A_F} \\
&\leq \sup_{F} \frac{\log P\left(\tau(B_1, \pi_1^*) < \tilde{t}\right)}{A_F} + \sup_{F} \frac{\log P\left(\tau(B_2, \pi_2^*) < \tilde{t}\right)}{A_F} \\
&= \inf_{\pi} \sup_{F} \frac{\log P\left(\tau(B_1, \pi) < \tilde{t}\right)}{A_F} + \inf_{\pi} \sup_{F} \frac{\log P\left(\tau(B_2, \pi_2^*) < \tilde{t}\right)}{A_F},
\end{aligned}
$$

which concludes the multinomial case and the proof of the lemma. $\qquad\square$

### D.4 Proof of Theorem 1 and Proposition 7

Let $(\alpha_k)_{k \in [K]}$ such that for any $k \in [K], \alpha_k > 0$ and $\sum_{k=1}^{K} \alpha_k = 1$. Recall that, by definition,

$$
A_F = \sum_{k=1}^{K} \alpha_k \inf_{Q_k : \mathbb{E}_{X \sim Q_k}[X] < 0} \frac{\mathrm{KL}(Q_k \| F_k)}{\mathbb{E}_{X \sim Q_k}[-X]} > 0.
$$

Let $\pi$ be any policy, and $B_0 > 0$ an initial budget. For any $n \geq 1$, let us denote by $\pi^B$ the policy defined recursively on $\{B \geq B_0\}$, such that $\pi^{B_0} = \pi$ and for any $B \geq B_0, \pi_t^B = \pi_t$ for $t \leq \tau(B_0, \pi)$ and then $\pi_t^B = \pi_t^{B'}$ for $t \geq \tau(B_0, \pi) + 1$, where $B' = B + \sum_{s=1}^{\tau(B, \pi)}$. Concretely, $\pi^B$ restarts $\pi$ every time it exhausts the budget $B_0$.

From now, we are going to study separately the general case of rewards bounded in $[-1, 1]$ and the case of multinomial arms of support $\{-1, 0, 1\}$.

**First case:** in the case of multinomial arm distributions in $\mathcal{F}_{\{-1,0,1\}}^K$, we assume that, for any arm distributions $F = (F_1, \ldots, F_K)$,
$$
\frac{\log P_F\left(\tau(B_0, \pi) < \infty\right)}{A_F} \leq -B_0,
$$

and that there exist some arm distributions $\bar{F} = (\bar{F}_1, \ldots, \bar{F}_K)$ and $C_{\bar{F}} > 0$ such that

$$
\frac{\log P_{\bar{F}}\left(\tau(B_0, \pi) < \infty\right)}{A_{\bar{F}}} \leq -(B_0 + C_{\bar{F}}),
$$

and we will show that there is contradiction.

By Lemma 7, for any arm distributions $F$ and for any $n \geq 1$,

$$
\begin{aligned}
\frac{\log P_F\left(\tau(nB_0, \pi^{nB_0}) < \infty\right)}{A_F} &\leq n \times \frac{\log P\left(\tau(B_0, \pi^{B_0}) < \infty\right)}{A_F} \\
&\leq -nB_0.
\end{aligned}
$$

Consequently, for any arm distributions $F$,

$$
\limsup_{n \to +\infty} \frac{1}{nB_0} \log P\left(\tau(nB_0, \pi) < \infty\right) \leq -A_F. \tag{32}
$$

Furthermore, the same computation applied to $\bar{F}$ gives

$$
\begin{aligned}
\frac{\log P_{\bar{F}}\left(\tau(nB_0, \pi^{nB_0}) < \infty\right)}{A_{\bar{F}}} &\leq n \times \frac{\log P\left(\tau(B_0, \pi^{B_0}) < \infty\right)}{A_{\bar{F}}} \\
&\leq -n(B_0 + C_{\bar{F}}),
\end{aligned}
$$

which, in turn, implies

$$\limsup_{n \to +\infty} \frac{1}{nB_0} \log P\left(\tau(nB_0, \pi) < \infty\right) \leq -\frac{B_0 + C_{\bar{F}}}{B_0} A_{\bar{F}} < -A_{\bar{F}}. \tag{33}$$

Eqs. (32) and (33) contradict Theorem 4. Therefore, we deduce that if there exist some arm distributions $\bar{F}$ such that

$$\frac{\log P_{\bar{F}}\left(\tau(B_0, \pi) < \infty\right)}{A_{\bar{F}}} < -B_0,$$

then there also exist some arm distributions $F$ such that

$$\frac{\log P_F\left(\tau(B_0, \pi) < \infty\right)}{A_F} > -B_0,$$

concluding the multinomial case and the proof of Theorem 1.

**Second case:**  in the general case of arm distributions in $\mathcal{F}^K_{[-1,1]}$, we assume that, for any arm distributions $F = (F_1, \ldots, F_K)$,

$$\frac{\log P_F\left(\tau(B_0, \pi) < \infty\right)}{A_F} \leq -(B_0 + 1),$$

and that there exist some arm distributions $\bar{F} = (\bar{F}_1, \ldots, \bar{F}_K)$ and $C_{\bar{F}} > 0$ such that

$$\frac{\log P_{\bar{F}}\left(\tau(B_0, \pi) < \infty\right)}{A_{\bar{F}}} \leq -(B_0 + 1 + C_{\bar{F}}),$$

and show that there is contradiction.

By Lemma 7, for any arm distributions $F$ and for any $n \geq 1$,

$$\frac{\log P_F\left(\tau(nB_0 + (n-1), \pi^{nB_0}) < \infty\right)}{A_F} \leq n \times \frac{\log P\left(\tau(B_0, \pi^{B_0}) < \infty\right)}{A_F}$$
$$\leq -n(B_0 + 1).$$

Consequently, for any arm distributions $F$,

$$\limsup_{n \to +\infty} \frac{1}{nB_0 + (n-1)} \log P\left(\tau(nB_0 + (n-1), \pi) < \infty\right) \leq -A_F. \tag{34}$$

Furthermore, the same computation applied to $\bar{F}$ gives

$$\frac{\log P_{\bar{F}}\left(\tau(nB_0 + (n-1), \pi^{nB_0}) < \infty\right)}{A_{\bar{F}}} \leq n \times \frac{\log P\left(\tau(B_0, \pi^{B_0}) < \infty\right)}{A_{\bar{F}}}$$
$$\leq -n(B_0 + 1 + C_{\bar{F}}),$$

which in turn, implies,

$$\limsup_{n \to +\infty} \frac{1}{nB_0 + (n-1)} \log P\left(\tau(nB_0, \pi) < \infty\right) \leq -\frac{B_0 + 1 + C_{\bar{F}}}{B_0 + 1} < -1. \tag{35}$$

Eqs. (34) and (35) contradict Theorem 4. Therefore, we deduce that if there exist some arm distributions $\bar{F}$ such that

$$\frac{\log P_{\bar{F}}\left(\tau(B_0, \pi) < \infty\right)}{A_{\bar{F}}} < -(B_0 + 1),$$

then there also exist some arm distributions $F$ such that

$$\frac{\log P_F\left(\tau(B_0, \pi) < \infty\right)}{A_F} > -(B_0 + 1),$$

concluding the general case and the proof of Proposition 7. ∎

# E   Proof of Lemma 1

Please note that applying (22) to the case of one single arm ($K = 1$) gives that, for any $\epsilon_0 \in \left(0, \frac{1}{3}\right)$ and any distribution $Q$ which has a negative expectation,

$$\liminf_{B \to +\infty} \frac{1}{B} \log P(\tau(B, 1) < \infty) \geq - \frac{\mathrm{KL}(Q \| F_1)}{\mathbb{E}_{X \sim Q}[-X]}.$$

By taking $\epsilon_0 \downarrow 0$, we deduce that

$$\liminf_{B \to +\infty} \frac{1}{B} \log P(\tau(B, 1) < \infty) \geq - \inf_{F : \mathbb{E}_{X \sim F}[X] < 0} \frac{\mathrm{KL}(F \| F_1)}{\mathbb{E}_{X \sim F}[-X]}.$$

It thus remains to prove that for any $B > 0$,

$$\frac{1}{B} \log P(\tau(B, 1) < \infty) \leq - \inf_{F : \mathbb{E}_{X \sim F}[X] < 0} \frac{\mathrm{KL}(F \| F_1)}{\mathbb{E}_{X \sim F}[-X]}.$$

The result being trivial if $\mathbb{E}_{X \sim F_1}[X] \leq 0$, we assume that $\mathbb{E}_{X \sim F_1}[X] > 0$. Let us define the logarithmic moment-generating function of $X$ by $\Lambda(\lambda) := \log \mathbb{E}\left[e^{\lambda X}\right]$. By Lemma 5, there exists $\lambda' < 0$ such that $\Lambda(\lambda') = 0$ and it satisfies

$$\inf_{F : \mathbb{E}_{X \sim F}[X] < 0} \frac{\mathrm{KL}(F \| F_1)}{\mathbb{E}_{X \sim F}[-X]} = -\lambda'.$$

Now, let $X \sim F_1$ and let $X_1, X_2, \cdots \sim F_1$ be i.i.d copies of $X$. We write $S_n = \sum_{i=1}^{n} X_i$. We define $\tau := \inf\{n \geq 1 : S_n \leq -B\}$ and $\tau_T := \min(\tau, T)$ for any $T \in \mathbb{N}$. Since $\tau_T$ is a bounded stopping time, by the optional stopping theorem, it holds for any $T$ that

$$\mathbb{E}\left[e^{\lambda' S_{\tau_T}}\right] = 1.$$

On the other hand,

$$\begin{aligned}
1 &= \mathbb{E}\left[e^{\lambda' S_{\tau_T}}\right] \\
&= \mathbb{E}\left[\mathbb{1}_{\tau_T < T} e^{\lambda' S_{\tau_T}}\right] + \mathbb{E}\left[\mathbb{1}_{\tau_T = T} e^{\lambda' S_{\tau_T}}\right] \\
&\geq \mathbb{E}\left[\mathbb{1}_{\tau_T < T} e^{\lambda' S_{\tau_T}}\right] \\
&\geq e^{-\lambda' B} P(\tau_T < T),
\end{aligned}$$

which implies that

$$P(\tau < \infty) = \lim_{T \to \infty} P(\tau_T < T) \leq e^{\lambda' B}.$$

Therefore we obtain

$$\frac{1}{B} \log P(\tau < \infty) \leq \lambda' = - \inf_{F : \mathbb{E}_{X \sim F}[X] < 0} \frac{\mathrm{KL}(F \| F_1)}{\mathbb{E}_{X \sim F}[-X]},$$

which completes the proof. ∎

# F   Proof of the Upper Bound on the Reward of EXPLOIT Policies (Proposition 3)

We introduce the following notation. Let $K_+$ be the number of arms such that $P\left(\tau\left(\frac{B}{K}, k\right) = \infty\right) > 0$ for $k \in [K]$. Recall that we ordered the arms in order of decreasing expectation, and therefore $\mu_1 \geq \cdots \geq \mu_{K_+} > 0$. Then, by definition of $K_+$,

$$\forall k \in [K_+], P\left(\tau\left(\frac{B}{K}, k\right) = \infty\right) > 0 \ ; \ \forall j \in \{K_+ + 1, \ldots, K\}, P\left(\tau\left(\frac{B}{K}, j\right) = \infty\right) = 0.$$

### F.1 Preliminary Lemma

In this subsection, we express the expected cumulative reward of any policy in EXPLOIT as the product between $p^{\mathrm{EX}}$ and a convex combination of $\mu_1, \dots, \mu_{K_+}$, and we explicitly give the coefficients of the convex combination. This decomposition will be useful both as a first step in the proof of Proposition 3 and in the proof of the reward bound of EXPLOIT-UCB, as a particular instance of policy in EXPLOIT.

For any $S \subseteq [K_+]$, we define the event

$$\Pi_S := \left\{ \forall j \in S, \tau\left(\frac{B}{K}, j\right) \geq T \ \text{ and } \ \forall j \in [K_+] \setminus S, \tau\left(\frac{B}{K}, j\right) < \sqrt{T} \right\}.$$

Given a policy $\pi$, an arm $k \in [K_+]$ and a set $S \subseteq [K_+]$, we define the coefficient $n_{\pi,k}(S)$ as

$$n_{\pi,k}(S) := \frac{1}{T}\mathbb{E}\left[\sum_{t=1}^{T} \mathbb{1}_{\pi_t=k,\tau(B,\pi)\geq t-1} \middle| \Pi_S \right].$$

When there is no ambiguity on the policy $\pi$, we simply write $n_k(S)$. Please note that for any fixed policy $\pi$ and any set $S \subseteq [K_+]$,

$$\sum_{k\in S} n_k(S) \leq \sum_{k=1}^{K} n_k(S) = \frac{1}{T}\mathbb{E}\left[\sum_{t=1}^{T} \mathbb{1}_{\tau(B,\pi)\geq t-1} \middle| \Pi_S \right] \leq 1.$$

The following result holds.

**Lemma 8.** *For any policy $\pi$ within the framework EXPLOIT, the expected cumulative reward satisfies*

$$\mathbb{E}\left[\sum_{t=1}^{T} X_t^{\pi_t} \mathbb{1}_{\tau(B,\pi)\geq t-1}\right] = \sum_{k=1}^{K_+} \left(\sum_{S\subseteq[K_+]:k\in S} P(\Pi_S) n_k(S)\right) \mu_k \times T + o(T). \tag{36}$$

*Proof.* First, we write the reward as a sum over the arms of positive probability of survival:

$$\mathbb{E}\left[\sum_{t=1}^{T} X_t^{\pi_t} \mathbb{1}_{\tau(B,\pi)\geq t-1}\right] = \sum_{k=1}^{K} \mu_k \mathbb{E}\left[\sum_{t=1}^{T} \mathbb{1}_{\pi_t=k} \mathbb{1}_{\tau(B,\pi)\geq t-1}\right]$$

$$= \sum_{k=1}^{K_+} \mu_k \mathbb{E}\left[\sum_{t=1}^{T} \mathbb{1}_{\pi_t=k} \mathbb{1}_{\tau(B,\pi)\geq t-1}\right] + o(T). \tag{37}$$

Then, we examine the term $\mathbb{E}\left[\sum_{t=1}^{T} \mathbb{1}_{\pi_t=k} \mathbb{1}_{\tau(B,\pi)\geq t-1}\right]$. In order to analyse it, we will introduce the following events, for any $S, S' \subseteq [K_+]$ such that $S \cap S' = \emptyset$:

$$\Pi_{S,S'} := \left\{ \forall j \in S, \tau\left(\frac{B}{K}, j\right) \geq T; \ \forall j \in S', \sqrt{T} \leq \tau\left(\frac{B}{K}, j\right) < T; \ \forall j \in [K_+] \setminus (S \cup S'), \tau\left(\frac{B}{K}, j\right) < \sqrt{T} \right\}.$$

Please note that $\Pi_S = \Pi_{S,\emptyset}$. We can then decompose, for any $k \in \{1, \dots, K_+\}$,

$$\mathbb{E}\left[\sum_{t=1}^{T} \mathbb{1}_{\pi_t=k} \mathbb{1}_{\tau(B,\pi)\geq t-1}\right] = \sum_{S,S'\subseteq[K_+]:S\cap S'=\emptyset} P\left(\Pi_{S,S'}\right) \mathbb{E}\left[\sum_{t=1}^{T} \mathbb{1}_{\pi_t=k} \mathbb{1}_{\tau(B,\pi)\geq t-1} \middle| \Pi_{S,S'} \right].$$

Consider the case $S' \neq \emptyset$ and let $k \in S'$. We can bound

$$P(\Pi_{S,S'}) \leq P\left(\sqrt{T} \leq \tau\left(\frac{B}{K}, k\right) < T\right)$$

$$= P\left(\tau\left(\frac{B}{K}, k\right) \geq \sqrt{T}\right) - P\left(\tau\left(\frac{B}{K}, k\right) \geq T\right).$$

Indeed, the sequence $\left(P\left(\tau\left(\frac{B}{K},k\right)\geq n\right)\right)_{n\geq 1}$ is increasing and upper-bounded by 1, and thus it has a limit and it implies that

$$P\left(\tau\left(\frac{B}{K},k\right)\geq\sqrt{T}\right)-P\left(\tau\left(\frac{B}{K},k\right)\geq T\right)=o(1).$$

We deduce that, for any $S,S'\subseteq[K_+]$ such that $S\cap S'=\emptyset$,

$$S'\neq\emptyset\implies P\left(\Pi_{S,S'}\right)=o(1).$$

This implies

$$\mathbb{E}\left[\sum_{t=1}^T\mathbb{1}_{\pi_t=k}\mathbb{1}_{\tau(B,\pi)\geq t-1}\right]=\sum_{S\subseteq[K_+]}P\left(\Pi_S\right)\mathbb{E}\left[\sum_{t=1}^T\mathbb{1}_{\pi_t=k}\mathbb{1}_{\tau(B,\pi)\geq t-1}\bigg|\Pi_S\right]+o(T).$$

Re-injecting in (37), we have

$$\mathbb{E}\left[\sum_{t=1}^T X_t^{\pi_t}\mathbb{1}_{\tau(B,\pi)\geq t-1}\right]=\sum_{k=1}^{K_+}\mu_k\sum_{S\subseteq[K_+]}P\left(\Pi_S\right)\mathbb{E}\left[\sum_{t=1}^T\mathbb{1}_{\pi_t=k}\mathbb{1}_{\tau(B,\pi)\geq t-1}\bigg|\Pi_S\right]+o(T)$$

$$=\sum_{S\subseteq[K_+]}P\left(\Pi_S\right)\sum_{k=1}^{K_+}\mu_k\mathbb{E}\left[\sum_{t=1}^T\mathbb{1}_{\pi_t=k}\mathbb{1}_{\tau(B,\pi)\geq t-1}\bigg|\Pi_S\right]+o(T).$$

Besides, it is clear, by definition of $\Pi_S$, that any policy $\pi$ in EXPLOIT satisfies

$$k\notin S\implies\mathbb{E}\left[\sum_{t=1}^T\mathbb{1}_{\pi_t=k}\mathbb{1}_{\tau(B,\pi)\geq t-1}\bigg|\Pi_S\right]=o(T).$$

We deduce that

$$\mathbb{E}\left[\sum_{t=1}^T X_t^{\pi_t}\mathbb{1}_{\tau(B,\pi)\geq t-1}\right]=\sum_{S\subseteq[K_+]}P\left(\Pi_S\right)\sum_{k\in S}\mu_k\mathbb{E}\left[\sum_{t=1}^T\mathbb{1}_{\pi_t=k}\mathbb{1}_{\tau(B,\pi)\geq t-1}\bigg|\Pi_S\right]+o(T)$$

$$=\sum_{S\subseteq[K_+]}P\left(\Pi_S\right)\sum_{k\in S}\mu_k n_k(S)T+o(T)$$

$$=\sum_{k=1}^{K_+}\left(\sum_{S\subseteq[K_+]:k\in S}P\left(\Pi_S\right)n_k(S)\right)\mu_k T+o(T),$$

which concludes the proof of the lemma. $\qquad\square$

### F.2  Proof of Proposition 3

It remains to provide an upper bound to the right-hand side of the equality (36) in Lemma 8 in order to complete the proof of Proposition 3. In order to maximize the right term in (36), we should solve the following maximization problem:

$$\max\sum_{k=1}^{K_+}\left(\sum_{S\subseteq[K_+]:k\in S}P\left(\Pi_S\right)n_k(S)\right)\mu_k\quad\text{s.t.}\quad\sum_{k\in S}n_k(S)\leq 1\text{ and }\forall k\notin S,n_k(S)=0.$$

We can re-order the terms in the above sum as

$$\sum_{k=1}^{K_+}\left(\sum_{S\subseteq[K_+]:k\in S}P\left(\Pi_S\right)n_k(S)\right)\mu_k=\sum_{S\subseteq[K_+]}P\left(\Pi_S\right)\sum_{k\in S}\mu_k n_k(S),$$

and since, by hypothesis, $\mu_1 \geq \mu_2 \geq \cdots \geq \mu_{K_+} > 0$, we deduce the bound

$$\sum_{k=1}^{K_+} \left( \sum_{S \subseteq [K_+]: k \in S} P\left(\Pi_S\right) n_k(S) \right) \mu_k \leq \sum_{S \subseteq [K_+]} P\left(\Pi_S\right) \max_{k \in S} \mu_k,$$

where the above inequality is an equality for $n_k(S) = n_k^*(S)$, defined by

$$n_k^*(S) = \left\{ \begin{array}{ll} 1 & \text{if } k = \min S \\ 0 & \text{otherwise.} \end{array} \right.$$

This gives, for any given $k \in \{1, \ldots, K_+\}$,

$$\sum_{S \subseteq [K_+]: k \in S} P\left(\Pi_S\right) n_k^*(S)$$

$$= \sum_{S \subseteq \{k+1, \ldots, K_+\}} P\left(\Pi_{\{k\} \cup S}\right)$$

$$= \sum_{S \subseteq \{k+1, \ldots, K_+\}} P\left(\forall j \in \{k\} \cup S, \tau\left(\frac{B}{K}, j\right) \geq T;\ \forall j \in [K_+] \setminus (\{k\} \cup S), \tau\left(\frac{B}{K}, j\right) < \sqrt{T}\right)$$

$$= P\left(\forall j \in [k-1], \tau\left(\frac{B}{K}, j\right) < \sqrt{T};\ \tau\left(\frac{B}{K}, k\right) \geq T\right) + o(1)$$

$$= \left(1 - p^{\text{EX}}\right) \underbrace{\frac{1}{1 - p^{\text{EX}}} \prod_{j=1}^{k-1} P\left(\tau\left(\frac{B}{K}, j\right) < \infty\right) P\left(\tau\left(\frac{B}{K}, k\right) = \infty\right)}_{w_k} + o(1).$$

We deduce that

$$\mathbb{E}\left[ \sum_{t=1}^T X_t^{\pi_t} \mathbb{1}_{\tau(B, \pi) \geq t-1} \right] \leq \left(1 - p^{\text{EX}}\right) \sum_{k=1}^{K_+} w_k \mu_k T + o(T),$$

which concludes the proof of the proposition. ∎

## G  Proof of the Reward Bound of EXPLOIT-UCB (Proposition 4)

In this appendix and in the next one, we will use the following notation. For any $k \in [K]$, let $Y_1^k, \ldots, Y_T^k \sim F_k$ be i.i.d. rewards drawn from arm $k$ and for any $t \geq 1$, and we denote by $N_k(t) := \sum_{s=1}^t \mathbb{1}_{\pi_s = k}$ the number of times arm $k$ has been pulled until round $t$. Please note that $X_t^{\pi_t} = \sum_{k=1}^K Y_{N_k(t)}^k \mathbb{1}_{\pi_t = k}$. Given that arm $k$ has been pulled $n_k$ times, we also introduce $\hat{Y}_{n_k}^k := \frac{1}{n_k} \sum_{n=1}^{n_k} Y_n^k$ as the empirical average of arm $k$ at round $t$. Please note that for any $k \in [K]$ and any round $t$,

$$\hat{Y}_{N_k(t)}^k = \hat{X}_t^k.$$

For any $t \geq 1$, let

$$C_k(t) := \hat{Y}_{N_k(t)}^k + \sqrt{\frac{6 \log(t)}{N_k(t)}}.$$

### G.1  Preliminary Lemma

EXPLOIT-UCB is based on the classic bandit algorithm UCB1 (which has a sublinear regret in the classic stochastic MAB), and therefore has the following characteristic which is going to be useful in the proof of the reward bound of both EXPLOIT-UCB and EXPLOIT-UCB-DOUBLE. For the sake of clarity, we assume the arms are ordered in decreasing expectation: $\mu_1 \geq \mu_2 \geq \cdots \geq \mu_K$.

**Lemma 9.** *Let $(\pi_t)_{t \geq 1}$ be the policy associated to EXPLOIT-UCB. Assume that $\mu_1 > \mu_k$. Then*

$$\mathbb{E}\left[\sum_{t=1}^{T} \mathbb{1}_{\pi_t=k, C_k(t) \geq C_1(t)}\right] = o(T).$$

*Proof.* This proof completely follows the proof of Theorem 1 in Auer et al. (2002). Let $r \leq T$, the quantity to bound can be written as

$$\mathbb{E}\left[\sum_{t=1}^{T} \mathbb{1}_{\pi_t=k, C_k(t) \geq C_1(t)}\right] = \mathbb{E}\left[\sum_{t=1}^{T} \mathbb{1}_{\pi_t=k, \ \hat{Y}_{N_k(t)}^{k}(t) + \sqrt{\frac{6 \log t}{N_k(t)}} \geq \hat{Y}_{N_1(t)}^{1}(t) + \sqrt{\frac{6 \log t}{N_1(t)}}}\right]$$

$$= \mathbb{E}\left[\sum_{t=T^{1/2}}^{T} \mathbb{1}_{\pi_t=k, \ \hat{Y}_{N_k(t)}^{k}(t) + \sqrt{\frac{6 \log t}{N_k(t)}} \geq \hat{Y}_{N_1(t)}^{1}(t) + \sqrt{\frac{6 \log t}{N_1(t)}}}\right] + o(T)$$

$$\leq r + \mathbb{E}\left[\sum_{t \in \Delta_{T,r}} \mathbb{1}_{\pi_t=k, \ \max_{r \leq n_k < t} \hat{Y}_{n_k}^{k} + \sqrt{\frac{6 \log t}{n_k}} \geq \min_{1 \leq n_1 < t} \hat{Y}_{n_1}^{1} + \sqrt{\frac{6 \log t}{n_1}}}\right] + o(T)$$

$$\leq r + \mathbb{E}\left[\sum_{t \in \Delta_{T,r}} \mathbb{1}_{\max_{r \leq n_k < t} \hat{Y}_{n_k}^{k} + \sqrt{\frac{6 \log t}{n_k}} \geq \min_{1 \leq n_1 < t} \hat{Y}_{n_1}^{1} + \sqrt{\frac{6 \log t}{n_1}}}\right] + o(T)$$

$$\leq r + \mathbb{E}\left[\sum_{t \in \Delta_{T,r}} \sum_{n_1=1}^{t-1} \sum_{n_k=r}^{t-1} \mathbb{1}_{\hat{Y}_{n_k}^{k} + \sqrt{\frac{6 \log t}{n_k}} \geq \hat{Y}_{n_1}^{1} + \sqrt{\frac{6 \log t}{n_1}}}\right] + o(T),$$

where $\Delta_{T,r} = \left\{t \in \{1, \ldots, T\} : \sum_{s=1}^{t} \mathbb{1}_{\pi_s=k} \geq r\right\}$. Similarly as in Auer et al. (2002), we use the fact that the probability event $\left\{\hat{Y}_{n_k}^{k} + \sqrt{\frac{6 \log t}{n_k}} \geq \hat{Y}_{n_1}^{1} + \sqrt{\frac{6 \log t}{n_1}}\right\}$ implies at least one of the following:

$$\hat{Y}_{n_1}^{1} \leq \mu_1 - \sqrt{\frac{6 \log t}{n_1}}$$

$$\hat{Y}_{n_k}^{k} \geq \mu_k + \sqrt{\frac{6 \log t}{n_k}}$$

$$\mu_1 < \mu_k + 2\sqrt{\frac{6 \log t}{n_k}}.$$

Therefore, we can write

$$\mathbb{E}\left[\sum_{t=1}^{T} \mathbb{1}_{\pi_t=k, C_k(t) \geq C_1(t)}\right]$$

$$\leq r + \mathbb{E}\left[\sum_{t \in \Delta_{T,r}} \sum_{n_1=1}^{t-1} \sum_{n_k=r}^{t-1} \mathbb{1}_{\hat{Y}_{n_k}^{k} + \sqrt{\frac{6 \log t}{n_k}} \geq \hat{Y}_{n_1}^{1} + \sqrt{\frac{6 \log t}{n_1}}}\right] + o(T)$$

$$\leq r + \mathbb{E}\left[\sum_{t \in \Delta_{T,r}} \sum_{n_1=1}^{t-1} \sum_{n_k=r}^{t-1} \left(\mathbb{1}_{\hat{Y}_{n_1}^{1} \leq \mu_1 - \sqrt{\frac{6 \log T}{n_1}}} + \mathbb{1}_{\hat{Y}_{n_k}^{k} \geq \mu_k + \sqrt{\frac{6 \log T}{n_k}}} + \mathbb{1}_{\mu_1 < \mu_k + 2\sqrt{\frac{6 \log T}{n_k}}}\right)\right] + o(T).$$

The choice $r = \left\lceil \frac{24 \log T}{(\mu_1-\mu_k)^2} \right\rceil$ ensures that, for any $n_k \geq r$,

$$\mu_1 - \mu_k - 2\sqrt{\frac{6 \log T}{n_k}} \geq 0,$$

which implies

$$
\mathbb{E}\left[\sum_{t=1}^{T} \mathbb{1}_{\pi_t=k, C_k(t) \geq C_1(t)}\right]
$$

$$
\leq \left\lceil \frac{24 \log T}{(\mu_1 - \mu_k)^2} \right\rceil + \mathbb{E}\left[\sum_{t \in \Delta_{T,r}} \sum_{n_1=1}^{t-1} \sum_{n_k=r}^{t-1} \left(\mathbb{1}_{\hat{Y}_{n_1}^1 \leq \mu_1 - \sqrt{\frac{6 \log T}{n_1}}} + \mathbb{1}_{\hat{Y}_{n_k}^k \geq \mu_k + \sqrt{\frac{6 \log T}{n_k}}}\right)\right] + o(T)
$$

$$
= T \times \sum_{n_1=1}^{T} \sum_{n_k=r}^{T} \mathbb{E}\left[\left(\mathbb{1}_{\hat{Y}_{n_1}^1 \leq \mu_1 - \sqrt{\frac{6 \log T}{n_1}}} + \mathbb{1}_{\hat{Y}_{n_k}^k \geq \mu_k + \sqrt{\frac{6 \log T}{n_k}}}\right)\right] + o(T)
$$

$$
= T \times \sum_{n_1=1}^{T} \sum_{n_k=r}^{T} \left(P\left(\hat{Y}_{n_1}^1 \leq \mu_1 - \sqrt{\frac{6 \log T}{n_1}}\right) + P\left(\hat{Y}_{n_k}^k \geq \mu_k + \sqrt{\frac{6 \log T}{n_k}}\right)\right) + o(T). \tag{38}
$$

Using Hoeffding's inequality, for any $n_1 \in \{1, \dots, T\}$, we have

$$
P\left(\hat{Y}_{n_1}^1 \leq \mu_1 - \sqrt{\frac{6 \log T}{n_1}}\right) \leq \frac{1}{T^3}.
$$

Similarly, for any $n_k \in \{r, \dots, T\}$, we have

$$
P\left(\hat{Y}_{n_k}^k \geq \mu_k + \sqrt{\frac{6 \log T}{n_k}}\right) \leq \frac{1}{T^3}.
$$

We can then replace in (38):

$$
\mathbb{E}\left[\sum_{t=1}^{T} \mathbb{1}_{\pi_t=k, C_k(t) \geq C_1(t)}\right] \leq T \sum_{n_1=1}^{T} \sum_{n_k=r}^{T} \frac{2}{T^3} + o(T)
$$

$$
\leq 2 + o(T)
$$

$$
= o(T),
$$

which concludes the proof of the lemma. □

## G.2 Proof of Proposition 4

Let $S \subseteq [K_+]$. Recall that for any arm $k \in [K_+]$, $n_k(S)$ is defined as

$$
n_k(S) = \frac{1}{T}\mathbb{E}\left[\sum_{t=1}^{T} \mathbb{1}_{\pi_t=k, \tau(B,\pi) \geq t-1} \,\bigg|\, \Pi_S\right],
$$

and Lemma 8 states that

$$
\mathbb{E}\left[\sum_{t=1}^{T} X_t^{\pi_t} \mathbb{1}_{\tau(B,\pi) \geq t-1}\right] = \sum_{k=1}^{K_+} \left(\sum_{S \subseteq [K_+]:k \in S} P(\Pi_S) n_k(S)\right) \mu_k \times T + o(T).
$$

Denoting by $S_0$ the set of positive or zero arms,

$$
S_0 := \left\{j \in [K_+] : P\left(\tau\left(\frac{B}{K}, j\right) = \infty\right) = 1\right\},
$$

it is clear by definition of $\Pi_S$ that

$$
S_0 \not\subseteq S \implies P(\Pi_s) = 0,
$$

and hence,

$$\mathbb{E}\left[\sum_{t=1}^{T} X_t^{\pi_t} \mathbb{1}_{\tau(B,\pi)\geq t-1}\right] = \sum_{k=1}^{K_+}\left(\sum_{\substack{S\subseteq[K_+]:\\ S_0\subseteq S,\ k\in S}} P(\Pi_S)n_k(S)\right)\mu_k \times T + o(T).$$

Given that EXPLOIT-UCB is in EXPLOIT, by Lemma 8, it suffices to show that for any $S_0 \subseteq S \subseteq [K_+]$ and any $k \in S$,

$$n_k(S) = \begin{cases} 1 + o(1) & \text{if } k = \min S \\ o(1) & \text{otherwise.} \end{cases}$$

Then, please note that $\sum_{k=1}^{K} n_k(S) = 1$. Therefore, it suffices to prove that for any $k \in S \setminus \{\min S\}, n_k(S) = o(1)$. Thus, let $k \in S \setminus \{\min S\}$. Then, on the one hand, we are going to provide a lower bound $P(\Pi_S)$ which is independent of $T$. Indeed, since $S_0 \subseteq S$, there is no positive or zero arm $k \in [K_+] \setminus S$, and therefore, we know that there exists $\epsilon > 0$ such that

$$\forall k \in [K_+], P_{X\sim F_k}(X \leq -\epsilon) > 0.$$

We fix such an $\epsilon$ and we deduce that

$$\prod_{k\in[K_+]\setminus S} P\left(\tau\left(\frac{B}{K},k\right) \leq \frac{B}{\epsilon K}\right) > 0.$$

We can therefore provide the following lower bound, independent of $T$ and positive by definition of $K_+$. For any $T \geq \left(\frac{B}{\epsilon K}\right)^2$,

$$P(\Pi_S) = \prod_{k\in S} P\left(\tau\left(\frac{B}{K},k\right) \geq T\right) \prod_{k\in[K_+]\setminus S} P\left(\tau\left(\frac{B}{K},k\right) < \sqrt{T}\right)$$

$$\geq \prod_{k\in S} P\left(\tau\left(\frac{B}{K},k\right) = \infty\right) \prod_{k\in[K_+]\setminus S} P\left(\tau\left(\frac{B}{K},k\right) < \frac{B}{\epsilon K}\right)$$

$$> 0.$$

On the other hand, an upper bound to $\mathbb{E}\left[\sum_{t=1}^{T} \mathbb{1}_{\pi_t=k,\Pi_S}\right]$ is obtained by

$$\mathbb{E}\left[\sum_{t=1}^{T} \mathbb{1}_{\pi_t=k,\Pi_S}\right] \leq \mathbb{E}\left[\sum_{t=1}^{T} \mathbb{1}_{\pi_t=k,\forall j\in S,\tau\left(\frac{B}{K},j\right)\geq T}\right]$$

$$\leq \mathbb{E}\left[\sum_{t=1}^{T} \mathbb{1}_{\pi_t=k,C_k(t-1)\geq C_{\min S}(t-1)}\right]$$

$$= o(T)$$

by Lemma 9. We deduce the following bound on $n_k(S)$:

$$n_k(S) = \frac{1}{T}\frac{\mathbb{E}\left[\sum_{t=1}^{T} \mathbb{1}_{\pi_t=k,\Pi_S}\right]}{P(\Pi_S)}$$

$$\leq \frac{1}{T}\frac{\mathbb{E}\left[\sum_{t=1}^{T} \mathbb{1}_{\pi_t=k,C_k(t-1)\geq C_{\min S}(t-1)}\right]}{\prod_{k\in S} P\left(\tau\left(\frac{B}{K},k\right) = \infty\right) \prod_{k\in[K_+]\setminus S} P\left(\tau\left(\frac{B}{K},k\right) < \frac{B}{\epsilon K}\right)}$$

$$= o(1),$$

which concludes the proof of the proposition. ∎

# H   The Performance of EXPLOIT-UCB-DOUBLE

In the proofs of the performance of EXPLOIT-UCB-DOUBLE, for the sake of clarity, we drop the exponent $n$ in the notation of EXPLOIT-UCB-DOUBLE, and $\pi^n$ becomes $\pi$. Recall the following notation from the previous section, for any $t \geq 1$,

$$C_k(t) := \hat{Y}^k_{N_k(t)} + \sqrt{\frac{6\log(t)}{N_k(t)}}.$$

## H.1   Proof of Proposition 5

For any policy $\pi$ and any budget $B'$, we will denote $\pi \in \text{EXPLOIT}(B')$ if at round $t$, $\pi$ only pulls arms $k \in [K]$ such that $\sum_{s=1}^{t} X_s^{\pi_s} \mathbb{1}_{\pi_s=k} \geq -\frac{B'}{K} + 1$.

The probability of ruin of EXPLOIT-UCB-DOUBLE can be decomposed as

$$P(\tau(B,\pi) < T) = \sum_{j=0}^{\infty} P(\tau(B,\pi) < T \cap t_j \leq \tau(B,\pi) < t_{j+1}). \tag{39}$$

Let us first examine the term in $j = 0$. Then,

$$P(\tau(B,\pi) < T \cap \tau(B,\pi) < t_1) \leq P\left(\tau(B,\pi) < T \text{ and } \forall t \leq T, B + \sum_{s=1}^{t} X_s^{\pi_s} \mathbb{1}_{\tau(B,\pi) \geq s-1} < nB^2\right)$$

$$\leq P\left(\tau(B,\pi) < T \text{ and } \pi \in \text{EXPLOIT}(B)\right)$$

$$\leq p^{\text{EX}}.$$

Then, let us examine the other terms in the sum. Let $j \geq 1$. For any $t \geq 1$, we will denote $\tilde{\pi}_t := \pi_{t_j+t}$. Let us re-write each of the terms in the sum as

$$P\left(\tau(B,\pi) < T \text{ and } t_j \leq \tau(B,\pi) < t_{j+1}\right) = P\left(t_j \leq \tau(B,\pi) < t_{j+1};\ B + \sum_{t=1}^{T} X_t^{\pi_t} \mathbb{1}_{\tau(B,\pi) \geq t-1} < 0\right).$$

This is re-written as

$$P\left(\tau(B,\pi) < T \text{ and } t_j \leq \tau(B,\pi) < t_{j+1}\right)$$

$$= P\left(t_j \leq \tau(B,\pi) < t_{j+1};\ B + \sum_{t=1}^{T} X_t^{\pi_t} \mathbb{1}_{\forall s \leq t-1, B + \sum_{r=1}^{s} X_r^{\pi_r} > 0} < 0\right).$$

But then, by definition of $t_j$, under the condition that $t_j < T$, we have that

$$B + \sum_{t=1}^{t_j} X_t^{\pi_t} \geq jnB^2,$$

which implies that, for any $t \geq t_j + 1$,

$$B + \sum_{s=1}^{t} X_s^{\pi_s} \geq jnB^2 + \sum_{s=t_j+1}^{t} X_s^{\pi_s}.$$

We can then replace in the previous equation:

$$P\left(\tau(B,\pi) < T \text{ and } t_j \leq \tau(B,\pi) < t_{j+1}\right) \leq$$

$$P\left(t_j \leq \tau(B,\pi) < t_{j+1};\ jnB^2 + \sum_{t=t_j+1}^{T} X_t^{\pi_t} \mathbb{1}_{\forall s \leq t-1, jnB^2 + \sum_{r=t_j+1}^{s} X_r^{\pi_r} > 0} < 0\right).$$

This is re-written as

$$P\left(\tau(B,\pi) < T \text{ and } t_j \leq \tau(B,\pi) < t_{j+1}\right) \leq$$

$$P\left(t_j \leq \tau(B,\pi) < t_{j+1}; \; jnB^2 + \sum_{t=1}^{T-t_j} X_{t+t_j}^{\tilde{\pi}_t} \mathbb{1}_{\forall s \leq t-1, jnB^2 + \sum_{r=1}^s X_{r+t_j}^{\tilde{\pi}_{t_j}} > 0} < 0\right),$$

and then

$$P\left(\tau(B,\pi) < T \text{ and } t_j \leq \tau(B,\pi) < t_{j+1}\right) \leq P\left(t_j \leq \tau(B,\pi) < t_{j+1}; \; \tau(B,\tilde{\pi}) < T - t_j\right)$$
$$\leq P\left(\tau(B,\tilde{\pi}) < \infty, \tilde{\pi} \in \text{ EXPLOIT } (jnB^2)\right)$$
$$\leq \left(p^{\text{EX}}\right)^{jnB}.$$

We can then replace in (39):

$$P(\tau(B,\pi) < T) = \sum_{j=0}^{\infty} P(\tau(B,\pi) < T \cap t_j \leq \tau(B,\pi) < t_{j+1})$$

$$\leq p^{\text{EX}} + \sum_{j=1}^{\infty} \left(p^{\text{EX}}\right)^{jnB}$$

$$= p^{\text{EX}} + \frac{\left(p^{\text{EX}}\right)^{nB}}{1 - \left(p^{\text{EX}}\right)^{nB}},$$

which gives the desired result. Let $\epsilon > 0$, then

$$n \geq \frac{\log \frac{\epsilon}{1+\epsilon}}{B \log p^{\text{EX}}} \implies \frac{\left(p^{\text{EX}}\right)^{nB}}{1 - \left(p^{\text{EX}}\right)^{nB}} \leq \epsilon \leq \frac{\epsilon}{\mu^*},$$

hence

$$P(\tau(B,\pi) < \infty) \leq p^{\text{EX}} + \frac{\epsilon}{\mu^*},$$

which concludes the proof of the proposition. ∎

### H.2 Proof of Proposition 6

We will assume that the arm with the biggest expectation is arm 1 and that it is unique for the sake of clarity. Let $j := \lceil \frac{T^{1/4}}{nB^2} - 1 \rceil$, recall that

$$t_j = \inf\left\{t \in \{0, \dots, \min(\tau(B,\pi), T)\} : B + \sum_{s=1}^t X_s^{\pi_s} > jnB^2\right\}.$$

We still denote $\tilde{\pi}_t := \pi_{t_j + t}$ for any $t \geq 1$. We can then decompose the reward as follows:

$$\mathbb{E}\left[\sum_{t=1}^T X_t^{\pi_t} \mathbb{1}_{\tau(B,\pi) \geq t-1}\right] = \sum_{k=1}^K \mathbb{E}\left[\sum_{t=1}^T X_t^{\pi_t} \mathbb{1}_{\pi_t=k} \mathbb{1}_{\tau(B,\pi) \geq t-1}\right]$$

$$= \mu_1 \mathbb{E}\left[\sum_{t=1}^T \mathbb{1}_{\pi_t=1} \mathbb{1}_{\tau(B,\pi) \geq t-1}\right] + \sum_{k=2}^K \mu_k \mathbb{E}\left[\sum_{t=1}^T \mathbb{1}_{\pi_t=k} \mathbb{1}_{\tau(B,\pi) \geq t-1}\right]$$

$$= \mu_1 \underbrace{\mathbb{E}\left[\sum_{t=1}^T \mathbb{1}_{\pi_t=1} \mathbb{1}_{\tau(B,\pi) \geq t-1}\right]}_{(A)} + \sum_{k=2}^K \mu_k \underbrace{\mathbb{E}\left[\sum_{t=t_j+1}^T \mathbb{1}_{\pi_t=k} \mathbb{1}_{\tau(B,\pi) \geq t-1}\right]}_{(B_k)} + O\left(\mathbb{E}[t_j]\right).$$

If we prove that

$$(A) = P(\tau(B,\pi) = \infty)T + o(T), \quad \forall k \in \{2,\ldots,K\}, (B_k) = o(T), \quad \mathbb{E}[t_j] = o(T),$$

then, we can write that

$$\mathbb{E}\left[\sum_{t=1}^{T} X_t^{\pi_t} \mathbb{1}_{\tau(B,\pi) \geq t-1}\right] = \mu_1 P(\tau(B,\pi) = \infty)T + o(T),$$

and using Proposition 5 gives the result:

$$\mathbb{E}\left[\sum_{t=1}^{T} X_t^{\pi_t} \mathbb{1}_{\tau(B,\pi) \geq t-1}\right] \geq \mu_1 \left(1 - p^{\mathrm{EX}} - \frac{\left(p^{\mathrm{EX}}\right)^{nB}}{1 - \left(p^{\mathrm{EX}}\right)^{nB}}\right) T + o(T).$$

Let $\epsilon > 0$. Then by Proposition 5,

$$n \geq \frac{\log \frac{\epsilon}{1+\epsilon}}{B \log p^{\mathrm{EX}}} \implies P(\tau(B,\pi) = \infty) \geq 1 - p^{\mathrm{EX}} - \epsilon.$$

In particular, the choice of $\epsilon = \frac{T^{B \log p^{\mathrm{EX}}}}{1 - T^{B \log p^{\mathrm{EX}}}} = o_T(1)$ gives $\frac{\log \frac{\epsilon}{1+\epsilon}}{B \log p^{\mathrm{EX}}} = \log T$, and hence

$$n \geq \log T \implies \mathbb{E}\left[\sum_{t=1}^{T} X_t^{\pi_t} \mathbb{1}_{\tau(B,\pi) \geq t-1}\right] \geq \mu_1 (1 - p^{\mathrm{EX}})T + o(T),$$

which concludes the proof. It only remains to study each of the terms $(A), (B_k)$ for $k \geq 2$ and $\mathbb{E}[t_j]$.

**Study of** $(B_k)$

Let $k \in \{2,\ldots,K\}$. We can decompose the term $(B_k)$ as follows:

$$(B_k) = \mathbb{E}\left[\sum_{t=t_j+1}^{T} \mathbb{1}_{\pi_t=k} \mathbb{1}_{\tau(B,\pi) \geq t-1}\right]$$

$$= \mathbb{E}\left[\sum_{t=t_j+1}^{T} \mathbb{1}_{\pi_t=k} \mathbb{1}_{\tau(B,\pi) \geq t_j} \mathbb{1}_{\forall s \leq t-1, B+\sum_{r=1}^{s} X_r^{\pi_r} > 0}\right]$$

$$= \mathbb{E}\left[\sum_{t=t_j+1}^{T} \mathbb{1}_{\pi_t=k} \mathbb{1}_{\tau(B,\pi) \geq t_j} \mathbb{1}_{\forall s \in \{t_j+1,\ldots,t-1\}, B+\sum_{r=1}^{t_j} X_r^{\pi_r} + \sum_{r=t_j+1}^{s} X_r^{\pi_r} > 0}\right].$$

But then, by definition of $t_j$, if $t_j \leq \min(\tau(B,\pi), T)$,

$$B + \sum_{r=1}^{t_j} X_r^{\pi_r} < jnB^2 + 1 = T^{1/4} + 1.$$

This implies that, if $t_j \leq \min(\tau(B,\pi), T)$, denoting $\tilde{\pi}_s := \pi_{s+t_j}$ for any $s \geq 1$,

$$\mathbb{1}_{\forall s \in \{t_j+1,\ldots,t-1\}, B+\sum_{r=1}^{t_j} X_r^{\pi_r} + \sum_{r=t_j+1}^{s} X_r^{\pi_r} > 0} \leq \mathbb{1}_{\forall s \in \{t_j+1,\ldots,t-1\}, T^{1/4}+1+\sum_{r=t_j+1}^{s} X_r^{\pi_r} > 0}$$

$$= \mathbb{1}_{\forall s \in \{1,\ldots,t-t_j-1\}, T^{1/4}+1+\sum_{r=1}^{s-t_j} X_{r+t_j}^{\tilde{\pi}_r} > 0}$$

$$= \mathbb{1}_{\tau(T^{1/4}+1,\tilde{\pi}) \geq t-t_j-1},$$

and thus, if $t_j \leq \min(\tau(B,\pi), T)$,

$$(B_k) = \mathbb{E}\left[\sum_{t=t_j+1}^{T} \mathbb{1}_{\pi_t=k}\mathbb{1}_{\tau(B,\pi)\geq t_j}\mathbb{1}_{\forall s \in \{t_j+1,\dots,t-1\}, B+\sum_{r=1}^{t_j} X_r^{\pi_r}+\sum_{r=t_j+1}^{s} X_r^{\pi_r}>0}\right]$$

$$\leq \mathbb{E}\left[\sum_{t=t_j+1}^{T} \mathbb{1}_{\pi_t=k}\mathbb{1}_{\tau(B,\pi)\geq t_j}\mathbb{1}_{\tau(T^{1/4}+1,\tilde{\pi})\geq t-t_j-1}\right]$$

$$= \mathbb{E}\left[\sum_{t=1}^{T-t_j} \mathbb{1}_{\pi_{t+t_j}=k}\mathbb{1}_{\tau(B,\pi)\geq t_j}\mathbb{1}_{\tau(T^{1/4}+1,\tilde{\pi})\geq t-1}\right].$$

This inequality being a trivial equality if $t_j = T+1$ (because the sum is a sum on an empty set) or if $t_j = \tau(B,\pi)+1$ $(0 \leq 0)$, we deduce that in any case,

$$(B_k) \leq \mathbb{E}\left[\sum_{t=1}^{T-t_j} \mathbb{1}_{\pi_{t+t_j}=k}\mathbb{1}_{\tau(B,\pi)\geq t_j}\mathbb{1}_{\tau(T^{1/4}+1,\tilde{\pi})\geq t-1}\right].$$

Then, denoting by $s_j$ the realized value of $t_j$ and decomposing classically, we have

$$(B_k) \leq \mathbb{E}\left[\sum_{t=1}^{T-t_j} \mathbb{1}_{\pi_{t+t_j}=k}\mathbb{1}_{\tau(B,\pi)\geq t_j}\mathbb{1}_{\tau(T^{1/4}+1,\tilde{\pi})\geq t-1}\right]$$

$$= \sum_{s_j=T^{1/4}}^{T} \mathbb{E}\left[\sum_{t=1}^{T-s_j} \mathbb{1}_{\pi_{t+s_j}=k}\mathbb{1}_{\tau(B,\pi)\geq s_j}\mathbb{1}_{\tau(T^{1/4}+1,\tilde{\pi})\geq t-1}\mathbb{1}_{t_j=s_j}\right]$$

$$= \sum_{s_j=T^{1/4}}^{T}\sum_{t=1}^{T-s_j} \mathbb{E}\left[\mathbb{1}_{\pi_{t+s_j}=k}\mathbb{1}_{\tau(B,\pi)\geq s_j}\mathbb{1}_{\tau(T^{1/4}+1,\tilde{\pi})\geq t-1}\mathbb{1}_{t_j=s_j}\right]$$

$$= \sum_{s_j=T^{1/4}}^{T}\sum_{t=1}^{T-s_j} P\left(\tau(B,\pi)\geq s_j, \tau(T^{1/4}+1,\tilde{\pi})\geq t-1, t_j=s_j, \pi_{t+s_j}=k\right).$$

Let us then study the probability $P\left(\tau(B,\pi)\geq s_j, \tau(T^{1/4}+1,\tilde{\pi})\geq t-1, t_j=s_j, \pi_{t+s_j}=k\right)$ and decompose it as

$$P\left(\tau(B,\pi)\geq s_j, \tau(T^{1/4}+1,\tilde{\pi})\geq t-1, t_j=s_j, \pi_{t+s_j}=k\right) =$$

$$P\left(\tau(B,\pi)\geq s_j, \tau(T^{1/4}+1,\tilde{\pi})\geq t-1, t_j=s_j, \pi_{t+s_j}=k, \forall t\geq s_j, \sum_{s=1}^{t} X_s^1 \mathbb{1}_{\pi_s=1} > -\frac{T^{1/4}+1}{K}\right)$$

$$+ P\left(\tau(B,\pi)\geq s_j, \tau(T^{1/4}+1,\tilde{\pi})\geq t-1, t_j=s_j, \pi_{t+s_j}=k, \exists t\geq s_j, \sum_{s=1}^{t} X_s^1 \mathbb{1}_{\pi_s=1} \leq -\frac{T^{1/4}+1}{K}\right). \quad (40)$$

The second term on the right, though, can easily be bounded as follows:

$$P\left(\tau(B,\pi)\geq s_j, \tau(T^{1/4}+1,\tilde{\pi})\geq t-1, t_j=s_j, \pi_{t+s_j}=k, \exists t\geq s_j, \sum_{s=1}^{t} X_s^1 \mathbb{1}_{\pi_s=1} \leq -\frac{T^{1/4}+1}{K}\right)$$

$$\leq P\left(\exists t\geq T^{1/4}, \sum_{s=1}^{t} X_s^1 \mathbb{1}_{\pi_s=1} \leq -\frac{T^{1/4}+1}{K}\right).$$

Then, let us denote by $\delta_1^\pi < \delta_2^\pi < ...$ the rounds $t \geq T^{1/4}$ at which $\pi_t = 1$, in other words, denoting $\delta_0^\pi := T^{1/4}$,

$$\forall j \geq 1, \delta_j^\pi := \inf\{t \geq \delta_{j-1}^\pi : \pi_t = 1\}.$$

Then, we can bound the probability as

$$P\left(\exists t \geq T^{1/4}, \sum_{s=1}^t X_s^1 \mathbb{1}_{\pi_s=1} \leq -\frac{T^{1/4}}{K}\right) = P\left(\exists j \geq 0, \sum_{s=1}^{\delta_j^\pi} X_s^1 \mathbb{1}_{\pi_s=1} \leq -\frac{T^{1/4}+1}{K}\right)$$

$$= \sum_{n_1=\frac{T^{1/4}}{K}}^{\infty} P\left(\exists j \geq 0, \sum_{s=1}^{\delta_j^\pi} X_s^1 \mathbb{1}_{\pi_s=1} \leq -\frac{T^{1/4}+1}{K}, \sum_{s=1}^{T^{1/4}} \mathbb{1}_{\pi_s=1} = n_1\right)$$

$$\leq \sum_{n_1=\frac{T^{1/4}}{K}}^{\infty} \sum_{j=0}^{\infty} P\left(\sum_{s=1}^{\delta_j^\pi} X_s^1 \mathbb{1}_{\pi_s=1} \leq -\frac{T^{1/4}+1}{K}, \sum_{s=1}^{T^{1/4}} \mathbb{1}_{\pi_s=1} = n_1\right),$$

because the rewards are bounded in $[-1, 1]$. Then, using Hoeffding's inequality, for any $n_1 \geq \frac{T^{1/4}}{K}$ and $j \geq 0$,

$$P\left(\sum_{s=1}^{\delta_j^\pi} X_s^1 \mathbb{1}_{\pi_s=1} \leq -\frac{T^{1/4}+1}{K}, \sum_{s=1}^{T^{1/4}} \mathbb{1}_{\pi_s=1} = n_1\right) \leq P\left(\frac{\sum_{s=1}^{\delta_j^\pi} X_s^1 \mathbb{1}_{\pi_s=1}}{\sum_{s=1}^{\delta_j^\pi} \mathbb{1}_{\pi_s=1}} - \mu_1 \leq -\mu_1, \sum_{s=1}^{T^{1/4}} \mathbb{1}_{\pi_s=1} = n_1\right)$$

$$\leq \exp\left(-\frac{(n_1+j)\mu_1^2}{2}\right).$$

Summing over $n_1$ and $j$ gives

$$\sum_{n_1=T^{1/4}/K}^{\infty} \sum_{j=0}^{\infty} P\left(\sum_{s=1}^{\delta_j^\pi} X_s^1 \mathbb{1}_{\pi_s=1} \leq -\frac{T^{1/4}+1}{K}, \sum_{s=1}^{T^{1/4}} \mathbb{1}_{\pi_s=1} = n_1\right) \leq \sum_{n_1=T^{1/4}/K}^{\infty} \sum_{j=0}^{\infty} \exp\left(-\frac{(n_1+j)\mu_1^2}{2}\right)$$

$$= \frac{\exp\left(-\frac{T^{1/4}\mu_1^2}{2K}\right)}{\left(1-e^{-\frac{\mu_1^2}{2}}\right)^2}.$$

We then deduce that

$$P\left(\tau(B,\pi) \geq s_j, \tau(T^{1/4}+1, \tilde\pi) \geq t-1, t_j = s_j, \pi_{t+s_j} = k, \exists t \geq s_j, \sum_{s=1}^t X_s^1 \mathbb{1}_{\pi_s=1} \leq -\frac{T^{1/4}+1}{K}\right)$$

$$= O\left(\exp\left(-\frac{T^{1/4}\mu_1^2}{2K}\right)\right),$$

and thus, plugging it in (40),

$$P\left(\tau(B,\pi) \geq s_j, \tau(T^{1/4}+1, \tilde\pi) \geq t-1, t_j = s_j, \pi_{t+s_j} = k\right) =$$

$$P\left(\tau(B,\pi) \geq s_j, \tau(T^{1/4}+1, \tilde\pi) \geq t-1, t_j = s_j, \pi_{t+s_j} = k, \forall t \geq s_j, \sum_{s=1}^t X_s^1 \mathbb{1}_{\pi_s=1} > -\frac{T^{1/4}+1}{K}\right)$$

$$+ O\left(\exp\left(-\frac{T^{1/4}\mu_1^2}{2K}\right)\right). \quad (41)$$

Let us also bound the first term in the decomposition in (40):

$$P\left(\tau(B,\pi) \geq s_j, \tau(T^{1/4}+1,\tilde{\pi}) \geq t-1, t_j = s_j, \pi_{t+s_j} = k, \forall t \geq s_j, \sum_{s=1}^{t} X_s^1 \mathbb{1}_{\pi_s=1} > -\frac{T^{1/4}+1}{K}\right)$$

$$\leq P\left(\pi_{t+s_j} = k, t_j = s_j, C_k(t+s_j) \geq C_1(t+s_j)\right).$$

We can then re-write the previous term in the sum, as

$$\sum_{s_j=T^{1/4}}^{T} \sum_{t=1}^{T-s_j} P\left(\pi_{t+s_j} = k, t_j = s_j, C_k(t+s_j) \geq C_1(t+s_j)\right)$$

$$= \sum_{s_j=T^{1/4}}^{T} \sum_{t=1}^{T-s_j} \mathbb{E}\left[\mathbb{1}_{\pi_{t+s_j}=k, t_j=s_j, C_k(t+s_j) \geq C_1(t+s_j)}\right]$$

$$= \mathbb{E}\left[\sum_{t=1}^{T-t_j} \mathbb{1}_{\pi_{t+t_j}=k, C_k(t+t_j) \geq C_1(t+t_j)}\right]$$

$$\leq \sum_{t=1}^{T} \mathbb{E}\left[\mathbb{1}_{\pi_t=k, C_k(t) \geq C_1(t)}\right]$$

$$= o(T),$$

by Lemma 9. Overall, we can replace in (41) and deduce that

$$\sum_{s_j=T^{1/4}}^{T} \sum_{t=1}^{T-s_j} P\left(\tau(B,\pi) \geq s_j, \tau(T^{1/4}+1,\tilde{\pi}) \geq t-1, t_j = s_j, \pi_{t+s_j} = k\right) = o(T),$$

which straightforwardly implies that

$$(B_k) = o(T).$$

**Study of $\mathbb{E}[t_j]$**

We can decompose

$$\mathbb{E}[t_j] = \mathbb{E}\left[t_j \mathbb{1}_{t_j=\min(\tau(B,\pi),T)+1}\right] + \mathbb{E}\left[t_j \mathbb{1}_{t_j \leq \min(\tau(B,\pi),T)}\right]$$

$$= (T+1) \underbrace{P(t_j = T+1)}_{(C)} + \underbrace{\mathbb{E}\left[(\tau(B,\pi)+1)\mathbb{1}_{t_j=\tau(B,\pi)+1}\right]}_{(D)} + \underbrace{\mathbb{E}\left[t_j \mathbb{1}_{t_j \leq \min(\tau(B,\pi),T)}\right]}_{(E)}.$$

We can first bound the term $(D)$, as

$$(D) \leq \left(\sqrt{T}+1\right) P\left(\tau(B,\pi) \geq \sqrt{T}\right) + (T+1)P\left(\sqrt{T} < \tau(B,\pi) \leq T\right).$$

But since $(P(\tau(B,\pi) \leq n))_{n \geq 1}$ is a sequence which converges to $P(\tau(B,\pi) < \infty)$, we deduce that

$$P\left(\sqrt{T} < \tau(B,\pi) \leq T\right) = P(\tau(B,\pi) \leq T) - P\left(\tau(B,\pi) \leq \sqrt{T}\right) \to_{T \to +\infty} 0,$$

and therefore,

$$(D) = o(T).$$

We then deduce that

$$\mathbb{E}[t_j] = (C) + (E) + o(T).$$

Then, let us study the term $(C)$ and bound it by:

$$(C) \leq P\left(\tau(B,\pi) \geq T, \ \forall t \leq T, B + \sum_{s=1}^{t} X_s^{\pi_s} \leq T^{\frac{1}{4}}, \ t_j = T+1\right)$$

$$\leq P\left(\tau(B,\pi) \geq T, \ B + \sum_{s=1}^{T^{3/4}} X_s^{\pi_s} \leq T^{\frac{1}{4}}, \ t_j = T+1\right)$$

$$\leq P\left(\tau(B,\pi) \geq T^{\frac{3}{4}}, \ B + \sum_{s=1}^{T^{3/4}} X_s^{\pi_s} \leq T^{\frac{1}{4}}, \ t_j \geq T^{\frac{3}{4}}\right). \tag{42}$$

Then, let us study the term $(E)$. Actually,

$$P(t_j \geq T^{3/4}, t_j \leq \min(\tau(B,\pi),T)) = P\left(\tau(B,\pi) \geq T^{\frac{3}{4}}, \ \forall t \leq T^{3/4}, B + \sum_{s=1}^{t} X_s^{\pi_s} \leq T^{1/4}, \ t_j \geq T^{\frac{3}{4}}\right)$$

$$\leq P\left(\tau(B,\pi) \geq T^{\frac{3}{4}}, \ B + \sum_{s=1}^{T^{3/4}} X_s^{\pi_s} \leq T^{1/4}, \ t_j \geq T^{\frac{3}{4}}\right).$$

We then deduce that

$$(E) \leq TP\left(t_j \geq T^{3/4}, t_j \leq \min(\tau(B,\pi),T)\right) + T^{3/4} P\left(t_j < T^{3/4}, t_j \leq \min(\tau(B,\pi),T)\right)$$

$$= TP\left(\tau(B,\pi) \geq T^{\frac{3}{4}}, \ B + \sum_{s=1}^{T^{3/4}} X_s^{\pi_s} \leq T^{1/4}, \ t_j \geq T^{\frac{3}{4}}\right) + o(T). \tag{43}$$

Using (42) and (43), we deduce that

$$\mathbb{E}[t_j] \leq 2TP\left(\tau(B,\pi) \geq T^{\frac{3}{4}}, \ B + \sum_{s=1}^{T^{3/4}} X_s^{\pi_s} \leq T^{1/4}, \ t_j \geq T^{\frac{3}{4}}\right) + o(T).$$

Then, consider the set of conditions $\left\{\tau(B,\pi) \geq T^{\frac{3}{4}}, \ B + \sum_{s=1}^{T^{3/4}} X_s^{\pi_s} \leq T^{\frac{1}{4}}, \ t_j \geq T^{\frac{3}{4}}\right\}$ and assume there exists an arm $k_0 \in [K]$ such that $\sum_{s=1}^{T^{3/4}} X_s^{\pi_s} \mathbb{1}_{\pi_s=k_0} > T^{\frac{1}{3}}$. Since $T^{3/4} \leq t_j$, we know that for any $k \in [K]$,

$$\sum_{t=1}^{T^{3/4}} X_t^{\pi_t} \mathbb{1}_{\pi_t=k} \geq -\frac{T^{\frac{1}{4}}}{K} - 1,$$

and hence,

$$B + \sum_{t=1}^{T^{3/4}} X_t^{\pi_t} = B + \sum_{t=1}^{T^{3/4}} X_t^{\pi_t} \mathbb{1}_{\pi_t=k_0} + \sum_{k \neq k_0} \sum_{t=1}^{T^{3/4}} X_t^{\pi_t} \mathbb{1}_{\pi_t=k}$$

$$\geq T^{\frac{1}{3}} - \frac{K-1}{K} T^{\frac{1}{4}} - (K-1)$$

$$= \Omega(T^{\frac{1}{3}}),$$

which contradicts the hypothesis $B + \sum_{t=1}^{T^{3/4}} X_t^{\pi_t} \leq T^{\frac{1}{4}}$. We deduce that

$$P\left(\tau(B,\pi) \geq T^{\frac{3}{4}}, \ B + \sum_{s=1}^{T^{3/4}} X_s^{\pi_s} \leq T^{1/4}, \ t_j \geq T^{\frac{3}{4}}\right)$$

$$\leq P\left(\tau(B,\pi) \geq T^{\frac{3}{4}}, \ \forall k \in [K], \sum_{t=1}^{T^{3/4}} X_t^{\pi_t} \mathbb{1}_{\pi_t=k} \leq T^{\frac{1}{3}}\right)$$

$$\leq P\left(\exists k \in [K], \sum_{t=1}^{T^{3/4}} \mathbb{1}_{\pi_t=k} \geq \frac{T^{3/4}}{K} \text{ and } \sum_{t=1}^{T^{3/4}} X_t^{\pi_t} \mathbb{1}_{\pi_t=k} \leq T^{\frac{1}{3}}\right)$$

$$= P\left(\exists k \in [K]: \ -\frac{T^{1/4}}{K} - 1 < \sum_{s=1}^{T^{3/4}} X_s^{\pi_s} \mathbb{1}_{\pi_s=k} < T^{1/3}, \sum_{s=1}^{T^{3/4}} \mathbb{1}_{\pi_s=k} \geq \frac{T^{3/4}}{K}\right)$$

$$\leq P\left(\exists k \in [K]: \ \left|\sum_{s=1}^{T^{3/4}} X_s^{\pi_s} \mathbb{1}_{\pi_s=k}\right| < T^{1/3}, \sum_{s=1}^{T^{3/4}} \mathbb{1}_{\pi_s=k} \geq \frac{T^{3/4}}{K}\right)$$

$$\leq \sum_{k=1}^{K} P\left(\left|\sum_{s=1}^{T^{3/4}} X_s^{\pi_s} \mathbb{1}_{\pi_s=k}\right| < T^{1/3}, \sum_{s=1}^{T^{3/4}} \mathbb{1}_{\pi_s=k} \geq \frac{T^{3/4}}{K}\right).$$

Then, for any arm $k \in [K]$, there are two cases: either $\mathbb{E}[X_1^k] \neq 0$, or $\mathbb{E}[X_1^k] = 0$. In the former case, we can use Hoeffding's inequality to bound the above probability:

$$P\left(\left|\sum_{s=1}^{T^{3/4}} X_s^{\pi_s} \mathbb{1}_{\pi_s=k}\right| < T^{1/3}, \sum_{s=1}^{T^{3/4}} \mathbb{1}_{\pi_s=k} \geq \frac{T^{3/4}}{K}\right)$$

$$\leq P\left(\left|\sum_{s=1}^{T^{3/4}} X_s^{\pi_s} \mathbb{1}_{\pi_s=k} - \mathbb{E}[X_1^k]\right| < \frac{K}{T^{5/12}} - \mathbb{E}[X_1^k], \sum_{s=1}^{T^{3/4}} \mathbb{1}_{\pi_s=k} \geq \frac{T^{3/4}}{K}\right)$$

$$\leq \exp\left(\frac{T^{3/4}\left(\frac{K}{T^{5/12}} - \mathbb{E}[X_1^k]\right)^2}{4}\right)$$

$$= o(1).$$

In the latter case, we can bound this probability as follows:

$$P\left(\left|\sum_{s=1}^{T^{3/4}} X_s^{\pi_s} \mathbb{1}_{\pi_s=k}\right| < T^{1/3}, \sum_{s=1}^{T^{3/4}} \mathbb{1}_{\pi_s=k} \geq \frac{T^{3/4}}{K}\right) \leq P\left(\left|\frac{\sum_{s=1}^{T^{3/4}} X_s^{\pi_s} \mathbb{1}_{\pi_s=k}}{\sqrt{\sum_{s=1}^{T^{3/4}} \mathbb{1}_{\pi_s=k}}}\right| < \frac{K}{T^{1/24}}, \sum_{s=1}^{T^{3/4}} \mathbb{1}_{\pi_s=k} \geq \frac{T^{3/4}}{K}\right).$$

Then, under the assumption that there is no zero arm, $\text{Var}(X_1^k) > 0$ and

$$\frac{\sum_{s=1}^{T^{3/4}} X_s^{\pi_s} \mathbb{1}_{\pi_s=k}}{\sqrt{\sum_{s=1}^{T^{3/4}} \mathbb{1}_{\pi_s=k}}} \xrightarrow{d} \mathcal{N}(0, \text{Var}(X_1^k)),$$

and since $\frac{1}{T^{1/24}} \xrightarrow{T \to +\infty} 0$, we deduce that

$$P\left(t_j \geq T^{3/4}\right) = o(1).$$

In any case, we have that

$$P\left(\left|\sum_{s=1}^{T^{3/4}} X_s^{\pi_s} \mathbb{1}_{\pi_s=k}\right| < T^{1/3}, \sum_{s=1}^{T^{3/4}} \mathbb{1}_{\pi_s=k} \geq \frac{T^{3/4}}{K}\right) = o(1),$$

and hence,

$$P\left(\tau(B,\pi) \geq T^{\frac{3}{4}},\ B + \sum_{s=1}^{T^{3/4}} X_s^{\pi_s} \leq T^{1/4},\ t_j \geq T^{\frac{3}{4}}\right) = o(1).$$

We then deduce that

$$\mathbb{E}[t_j] = o(T).$$

**Study of** $(A)$

This term is the main one in the previous decomposition.

$$\mathbb{E}\left[\sum_{t=1}^{T} \mathbb{1}_{\tau(B,\pi)\geq t-1}\right] = \mathbb{E}\left[\sum_{t=1}^{T} \mathbb{1}_{\pi_t=1} \mathbb{1}_{\tau(B,\pi)\geq t-1}\right] + \sum_{k=2}^{K} \mathbb{E}\left[\sum_{t=1}^{T} \mathbb{1}_{\pi_t=k} \mathbb{1}_{\tau(B,\pi)\geq t-1}\right]$$

$$= (A) + \sum_{k=2}^{K} \mathbb{E}\left[\sum_{t=1}^{t_j} \mathbb{1}_{\pi_t=k} \mathbb{1}_{\tau(B,\pi)\geq t-1}\right] + \sum_{k=2}^{K} \mathbb{E}\left[\sum_{t=t_j+1}^{T} \mathbb{1}_{\pi_t=k} \mathbb{1}_{\tau(B,\pi)\geq t-1}\right]$$

$$= (A) + \sum_{k=2}^{K} (B_k) + O(\mathbb{E}[t_j]).$$

But then, using the previous bounds on $(B_k)$ and $\mathbb{E}[t_j]$, we deduce that

$$(A) = \mathbb{E}\left[\sum_{t=1}^{T} \mathbb{1}_{\tau(B,\pi)\geq t-1}\right] + o(T).$$

Then, we can simply replace the factor with the expectation by the probability of survival, as

$$\mathbb{E}\left[\sum_{t=1}^{T} \mathbb{1}_{\tau(B,\pi)\geq t-1}\right] = \sum_{t=1}^{T} P(\tau(B,\pi) \geq t-1)$$
$$= T P(\tau(B,\pi) = \infty) + o(T).$$

Hence,

$$(A) = \mu_1 P(\tau(B,\pi) = \infty)T + o(T),$$

which concludes the proof of the proposition. ∎

# I Proof of the Pareto Optimality of EXPLOIT-UCB-DOUBLE (Theorem 3)

In the proofs of the performance of EXPLOIT-UCB-DOUBLE, for the sake of clarity, we drop the exponent $n$ in the notation of EXPLOIT-UCB-DOUBLE, and $\pi^n$ becomes $\pi$. The main objective of this section is to prove that EXPLOIT-UCB-DOUBLE is regret-wise Pareto-optimal in the case of rewards in $\{-1, 0, 1\}$ and with parameter $n = \log T$.

The first subsection provides a preliminary lemma, useful for the proof of the Pareto-optimality exposed in the second subsection. The last subsection makes use of the preliminary lemma to derive an upper bound on the relative regret of EXPLOIT-UCB-DOUBLE in the general case.

### I.1 Preliminary Lemma

**Lemma 10.** *Let $\pi$ be any policy. Then, it holds that*

$$\text{Rew}_T(\pi) \leq P(\tau(B,\pi) \geq \sqrt{T}) \times \max_{k \in [K]} \mu_k T + o(T). \tag{44}$$

*Furthermore, if $\pi$ is an anytime policy, it holds that*

$$\text{Rew}_T(\pi) = P(\tau(B,\pi) = \infty) \mathbb{E}\left[\sum_{t=1}^{T} X_t^{\pi_t} \Big| \tau(B,\pi) \geq T\right] + o(T).$$

*Proof.* In order to prove the first statement of the lemma, we decompose the expected cumulative reward as follows:

$$\text{Rew}_T(\pi) = \mathbb{E}\left[\sum_{t=1}^{\tau(B,\pi)} X_t^{\pi_t} \mathbb{1}_{\tau(B,\pi) < \sqrt{T}}\right] + \mathbb{E}\left[\sum_{t=1}^{T} X_t^{\pi_t} \mathbb{1}_{\tau(B,\pi) \geq t-1} \mathbb{1}_{\tau(B,\pi) \geq \sqrt{T}}\right]$$

$$\leq -B + P(\tau(B,\pi) \geq \sqrt{T}) \times \mathbb{E}\left[\sum_{t=1}^{T} X_t^{\pi_t} \mathbb{1}_{\tau(B,\pi) \geq t-1} \Big| \tau(B,\pi) \geq \sqrt{T}\right]$$

$$\leq -B + \sqrt{T} + P(\tau(B,\pi) \geq \sqrt{T}) \times \mathbb{E}\left[\sum_{t=\sqrt{T}+1}^{T} X_t^{\pi_t} \mathbb{1}_{\tau(B,\pi) \geq t-1} \Big| \tau(B,\pi) \geq \sqrt{T}\right]$$

$$\leq -B + \sqrt{T} + P(\tau(B,\pi) \geq \sqrt{T}) \times \max_{k \in [K]} \mu_k (T - \sqrt{T})$$

$$\leq P(\tau(B,\pi) \geq \sqrt{T}) \times \max_{k \in [K]} \mu_k T + 2\sqrt{T},$$

which gives the desired result. For the second statement, we start by writing the reward as

$$\text{Rew}_T(\pi) = \mathbb{E}\left[\sum_{t=1}^{T} X_t^{\pi_t} \mathbb{1}_{\tau(B,\pi) \geq t-1}\right]$$

$$= \mathbb{E}\left[\sum_{t=1}^{T} X_t^{\pi_t} \mathbb{1}_{\tau(B,\pi) \geq t-1} \mathbb{1}_{\tau(B,\pi) < T}\right] + \mathbb{E}\left[\sum_{t=1}^{T} X_t^{\pi_t} \mathbb{1}_{\tau(B,\pi) \geq T}\right]$$

$$= \mathbb{E}\left[\sum_{t=1}^{\tau(B,\pi)} X_t^{\pi_t} \mathbb{1}_{\tau(B,\pi) < T}\right] + \mathbb{E}\left[\sum_{t=1}^{T} X_t^{\pi_t} \mathbb{1}_{\tau(B,\pi) \geq T}\right].$$

By definition of $\tau(B,\pi)$,

$$\mathbb{E}\left[\sum_{t=1}^{\tau(B,\pi)} X_t^{\pi_t} \mathbb{1}_{\tau(B,\pi) < T}\right] \leq -B < \mathbb{E}\left[\sum_{t=1}^{\tau(B,\pi)-1} X_t^{\pi_t} \mathbb{1}_{\tau(B,\pi) < T}\right],$$

and since the rewards are bounded in $[-1, 1]$, we deduce that

$$-(B+1) < \mathbb{E}\left[\sum_{t=1}^{\tau(B,\pi)} X_t^{\pi_t} \mathbb{1}_{\tau(B,\pi) < T}\right] \leq -B,$$

which implies

$$\text{Rew}_T(\pi) = \mathbb{E}\left[\sum_{t=1}^{T} X_t^{\pi_t} \mathbb{1}_{\tau(B,\pi) \geq T}\right] + o(T).$$

It is trivial that, for any anytime policy $\pi$,

$$P(\tau(B,\pi) \geq T) = P(\tau(B,\pi) = \infty) + o_{T \to +\infty}(1). \tag{45}$$

This implies

$$\text{Rew}_T(\pi) = \mathbb{E}\left[\sum_{t=1}^T X_t^{\pi_t} \mathbb{1}_{\tau(B,\pi) \geq T}\right] + o(T)$$

$$= P(\tau(B,\pi) \geq T)\mathbb{E}\left[\sum_{t=1}^T X_t^{\pi_t} \middle| \tau(B,\pi) \geq T\right] + o(T)$$

$$= P(\tau(B,\pi) = \infty)\mathbb{E}\left[\sum_{t=1}^T X_t^{\pi_t} \middle| \tau(B,\pi) \geq T\right] + o(T),$$

which concludes the proof of the lemma. $\qquad\square$

## I.2 Proof of the Pareto Optimality

We denote by $\pi^n$ the anytime policy EXPLOIT-UCB-DOUBLE with parameter $n \geq 1$. Then, Propositions 5 and 6, along with Lemma 10 give the cumulative reward of EXPLOIT-UCB-DOUBLE:

$$\text{Rew}_T(\pi^n) \geq \left(1 - p^{\text{EX}} - \frac{(p^{\text{EX}})^{nB}}{1 - (p^{\text{EX}})^{nB}}\right) \max_{k \in [K]} \mu_k T + o(T).$$

In particular, we deduce the reward of the (non-anytime) policy EXPLOIT-UCB-DOUBLE with parameter $n = \log T$:

$$\text{Rew}_T(\pi^{\log T}) = \left(1 - p^{\text{EX}}\right) \max_{k \in [K]} \mu_k T + o(T).$$

Recall from Lemma 10 that, for any policy $\tilde{\pi}$,

$$\text{Rew}_T(\tilde{\pi}^T) \leq P\left(\tau(B, \tilde{\pi}^T) \geq \sqrt{T}\right) \max_{k \in [K]} \mu_k T + o(T),$$

and as a result,

$$\frac{\text{Rew}_T(\tilde{\pi}^T) - \text{Rew}_T(\pi^{\log T})}{T} \leq \left(P\left(\tau(B, \tilde{\pi}^T) \geq \sqrt{T}\right) - (1 - p^{\text{EX}})\right) \max_{k \in [K]} \mu_k + o(1).$$

For $T \geq \frac{9B^2}{\Delta_F^2}$, it holds that

$$\frac{\text{Rew}_T(\tilde{\pi}^T) - \text{Rew}_T(\pi^{\log T})}{T} \leq \left(P\left(\tau(B, \tilde{\pi}^T) \geq \frac{3B}{\Delta_F}\right) - (1 - p^{\text{EX}})\right) \max_{k \in [K]} \mu_k + o(1),$$

and taking the limit gives

$$\text{Reg}_F(\pi \| \tilde{\pi}) \leq \left(p^{\text{EX}} - P\left(\tau(B, \tilde{\pi}^T) < \frac{3B}{\Delta_F}\right)\right) \max_{k \in [K]} \mu_k,$$

where $\pi$ denotes $(\pi^{\log T})_{T \geq 1}$ the optimally-tuned EXPLOIT-UCB-DOUBLE. Then, assume that

$$\sup_F \text{Reg}_F(\pi \| \tilde{\pi}) > 0,$$

which implies that there exists some arm distributions $\tilde{F}$ such that

$$p^{\text{EX}} - P_{\tilde{F}}\left(\tau(B, \tilde{\pi}^T) < \frac{3B}{\Delta_{\tilde{F}}}\right) > 0.$$

Then, by Theorem 1, there exist some other arm distributions $F$ such that

$$p^{\text{EX}} - P_F\left(\tau(B, \tilde{\pi}^T) < \frac{3B}{\Delta_F}\right) < 0,$$

and since $\max_{k \in [K]} \mu_k > 0$, this implies

$$\inf_F \text{Reg}_F(\pi \| \tilde{\pi}) < 0,$$

proving that $(\pi^{\log T})_{T \geq 1}$ is regret-wise Pareto-optimal.

### I.3  Relative Regret of EXPLOIT-UCB-DOUBLE in the General Case

In the general case, Propositions 5 and 6 and Lemma 10 imply that the cumulative reward of EXPLOIT-UCB-DOUBLE $\pi^n$ with parameter $n \geq 1$ is

$$\text{Rew}_T(\pi^n) \geq \left(1 - p^{\text{EX}} - \frac{\left(p^{\text{EX}}\right)^{nB}}{1 - \left(p^{\text{EX}}\right)^{nB}}\right) \max_{k \in [K]} \mu_k T + o(T). \tag{46}$$

Let $\pi'$ be any policy. The previous result implies that there exist some arm distributions $F$ such that

$$\lim_{T \to +\infty} \frac{\text{Rew}_T(\pi') - \left(1 - p^{\text{EX}}\right) \max_{k \in [K]} \mu_k T}{T} < 0.$$

With (46), this implies that EXPLOIT-UCB-DOUBLE $\pi^n$ achieves, for any policy $\pi'$,

$$\inf_F \text{Reg}_F(\pi^n \| \pi') = \inf_F \lim_{T \to +\infty} \frac{\text{Rew}_T(\pi') - \text{Rew}_T(\pi^n)}{T} < \frac{\left(p^{\text{EX}}\right)^{nB}}{1 - \left(p^{\text{EX}}\right)^{nB}} \max_{k \in [K]} \mu_k.$$

