# OpenReview forum: "The Survival Bandit Problem"
_TMLR — Accepted by TMLR_

### Review · Reviewer_DNsy · 2024-04-03

**Summary Of Contributions:**

This paper studies a novel extension of the stochastic Multi-Armed Bandit problem, namely the Survival Bandit problem. This extension introduces the notion of _ruin_, where the agent has to stop its trial whenever its cumulative reward falls below a certain threshold. The authors prove negative results on this setting, most importantly, the impossibility of sub-linear _survival_ regret for any policy. Thus, the goal becomes to provide algorithm that are optimal _in the senso of Pareto_. Authors characterize a class of algorithms called EXPLOIT, and show that when certain conditions are met then the probability of ruin can be bounded with a small value, matching the provided lower bound. Finally, exploiting a mechanism similar to doubling trick, authors incorporate a UCB-like mechanism in the EXPLOIT class of algorithms, obtaining a nearly optimal asymptotic survival regret and an optimal survival regret in the sense of Pareto.

**Audience:**

Yes

**Broader Impact Concerns:**

I have no ethical concerns with this paper, since it is a theoretical work.

**Claims And Evidence:**

Yes

**Requested Changes:**

- Presentation (this is a major point, however I sorted the points from the most important to the least)
    - Remove or modify Section 4.1. It is okay to provide an overview of paper’s main idea, but I don’t feel that this is the right way to do so. Probably some figures would be helpful.
    - Extend Subsection 5.2. It is very important to provide a grasp of what $\gamma(F_k)$ represents, possibly providing examples of its behavior in specific cases that are easy to visualize and communicate. Also, a longer comment on the implications of Lemma 1 would be useful: it would by ideal to connect an example of $\gamma(F_k)$ from Definition 6 to the result of Lemma 1, explicitly commenting on its behavior.
    - Since the most interesting part of this work comes after Section 4 (i.e., main theoretical results), I suggest the authors to shrink the first part to better highlight the second one.
    - Extend the experimental campaign in the main paper.
    - Find a better representation of Algorithm 2 than Figure 1.

- If I understood correctly, you assume that there is always an arm avoiding ruin (Assumption 3). Of course, without this assumption, an asymptotical analysis would lose a lot of meaning. However, I consider the scenario in which Assumption 3 does not hold to be very important: would it be possible to relax this assumption and provide result on a weaker notion of regret, e.g. regret before ruin? Moreover, under Assumption 3, would it be possible to define a new class of algorithms leveraging a Best-Arm Identification-like mechanism to identify the “safe” arm as quick as possible, and then trying to maximize the reward using algorithms dealing with long-term budget constraints (since, assuming that an arm capable of replenish your budget constraints has been identified, the algorithm can now spend it to explore more safely). This is not a critical point, but I would expand the discussion on Assumption 3 and bring it to the main paper.

- In Remark 3, you state that using a priori knowledge it would be possible to guide an EXPLOIT policy by setting specific values on the $B_k$. However, no example is provided of this and also it is unclear what form of prior information would lead to actual improvements in the performance. Thus, it would be beneficial to support this remark with some examples or more detailed description of the procedure.

- In Conjecture 1 you claim that for $\mathcal{F}\_{\\{-1,0,11\\}}$ no anytime policy is regret-wise Pareto optimal. However in Conjecture 2 you claim an algorithm is Pareto-optimal for $\mathcal{F}\_{[-1,1]}$. I think that the main point here is that EXPLOIT-UCB-DOUBLE is not anytime, and thus optimality can be achieved even in a more complex setting than $\{-1,0,1\}$. Are there any additional intuitions on why this may hold? I’m actually not so convinced by Conjecture 2. If you want to maintain Conjecture 2 as formal conjecture in this work, this is an important point for me.

**Strengths And Weaknesses:**

- Strengths
    - The proposed setting is a direct extension of the standard stochastic MAB problem and is motivated by applications.
    - The paper performs a thorough theoretical analysis of the setting, starting from negative results and providing an alternative point of view (i.e., optimality in the sense of Pareto), then providing positive results in that direction.
    - Authors provide a good literature review and the paper is properly compared to existing literature.
    - Even though I didn’t check all the proofs in details, the derivations seem correct and sound.

- Weaknesses
    - The presentation is a major issue of this work, especially in certain sections. In particular:
        - Section 4.1 is unnecessary, and providing calculations with the premise of them being incorrect is pointless in my opinion. Moreover, while I understand that the paper is still in its introduction, I would prefer avoiding terms that are too colloquial, such as “not too stupid” or “trapped”.
        - Section 4.3, namely “Structure of the Paper”, begins at page 11 when the paper’s length is slightly more than 24 pages. Since the most interesting part of this work comes after Section 4 (I.e., main theoretical results), I suggest the authors to shrink the first part to better highlight the second one.
        - In Section 5, authors introduce key concepts for the S-MAB setting. In particular, Definition 6 and Lemma 1 play a key role in the subsequent sections. However, I feel that the presentation of Subsection 5.2 has to be extended, and the authors need to comment.
        - Figure 1 is hardly understandable for me. Maybe you can consider alternative forms of visualization.
        - The experimental section of the main paper is just Subsection 7.3. Since experiments provide interesting insights on the algorithms’ behavior, I feel that more space in the main paper can be dedicated to them.

    - I have some questions on Assumption 3, that can be found in the subsequent section.

---

> ### Author Response · Authors · 2024-05-30
>
> Thank you very much for your insightful comments and questions.
>
> **« Remove or modify Section 4.1. It is okay to provide an overview of paper’s main idea, but I don’t feel that this is the right way to do so. Probably some figures would be helpful. »**
>
> Thank you for your comment. We have considerably shortened Section 4.1. Now, we focus on the intuition why we study the probability of ruin, and we illustrate it with a figure, as you suggested.
>
> **« Extend Subsection 5.2. It is very important to provide a grasp of what $\gamma(F_k)$ represents, possibly providing examples of its behavior in specific cases that are easy to visualize and communicate. Also, a longer comment on the implications of Lemma 1 would be useful: it would be ideal to connect an example of $\gamma(F_k)$ from Definition 6 to the result of Lemma 1, explicitly commenting on its behavior. »**
>
> Thank you for your comment. We have provided an intuition on $\gamma(F_k) \in [0, +\infty]$ as a measure of the risk of ruin of the policy $\pi$ which constantly selects arm $F_k$. It is equal to $0$ when the probability of ruin is $1$, and to $+\infty$ when the risk of ruin is $0$. We have further included a figure which represents the evolution of $\gamma(F_k)$ as a function of the parameters of $F_k$ in the multinomial case (considered in the major part of our work).
>
> **« Presentation: Since the most interesting part of this work comes after Section 4 (i.e., main theoretical results), I suggest the authors to shrink the first part to better highlight the second one. »**
>
> Thank you for your comment. We have largely shrunk the first part of the draft, particularly Section 4.
>
> **« Extend the experimental campaign in the main paper. »**
>
> Thank you for your comment. We have added many experiments, which corroborate our theoretical results. We have further emphasized each phenomenon that appears through the experiments, and thanks to your comment, the experiment section is now more informative.
>
> **« Find a better representation of Algorithm 2 than Figure 1. »**
>
> Thank you for your comment. We changed the representation of Algorithm 2 for a better one, which shows how it behaves through time.
>
> **« If I understood correctly, you assume that there is always an arm avoiding ruin (Assumption 3). Of course, without this assumption, an asymptotical analysis would lose a lot of meaning. However, I consider the scenario in which Assumption 3 does not hold to be very important: would it be possible to relax this assumption and provide result on a weaker notion of regret, e.g. regret before ruin?»**
>
> Thank you for your comment. This is indeed an interesting discussion, and following your comment, we have included a discussion about this point in Section 7.2.
>
> We suppose you mean Assumption 1. To make the discussion clearer, we consider the multinomial case and assume that the initial budget $B$ is an integer.
>
> In most applications of bandit problems, the standard objective is to maximize the expected cumulative reward. Without the budget constraint, it is known that
> 	$$\text{Rew} = \mu^* T + o(T)$$
> where $\mu^*$ denotes the largest arm expectation. The expected cumulative regret is only the remaining part of the asymptotic development (in $T$) of the expected cumulative reward. At the time of ruin, the cumulative reward is always equal to $-B$ by definition, and therefore, at the time $\min(\tau, T)$, the expected cumulative reward is close to $-B$, up to small term. Precisely, we can study this small term by defining the regret as
> 	$$-B - \text{Rew} = -(1-P(\tau(B, \pi)\leq T)) (E[\sum_{t=1}^{T} X_t^{\pi_t} | \tau(B, \pi)>T] + B).$$
>
> From an application’s perspective, the gain brought by such a study is very small and hence, hard to justify (it is smaller than the budget $B$, viewed as a constant compared to the large horizon $T$). Yet, from a theoretical perspective, such a study correspond to the maximization of the survival time in a hostile environment (no arm avoids the ruin), and is an interesting future research direction that we included in the draft.

---

> > ### Author Response · Authors · 2024-05-30
> >
> > **« Moreover, under Assumption 3, would it be possible to define a new class of algorithms leveraging a Best-Arm Identification-like mechanism to identify the “safe” arm as quick as possible, and then trying to maximize the reward using algorithms dealing with long-term budget constraints (since, assuming that an arm capable of replenish your budget constraints has been identified, the algorithm can now spend it to explore more safely). »**
> >
> > Thank you for your comment. The method you describe consists in a type of ETC (Explore-Then-Commit), where the first phase is a budget-constrained exploration phase which aims at discovering a safe arm $k_s$ (i.e. an arm with high $\gamma(F_k)$), and the second phase is a standard exploration-exploitation with the advantage of having a safe arm $k_s$ to select when the budget runs low. While we have not tried to implement it, we would like to give two intuitions on why such a method may not reach the desired outcome:
> > - in the S-MAB, we first need to exploit in order to minimize the risk of ruin. Therefore, the early exploration should be minimal, and this suggests avoiding an early pure exploration phase, or at least limiting it in terms of budget consumption. In that case, the probability that $k_s$ is really safe is not so large, thus maintaining a large risk of ruin.
> > - even if a safe arm $k_s$ is correctly identified, its risk of ruin may be large. Indeed, while the probability of ruin of the safest arm is smaller than $1$ under Assumption 1, it may still be large. In such cases, no policy can avoid the ruin with large probability, let alone to replenish the budget when it runs low. This is in contrast with Kumar and Kleinberg (2022), who assume the existence of a positive arm which can replenish the budget.
> >
> > **« In Remark 3, you state that using a priori knowledge it would be possible to guide an EXPLOIT policy by setting specific values on the $B_k$. However, no example is provided of this and also it is unclear what form of prior information would lead to actual improvements in the performance. Thus, it would be beneficial to support this remark with some examples or more detailed description of the procedure. »**
> >
> > Thank you for your comment. Following your comment, we have added an example of situation where utilizing prior information may help to design better-than-uniform budget shares.
> >
> > Taking a Bayesian viewpoint on the problem, if we assume that there is a prior distribution $D_k$ for $\gamma(F_k)$ through, e.g., already observed samples. Then the expected probability of ruin is given by
> > 	$$E_{\gamma(F_k)\sim D_k}\left[\prod_{k=1}^K \exp\left(-B_k \gamma(F_k)\right)\right].$$
> >
> > When $D_k$ is the same for all the arms $k$, the above is trivially minimized for $B_k = B/K$. However, when this is not the case and when priors for $\gamma(F_k)$ are independent, the optimal $B_k$ is the one such that
> > 	$$ \frac{E\left[\gamma(F_k) \exp\left(-B_k \gamma(F_k)\right)\right]}{E\left[\exp\left(-B_k \gamma(F_k)\right)\right]} $$
> > becomes the same between arms.
> >
> > **« In Conjecture 1 you claim that for $F_{\{-1,0,1\}}$ no anytime policy is regret-wise Pareto optimal. However in Conjecture 2 you claim an algorithm is Pareto-optimal for $F_{[-1,1]}$. I think that the main point here is that EXPLOIT-UCB-DOUBLE is not anytime, and thus optimality can be achieved even in a more complex setting than $\{-1,0,1\}$. Are there any additional intuitions on why this may hold? I’m actually not so convinced by Conjecture 2. If you want to maintain Conjecture 2 as formal conjecture in this work, this is an important point for me. »**
> >
> > Thank you for your comment. You are absolutely correct. The crucial point is « anytime », and in Conjecture 2, we conjectured that non-anytime EXPLOIT-UCB-DOUBLE is Pareto optimal, even in the more complicated setting $F_{[-1, 1]}$.
> >
> > Following your comment, we have removed the formal Conjecture 2 and informally raised the open question of its regret-wise Pareto-optimality, in a more objective manner without leaning more toward the « yes » or the « no ».

---

> > > ### Author Response · Authors · 2024-06-27
> > >
> > > To address reviewer SZXv's comment, we have added Remark 8 (top of p21). The rest of the draft is entirely unchanged since the last modification.

---

### Review · Reviewer_byTA · 2024-04-20

**Summary Of Contributions:**

This paper studies the $K$-armed bandit setting where the bandit procedure gets interrupted if the cumulative reward falls below zero. They define the idea of survival regret which depends on the time of ruin, that is the first time the cumulative reward falls below a budget $B$. They discuss two approaches that might minimize the survival regret, and adopt the second approach. In this approach, they show that if the policy $\pi^T$ has managed to survive after, e.g., $\sqrt{T}$ rounds, then the risk of ruin almost disappears. Therefore, a good policy $\pi^T$ must first maximize $P\left(\tau\left(B, \pi^T\right)=\infty\right)$ until the risk of ruin disappears, then, maximize $\mathbb{E}\left[\sum_{t=1}^T X_t^{\pi_t^T}\right]$. They also set themselves apart from Perotto et al. (2022) (which works for $\{-1, 1\}$ rewards and Manome et. al (2023) (which looks into Bernoulli rewards) by analyzing multinomial arm distributions in this work. They start with providing an insightful lower bound to this setting and then introduce a framework of anytime policies, called EXPLOIT, which achieves the lower bound on the probability of ruin given in Theorem 1. They provide a regret upper bound in Proposition 3 for all policies that follow the EXPLOIT framework.

**Audience:**

Yes

**Broader Impact Concerns:**

Not applicable.

**Claims And Evidence:**

Yes

**Requested Changes:**

See the weakness section.

**Strengths And Weaknesses:**

Strengths:
1. The paper is well-written. I like the way the authors introduce the idea, slowly build up the story by providing the lower bound and then propose their variants.
2. The lower bound is interesting, however, it needs more discussions (see weakness).
3. They propose a new framework and give a regret upper bound for policies that follow that framework.




Weakness:
1. The setting also has a connection to [conservative bandits](https://arxiv.org/pdf/1602.04282.pdf) where the agent must ensure that the cumulative reward is above a threshold with high probability, and [thresholding bandits](https://arxiv.org/pdf/1605.08671.pdf). This should be discussed in detail.
2. More discussions are needed for the lower bound. Specifically, I am confused is this a completely separate result from Perotto et al. (2022) and Manome et. al (2023)?
3. More discussions are needed on the algorithmic approaches of the closest work of Perotto et al. (2022) and Manome et. al (2023). Do they also propose a UCB-based approach? How is then Algorithm 1 of Exploit-UCB different from them?
4. In the experiment section why is the performance of MTS worse than UCB? This is quite counter-intuitive. Generally in standard bandits, the opposite is observed. Please clarify this.
5. What are the baselines from Perotto et al. (2022) and Manome et. al (2023)? Have they been implemented?

---

> ### Author Response · Authors · 2024-05-30
>
> Thank you very much for your insightful comments and questions.
>
> **« The setting also has a connection to conservative bandits where the agent must ensure that the cumulative reward is above a threshold with high probability, and thresholding bandits. This should be discussed in detail. »**
>
> Thank you for your comment. We have made the following changes:
> - conservative bandits: we have added a longer and more in-depth discussion (Wu et al. 2016, Kazerouni et al. 2017, Garcelon et al. 2020)
> - thresholding bandits: we have now added a discussion on this important topic and its connection to the S-MAB. We further compared it to the existing literature (Locatelli et al. 2016, Tao et al. 2019).
>
> **« More discussions are needed for the lower bound. Specifically, I am confused is this a completely separate result from Perotto et al. (2022) and Manome et. al (2023)? »**
>
> Thank you for your comment. Both Perotto et al. (2022) and Manome et al. (2023) propose interesting algorithms to tackle the problem in the case of Bernoulli rewards. Their results are very interesting, however, they are purely experimental, and contain no theoretical result or bound. We have made it clear in the beginning of the literature review (Section 2).
>
> Following your comment, we have added Section 5.3 covering a discussion on the lower bound and the interpretation of the $\gamma(F_k)$ (including a figure), hence bringing more insight for the reader. In particular, we have discussed two important aspects:
> the connection between the Pareto-type lower bound and game theory;
> the interpretation (with a figure) of $\gamma(F_k)$ as a measure of the chance of survival of the constant policy which selects arm $k$;
> the bound is a product of the arms’ probability of ruin with budget shares, and hence suggests splitting the budget between the arms. This part of the discussion has been detailed and moved to Section 5.3, entirely dedicated to the interpretation of the lower bound.
>
> **« More discussions are needed on the algorithmic approaches of the closest work of Perotto et al. (2022) and Manome et. al (2023). Do they also propose a UCB-based approach? How is then Algorithm 1 of Exploit-UCB different from them? »**
>
> Thank you for your comment. We have added Section 7.4 to compare our results to those of Perotto et al. (2022) and Manome et al. (2023).
>
> While Perotto et al. (2022) and Manome et al. (2023) both use a UCB-based approach to the S-MAB, they are extremely different from ours. The approach from Perotto et al. (2022) is an empirical one: based on the heuristic that the remaining budget $B_t$ may be used as an exploration parameter, they introduced algorithms Gambler-UCB1 by replacing $\log t$ by $\log B_t$ in UCB1. The approach from Manome et al. (2023) is an experimental one: they introduced algorithm GWA-UCB1, which is a generalization of UCB which selects the arm maximizing a generalized weighted average of the empirical average and the UCB1 exploration term.
>
> In contrast, our work is a theoretically grounded one: we defined the first formal setting of the S-MAB, introduced the probability of ruin, showed a lower bound and derived matching policies, including EXPLOIT-UCB. And then, based on the boundedness of the reward of EXPLOIT-UCB, we increased the exploration and derived EXPLOIT-UCB-DOUBLE. Therefore, EXPLOIT-UCB and EXPLOIT-UCB-DOUBLE are the only S-MAB algorithms which have proven theoretical guarantees. Finally, our experiments show that their practical performance is significantly superior to the one of UCB, MTS, GWA-UCB1 and Gambler-UCB1.
>
> **« In the experiment section why is the performance of MTS worse than UCB? This is quite counter-intuitive. Generally in standard bandits, the opposite is observed. Please clarify this. »**
>
> Thank you for your comment. We have added an explanation to clarify it in the experiment section of the main body.
>
> As you mentioned, in the standard bandit setting, MTS largely outperforms UCB both theoretically and experimentally. However, as a Bayesian (and randomized) algorithm, the first few arm selections are random and used to create additional exploration. In the S-MAB setting, this creates early ruins, and therefore, undermines the survival regret. This fact is especially striking when the initial budget is small, as illustrated in experiments (1)-(4).
>
> Finally, we mention that this is in line with the results of Perotto et al. (2022) and Manome et al. (2023), who both show that Thompson Sampling suffers from frequent early ruins even in the Bernoulli setting.
>
> **« What are the baselines from Perotto et al. (2022) and Manome et. al (2023)? Have they been implemented? »**
>
> Thank you for your comment. We have implemented the baselines of Perotto et al. (2022) and Manome et al. (2023) in the settings we considered. We have also added the discussion related to those baselines. It still appears that EXPLOIT-UCB-DOUBLE outperforms all of the other algorithms in most of the settings considered.

---

> > ### Author Response · Authors · 2024-06-27
> >
> > To address reviewer SZXv's comment, we have added Remark 8 (top of p21). The rest of the draft is entirely unchanged since the last modification.

---

### Review · Reviewer_SZXv · 2024-05-19

**Summary Of Contributions:**

The paper considers a variant of the multi-armed bandit problem that stops if the cumulative reward falls below a fixed threshold, called budget. The authors claim that the new variant, survival bandit problem, differs significantly from classical bandit variants in the sense that it is not possible to obtain sublinear (classical) regret, and instead one has to consider optimality in the Pareto sense.

A class of Pareto optimal policies, EXPLOIT, is proposed, and UCB variants belonging to this class are shown to be regret-wise Pareto optimal. The theoretical results include the analysis of the new variant, including the linearity of the classical regret of any policy, a Pareto-type bound on the probability of ruin, and the guarantees for the proposed algorithms (Pareto and regret optimality).

A small empirical section evaluates the proposed algorithms on synthetic 3-armed multinomial bandit problems with UCB included as a baseline.

**Audience:**

Yes

**Broader Impact Concerns:**

Not applicable.

**Claims And Evidence:**

No

**Requested Changes:**

In the bandit literature the distinction between the selected arm (often denoted I_t) and the selection policy that may depend on the history is made far more explicit, and that is something that should be done in this paper as well.

**Strengths And Weaknesses:**

The considered bandit variant is relevant, and there is an extensive theoretical work analyzing both the problem and the proposed algorithms. However, I am not convinced by some of the arguments presented in the paper.

Some of the issues possibly stem from a lack of clarity on the policies being adaptive (i.e. depend on the history and as such on the arms' distribution) or not in various statements of the paper. One of the cornerstones of the paper is Proposition 1 (slightly modified as Proposition 10 in Appendix B). The proof starts with a policy sequence ${pi}$ with $k_0$ being an arm that is less frequently chosen in the sequence, and this arm is chosen as the one with highest reward. Such a proof technique can be used only if the policy sequence $\pi$ is fixed/non-adaptive. As such, if the statement of Propositions covers adaptive sequences that depend on the history, then the proof is incorrect. On the other hand, if it refers to fixed sequences, then the statement is weak, since it is obvious that a fixed sequence can be sufficiently suboptimal under an appropriately chosen reward distribution. That would the case equally for most bandit variants, and the general idea of the proof could be used for a more standard bandit variant. Therefore, I do not see the inadequacy of the classic regret framework as sufficiently supported, and the need for Pareto optimality can be questioned.

Obviously, the standard lower bound proof technique which involve to distribuiton slightly different that are difficult to differentiate could be used here as well, but that would require a more elaborate analysis and I am not sure that it would provide the necessary suboptimality guarantee.

The paper argues that for this variant the required strategy is to avoid ruin firstly and then focus on the reward. This is also an argument used for preferring the multinomial to the Bernoulli distribution. Probably a proof of case would be an arm with $P(X_0=0)=1$ and an arm with $P(X_1=1)>P(X_1=-1)>0$. In this case the paper would probably argue for playing arm 0 in the first phase to avoid the ruin, but I do not see the gain of obtaining the 0 value rewards in the beginning, instead of getting no (0) reward at the end by selecting continually arm 1. I assume that in such a case the optimal (and anytime) policy is still to select arm 1, even though it has higher probability of ruin. I did not fully compute the expected reward for the above example, so I could be wrong, and it is quite possible that in some cases the multinomial distribution (or a richer one) will provide case where there is no optimal anytime policy. Nevertheless, the intuition in the example is running counter to the idea of having such a strong emphasis on avoiding ruin.

---

> ### Author Response · Authors · 2024-05-30
>
> Thank you very much for your insightful comments and questions.
>
> **« A class of Pareto optimal policies, EXPLOIT, is proposed, and UCB variants belonging to this class are shown to be regret-wise Pareto optimal »**
>
> Thank you for summarizing. This is almost correct: all the policies in EXPLOIT (including EXPLOIT-UCB) are *ruin-wise* Pareto optimal, but none of them is *regret-wise* Pareto optimal. However, the doubling trick version of EXPLOIT-UCB called EXPLOIT-UCB-DOUBLE is both *ruin-wise* and *regret-wise* Pareto optimal, but it does not belong to EXPLOIT.
>
> **« The proof starts with a policy sequence ${pi}$ with $k_0$ being an arm that is less frequently chosen in the sequence, and this arm is chosen as the one with highest reward. Such a proof technique can be used only if the policy sequence $\pi$ is fixed/non-adaptive. »**
>
> Thank you for the comment and let us clarify this technical aspect of the proof. In the proof, $k_0$ is the least frequently chosen arm **at the first round**, and therefore, it never depends on any observed reward. The idea behind the proof is that choosing a suboptimal arm at the first round leads to a linear regret, defined in the classic sense. We have added a formal definition of the sequences of policies which highlights what variables $\pi^T_t$ may depend on, to bring clarification on this point (see the answer to your requested change below). If the concern is still not solved by this explanation, please point out the specific place of the proof where correctness for adaptive policies is unclear.
>
> **« Probably a proof of case would be an arm with $P(X_0=0)=1$ and an arm with $P(X_1=1)>P(X_1=-1)>0$. In this case the paper would probably argue for playing arm 0 in the first phase to avoid the ruin »**
>
> Thank you for your comment and let us clarify this tricky point on the example you provided.
> in the first phase, EXPLOIT-UCB-DOUBLE performs a standard UCB algorithm. And therefore, it **does** select arm 2 many times.
> in the second phase, there are 2 cases:
> - either the cumulative reward of arm 2 reaches $-B/2$. In this case, the algorithm will then pull arm 1 until round $T$.
> - or the cumulative reward of arm 2 never reaches $-B/2$ (which happens with positive probability). In this case, the algorithm will behave exactly like UCB, and then achieve the same regret as UCB.
>
> Therefore, EXPLOIT-UCB-DOUBLE *will* select arm 2 many times in the first phase.
>
> **Requested change: « In the bandit literature the distinction between the selected arm (often denoted I_t) and the selection policy that may depend on the history is made far more explicit, and that is something that should be done in this paper as well. »**
>
> Thank you for your comment and recommendation, which will help to clarify the paper. We should have made the following more explicit: for any $T\geq 1$ and any $t\in \{1, …, T\}$, $\pi^T_t$ belongs to the following sigma algebra:
> 	$$\sigma(\pi^T_1, X_{T, 1}^{\pi^T_1}, \pi^T_2, X_{T, 2}^{\pi^T_2}, …, \pi^T_{t-1}, X_{T, t-1}^{\pi^T_{t-1}})$$
>
> We have added a formal definition of a sequence of policies in Section 3.2, to make the dependency explicit.

---

> > ### Comment · Reviewer_SZXv · 2024-06-22
> >
> > You are correct, I missed the fact that k_0 depends only on the first step. This does highlight the fact that a policy that is sub-optimal in the first step will already suffer linear classical regret. I am somewhat worried about this fact w.r.t. the how appropriate the setting is.
> >
> > I am still unconvinced by the arguments for multinomial distribution. I think in the example choosing the arm with the positive reward  is still optimal even after reaching -B/2.
> >
> > I appreciate the authors' effort to deal with the comments of the reviewers.

---

> > > ### Author Response · Authors · 2024-06-27
> > >
> > > Thank you very much for the discussion.
> > >
> > > **This does highlight the fact that a policy that is sub-optimal in the first step will already suffer linear classical regret. I am somewhat worried about this fact w.r.t. how appropriate the setting is.**
> > >
> > > We agree that this result might be somewhat counter-intuitive. Still, we think that the theoretical setting should be based on the practical applications (as discussed in Section 1 and, e.g., Perotto et al., 2019 or Shmerling, 2021), and should not be discussed from the impossibility to always achieve a sublinear classical regret.
> > >
> > > **I am still unconvinced by the arguments for multinomial distribution. I think in the example choosing the arm with the positive reward is still optimal even after reaching -B/2.**
> > >
> > > The main problem is that the arm distributions $F_1$ and $F_2$ are unknown, and we only observe the rewards. Assume we observe the rewards:
> > > for arm1: 0, 0, 0, 0, …, 0
> > > for arm 2: +1, -1, -1, 0, +1, -1, … and the sum of those rewards is equal to $-B/2$.
> > >
> > > The true arm distributions $F_1$ and $F_2$ are \textbf{unknown}. For example, they could be:
> > > $F_1 = \delta_0$ which is a zero arm, and $F_2 = \mathrm{Mult}(0.4, 0.1, 0.5)$ which has a positive expectation $E_{F_2}[X] = 0.5-0.4 = 0.1$. In this case, as you said, arm 2 is the best arm.
> > > $F_1 = \mathrm{Mult}(0.01, 0.97, 0.02)$ which has a positive expectation $E_{F_1}[X] = 0.02-0.01 = 0.01$, and $F_2 = \mathrm{Mult}(0.5, 0.1, 0.4)$ which has a negative expectation $E_{F_2}[X] = 0.4-0.5 = -0.1$. In this case, arm 1 is the best arm.
> > >
> > > Therefore, if we choose to pull arm 2 soon after it reaches $-B/2$, we are going to be suboptimal in the second instance.
> > >
> > > However, if we observe a very large number $n$ of rewards from arm 1 all equal to 0, it is reasonable to switch to arm 2 even if it has reached $-B/2$, as suggested. By « large » $n$, we mean that it should tend to infinity as the horizon $T$ increases, e.g., $n = \log T$. But since we are considering the asymptotics in $T$, arm 1 would only generate such a reward stream if it is a zero arm (as in the example you mentioned), but that case does not seem to appear in practice and it was excluded by the assumption in Section 5.1. That said, it would be interesting from a theoretical perspective, and we have added Remark 8 (p 21) to discuss it following your comment.

---

### Decision · Action_Editor_udm4 · 2024-07-10

**Recommendation:** Accept as is

**Comment:**

There was unanimous support for accepting the paper. The revision removed the previous concerns of the reviewers.

**Audience:**

Yes. All reviewers agreed.

**Claims And Evidence:**

Yes. All reviewers agreed.